# Simultaneously achieving giant piezoelectricity and record coercive field enhancement in relaxor-based ferroelectric crystals

Liya Yang[1,2,3,9], Houbing Huang[4,9], Zengzhe Xi[5], Limei Zheng[1✉], Shiqi Xu[4], Gang Tian[1], Yuzhi Zhai[1], Feifei Guo[5], Lingping Kong[6], Yonggang Wang [6], Weiming Lü[7✉], Long Yuan[8], Minglei Zhao[1], Haiwu Zheng[2] & Gang Liu[6✉]

A large coercive field ($E_C$) and ultrahigh piezoelectricity are essential for ferroelectrics used in high-drive electromechanical applications. The discovery of relaxor-PbTiO$_3$ crystals is a recent breakthrough; they currently afford the highest piezoelectricity, but usually with a low $E_C$. Such performance deterioration occurs because high piezoelectricity is interlinked with an easy polarization rotation, subsequently favoring a dipole switch under small fields. Therefore, the search for ferroelectrics with both a large $E_C$ and ultrahigh piezoelectricity has become an imminent challenge. Herein, ternary Pb(Sc$_{1/2}$Nb$_{1/2}$)O$_3$–Pb(Mg$_{1/3}$Nb$_{2/3}$)O$_3$–PbTiO$_3$ crystals are reported, wherein the dispersed local heterogeneity comprises abundant tetragonal phases, affording a $E_C$ of 8.2 kV/cm (greater than that of Pb(Mg$_{1/3}$Nb$_{2/3}$)O$_3$–PbTiO$_3$ by a factor of three) and ultrahigh piezoelectricity ($d_{33}$ = 2630 pC/N; $d_{15}$ = 490 pC/N). The observed $E_C$ enhancement is the largest reported for ultrahigh-piezoelectric materials, providing a simple, practical, and universal route for improving functionalities in ferroelectrics with an atomic-level understanding.

[1] School of Physics, State Key Laboratory of Crystal Materials, Shandong University, 250100 Jinan, China. [2] International Joint Research Laboratory of New Energy Materials and Devices of Henan Province, School of Physics and Electronics, Henan University, 475004 Kaifeng, China. [3] Condensed Matter Science and Technology Institute, School of Instrumentation Science and Engineering, Harbin Institute of Technology, 150080 Harbin, China. [4] School of Materials Science and Engineering & Advanced Research Institute of Multidisciplinary Science, Beijing Institute of Technology, 100081 Beijing, China. [5] School of Materials and Chemical Engineering, Xi'an Technological University, 710032 Xi'an, China. [6] Center for High Pressure Science and Technology Advanced Research, 201203 Shanghai, China. [7] Spintronics Institute, School of Physics and Technology, University of Jinan, 250022 Jinan, China. [8] Key Laboratory of Functional Materials Physics and Chemistry of the Ministry of Education, Jilin Normal University, 130103 Changchun, China. [9] These authors contributed equally: Liya Yang, Houbing Huang. ✉email: zhenglm@sdu.edu.cn; sdy_lvwm@ujn.edu.cn; liugang@hpstar.ac.cn

Various ferroelectric device types exist. However, the same basic mechanism occurs in all devices: spontaneous polarization ($P_S$) changes under external stimuli and then converts mechanical to electrical energy, or vice versa; here, the polarization rotation, extension, and switch are critical[1–3]. Although ferroelectric materials differ, the core task for applications is to always make the above sequence of events possible, favorable, and stable, under both small and large drives[4,5]. Consequently, high piezoelectric response and a large coercive field ($E_C$) are of fundamental importance, enabling both high operation efficiency and a wide operational field range in numerous electromechanical applications, such as high-power transducers and high-field actuators[6,7]. In the past 30 years, ultrahigh piezoelectric perovskites, relaxor-PbTiO₃ (relaxor-PT) single crystals, have been discovered and greatly developed; they are the driving force for emerging electromechanical applications[8]. However, compositional modification for the simultaneous enhancement of piezoelectricity and $E_C$ is challenging. For example, inferior piezoelectricity $d_{33}$ (~1100 pC/N) is afforded and consequently considerable degradation of electromechanical response occurs upon the hard doping by manganese (Mn) in Pb(Mg$_{1/3}$Nb$_{2/3}$)O₃–PbTiO₃ (PMN–PT) crystals[9,10]; further, $E_C$ with a low magnitude (~2.4 kV/cm), which is unsuitable for high-power and high-field applications, is afforded when soft doping strategies are employed[11]. Figure 1a summarizes the relation between the coercive field $E_C$ and piezoelectric coefficient $d_{33}$ for various relaxor-PT ferroelectric crystals, demonstrating that high piezoelectric activity is generally associated with a low coercive field. Thus, while ultrahigh-piezoelectric relaxor-PT crystals are revolutionizing the electromechanical community, the crucial question to naturally arise is "is there a possibility to highly enhance the coercive field without sacrificing their ultrahigh piezoelectricity?"

Over the past decade, remarkable progress has been made toward achieving ultrahigh piezoelectricity in relaxor-PT by introducing an additional structural heterogeneity and a slush-like polar state to manipulate interfacial energies and/or expand the phase coexistence region[12–14], which can further flatten the energy landscape and consequently favor an easy polarization rotation (Fig. 1b). However, a remarkable high $E_C$ has still not been achieved, which requires a high potential barrier to make the dipole switch difficult (Fig. 1b)[15]. For example, although recent studies have achieved piezoelectric activity of over 4000 pC/N in Sm-doped PMN–PT single crystals[11], the crystals afford a low $E_C$ (~2.4 kV/cm). Furthermore, some studies demonstrated that $E_C$ of over 10 kV/cm can be achieved by doping relaxor-PT with Yb and Ho, but a weak piezoelectric response (~1100 pC/N) is inevitably afforded[10,16]. To date, to our knowledge, no study has

reported a method for simultaneously achieving ultrahigh piezoelectricity and large $E_C$ for relaxor-PT crystals, and the fundamental mechanism of this issue is not yet fully understood.

Herein, the thermodynamics and microstructure of the relaxor-PT system are re-scrutinized. As shown in Fig. 1b, the piezoelectric activity and $E_C$ could be simultaneously improved by making the potential wells flatter and enhancing the barrier between adjunct polar states; the former could be realized via nanoscale inhomogeneity and the latter is usually correlated to large tetragonality in the perovskite lattice[8]. Thus, if a high-piezoelectric parent matrix comprises strongly tetragonal polar nano-regions (PNRs), large $E_C$ enhancement with improved piezoelectricity could be achieved. One can see an obvious difference exists in the lattice constants between PMN and another relaxor Pb(Sc$_{1/2}$Nb$_{1/2}$)O₃ (PSN), (Supplementary Note 1 and Table S1), which is critical for creating a highly anisotropic microstructure with large tetragonality in a relaxor-PT system. Thus, we studied the effect of scandium (Sc) substitution for B-site cations in a model ultrahigh-piezoelectric relaxor-PT perovskite, PMN–PT. The resulting ternary 0.06PSN–0.61PMN–0.33PT crystals demonstrate excellent piezoelectric activity and electromechanical coupling response ($d_{33} = 2630$ pC/N, $k_{33} = 90.8\%$; $d_{15} = 490$ pC/N, $k_{15} = 54.7\%$), where the shear activity is twice that of the binary PMN–PT counterpart, and the ultrahigh longitudinal performance is maintained. Notably, $E_C$ was successfully improved by over three times to 8.2 kV/cm. Such an enhancement of the coercive field is the largest magnitude among all giant piezoelectric materials, far exceeding all experimentally observed results. Echoing to the proposed free energy landscape design, this work provid not only effective experimental routes but also vital theoretical guidelines for designing better ferroelectric materials.

## Results

**Materials properties.** The composition selection of the PSN–PMN–PT solid solution is based on two considerations. First, the morphotropic phase boundary (MPB) compositions are definitely required to optimize the piezoelectric properties[8,17]. From the composition-dependent phase diagram of binary PMN–PT and PSN–PT, we deduced the MPB regions of the ternary PSN–PMN–PT system, as shown in the blue region in Fig. 2a. Additionally, a low Sc content doping strategy was employed in the crystal design because the piezoelectric activity of PSN–PT is significantly weaker than that of PMN–PT[18,19]. Subsequently, high-quality crack-free 0.06PSN–(0.94-x)PMN–xPT ($x = 0.31$-0.35) crystals with a diameter of 25 mm were successfully grown. Figure 2b displays the photograph of the as-grown crystals. All samples were cut from the same thin crystal wafer with identified PT contents and then poled along [001]$_C$ for domain-engineered configurations. The quantitative compositions of the samples were demonstrated to be

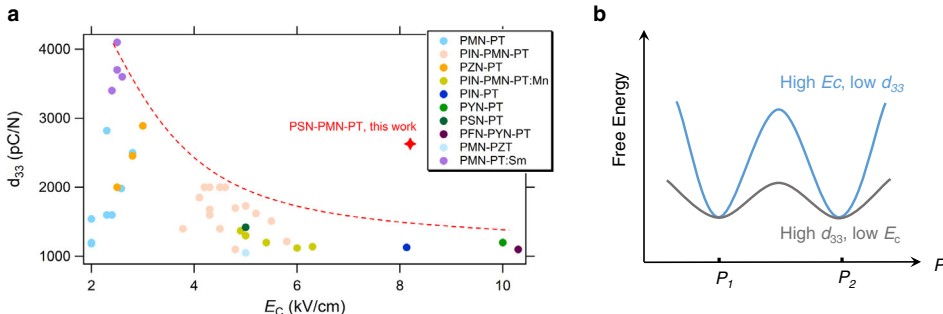

**Fig. 1 Dilemma in ferroelectrics: enhanced $E_C$ is usually achieved at the expense of piezoelectricity. a** $d_{33}$ vs. $E_C$ for various relaxor-PT single crystals. The red dashed line denotes the tendency of most relaxor-PT crystals. Generally, $E_C$ enhancement is accompanied by inferior piezoelectricity. Alternatively, our 0.06PSN–0.61PMN–0.33PT (red star) affords highly remarkable results. Data from refs. [9, 11, 17, 22, 31, 50–65] and this work. **b** Schematic of the different free energy landscapes and the corresponding macroscopic performances.

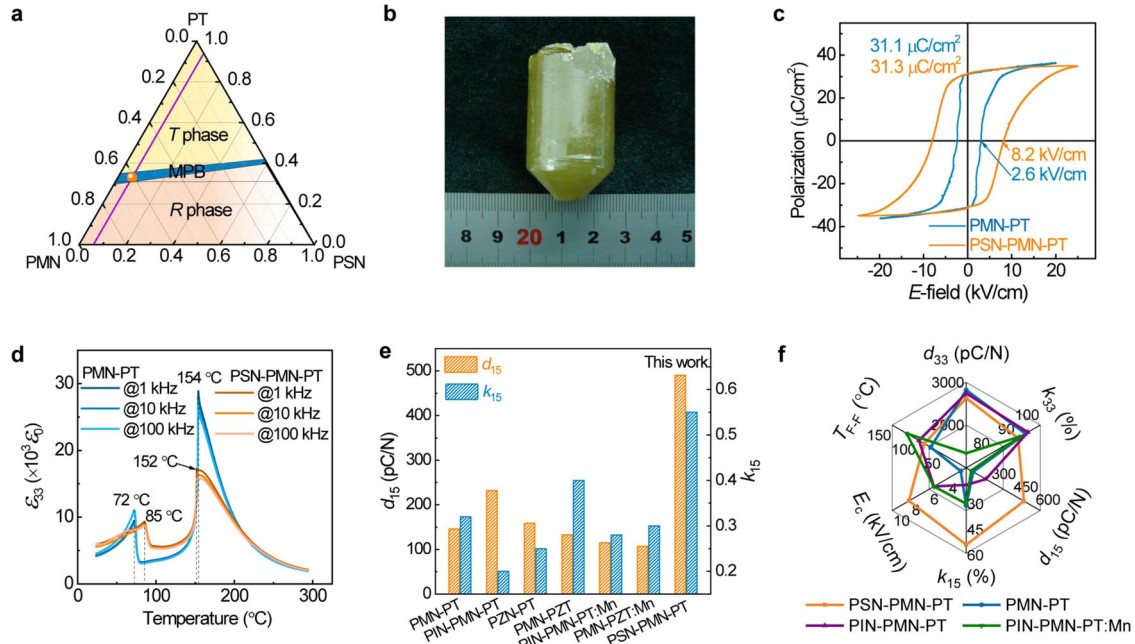

**Fig. 2 Functional characterizations of PSN–PMN–PT single crystals. a** Phase diagram of the PSN–PMN–PT ternary system. The blue area indicates the MPB region of the PSN–PMN–PT system. The orange point indicates the composition of our sample, 0.06PSN-0.61PMN-0.33PT, which is intensively investigated herein. **b** Photograph of the as-grown PSN–PMN–PT crystals, showing a large dimension of $\Phi25 \times 35$ mm³. **c** $P$–$E$ loop of 0.06PSN-0.61PMN-0.33PT in comparison with that of 0.67PMN–0.33PT. **d** Temperature dependence of the relativity dielectric permittivity of the two crystals. 0.06PSN-0.61PMN-0.33PT exhibits a comparable Curie temperature $T_C$ but a markedly improved $T_{F-F}$ than 0.67PMN–0.33PT. **e** Shear piezoelectric performance of various [001]$_C$-poled relaxor-PT crystals with MPB composition, demonstrating that 0.06PSN-0.61PMN-0.33PT exhibits significantly superior $d_{15}$ and $k_{15}$. **f** Comparison of the essential parameters for various [001]$_C$-oriented PbTiO₃-based relaxor ferroelectric single crystals with MPB composition. The 0.06PSN-0.61PMN-0.33PT crystals cover the largest area, denoting a superior overall performance. Data from refs. [22, 66, 67], and this work.

0.06PSN–0.61PMN–0.33PT via energy dispersive spectrometry, satisfying the material design requirements.

Compared to their PMN–PT binary counterpart, 0.06PSN–0.61PMN–0.33PT crystals exhibit high longitudinal piezoelectric activity as well as obviously higher shear piezoelectric activity, higher ferroelectric–ferroelectric phase transition temperature $T_{F-F}$, and a larger coercive field $E_C$. As shown in Fig. 2c, the polarization–electric hysteresis loop ($P$–$E$ loop) indicates that the 0.67PMN–0.33PT crystals afford a low $E_C$ of 2.6 kV/cm, severely hampering its potential applications. Notably, after a little Sc substitution, the coercive field of the resulting 0.06PSN–0.61PMN–0.33PT crystals is improved to 8.2 kV/cm, which is three times larger than that of PMN–PT. To the best of our knowledge, such an enhancement represents the most advanced enhancement reported to date for almost all investigated ferroelectrics with ultrahigh piezoelectric coefficients of over 2000 pC/N. To verify the repeatability of the extraordinary $E_C$ values, we analyzed several different samples; all samples afforded $E_C$ values of around 8 kV/cm (Fig. S1). Note that $E_C$ is not an intrinsic property of ferroelectrics[20,21]. Poling/de-poling is related to voltage-induced domain switching, including nucleation of new domains at a defect site (normally near domain walls) and domain growth. Thus, we considered the dynamics of domain switching by measuring the $E_C$ at various frequencies. As is well known, domain switching is considerably easy at low frequencies, which is related to domain switching under a very low field and long holding time of the applied fields. For the 0.06PSN–0.61PMN–0.33PT crystals, a large $E_C$ of ~7.5 kV/cm is maintained at frequencies as low as 0.1 Hz, and 11.8 kV/cm is afforded at 100 Hz (Supplementary Note 2 and Fig. S2). Moreover, we observed a

low conductive current during the entire poling process, confirming the high quality of the crystals (Supplementary Note 3 and Fig. S3).

The $T_{F-F}$ observed in 0.06PSN–0.61PMN–0.33PT crystals is particularly interesting, ~85 °C, which are 13 °C higher than that observed for the PMN–PT crystals with similar PT contents (Fig. 2d), promising for a wide temperature usage range and drive field stability. Figure S4 displays the temperature dependence of the piezoelectric response of the 0.06PSN–0.61PMN–0.33PT crystals, exhibiting a variation of 140% in $d_{33}$, which is considerably lower than that of PMN–PT crystals (200–300%)[8]. Furthermore, a relatively high $E_C$ of over 6.2 kV/cm can be maintained till the occurrence of the phase transition (Supplementary Note 4 and Fig. S5).

Remarkably, the piezoelectric coefficients $d_{15}$ and electromechanical coupling constants $k_{15}$ of the 0.06PSN–0.61PMN–0.33PT crystals were found to be 490 pC/N and 54.7%, respectively (Fig. 2e), featuring a significantly larger shear piezoelectric response than those of other [001]$_C$-poled relaxor-PT systems. Moreover, 0.06PSN–0.61PMN–0.33PT crystals exhibit almost the same ultrahigh longitudinal property ($d_{33}$ = 2630 pC/N) as the 0.67PMN–0.33PT crystals[22]. We also conducted detailed piezoelectric force microscopy (PFM) characterizations to investigate the polarization switching behavior and the local piezoelectric deformations of the PSN–PMN–PT crystals; the results further support their superior piezoelectric response from a microscopic perspective (Supplementary Note 5 and Fig. S6). Below $T_{F-F}$, no obvious changes were observed in the domain morphology with increasing temperature (Fig. S7), which well agrees with the weak variations of the piezoelectric response and electromechanical properties, namely, a relatively strong thermal stability of functionality (Fig. S4). Figure 2f presents a radar chart summarizing

the critical properties, including $d_{33}$, $k_{33}$, $d_{15}$, $k_{15}$, $E_C$, and $T_{F-F}$ of various $[001]_C$-poled relaxor-PT crystals with MPB compositions. The figure shows that the 0.06PSN–0.61PMN–0.33PT crystals cover an extremely large area, thereby demonstrating their superior overall performance and greater efficiency for potential device applications[20–22].

**Relaxor behavior**. A key feature of relaxor-ferroelectric solid solutions is the existence of local heterogeneity, such as PNRs, which contribute over 50% to the dielectric/piezoelectric response according to the recent cryogenic experimental measurements[13,23]. Therefore, we investigated the relaxor behavior of the ternary 0.06PSN–0.61PMN–0.33PT crystals and explored the possible differences from their binary 0.67PMN–0.33PT counterparts, with the aim to determine why 0.06PSN–0.61PMN–0.33PT crystals simultaneously exhibit ultrahigh piezoelectric activity and extremely large $E_C$, which seems unusual in most ferroelectric solid solution systems, as summarized in Fig. 1a.

Figure 3a shows the temperature-dependent reciprocal of the dielectric response of the 0.06PSN–0.61PMN–0.33PT and 0.67PMN–0.33PT crystals, indicating that the phase transitions proceed gradually rather than sharply with temperature. Such a diffuseness characteristic is a relaxor feature, causing the deviation from the Curie–Weiss law, where the Burns temperatures ($T_B$) of around 268 °C can be derived for 0.06PSN–0.61PMN–0.33PT, 40 °C higher than that of 0.67PMN–0.33PT (228 °C). Thus, we reasonably deduce that during paraelectric-to-ferroelectric transitions, PNRs appear earlier (at a higher temperature) in 0.06PSN–0.61PMN–0.33PT than in 0.67PMN–0.33PT, presenting polarized precursor clusters. Such a diffused characteristic is further supported by quantitative analysis via the modified Curie–Weiss law[24]:

$$\frac{1}{\varepsilon} - \frac{1}{\varepsilon_m} = \frac{(T - T_m)^\gamma}{C} \qquad (1)$$

where $\varepsilon_m$ is the maximum dielectric constant at $T_m$, $C$ is the Curie-like constant, and $\gamma$ describes the degree of diffuseness. Linear fitting of $\ln(1/\varepsilon_{33} - 1/\varepsilon_m)$ versus $\ln(T - T_m)$ data yields $\gamma$ values of 1.96 and 1.74 for 0.06PSN–0.61PMN–0.33PT and 0.67PMN–0.33PT, respectively (Fig. 3b), suggesting a stronger relaxor nature in the ternary system. Further autocorrelation function analysis based on PFM characterizations demonstrates that it is much more difficult for the ternary crystal to establish a homogeneous polarization long-range order between neighboring clusters (Supplementary Note 6 and Figs. S8 and S9).

Figure 3c shows the frequency dependence of high-temperature dielectric properties of the ternary 0.06PSN–0.61PMN–0.33PT crystals, where the magnitude of the dielectric permittivity decreases and the dielectric maximum shifts to higher temperatures with increasing frequency, again demonstrating strong relaxor behavior. Notably, 0.06PSN–0.61PMN–0.33PT exhibits a "$T_m$ shift" (shift of the dielectric maxima temperature with frequency over the range of 1 kHz–1 MHz) that is twice that of the 0.67PMN–0.33PT crystals (5 vs. 2.5 K, Fig. 3c and Fig. S10 in Supplementary Note 7), signifying strong interactions between PNRs and the development of local correlations[23]. Furthermore, the frequency dependence of the high-temperature dielectric data can be well fitted using the Vogel–Fulcher relation[25]:

$$f = f_0 \exp[-E_a/(k_B(T_m - T_{VF}))] \qquad (2)$$

where $f_0$ is the Debye frequency, $T_m$ is the temperature of the permittivity maximum, and $T_{VF}$ is the static freezing temperature, which can be deemed as $T_m$ at 0 Hz. $T_{VF}/T_m$ is a semi-quantitative parameter employed for evaluating PNR interactions[25]. $E_a$ represents the activation energy of the polarization fluctuation in an isolated cluster that stems from the development of a short-range order; thus, a larger $E_a$ suggests stronger interactions between neighboring PNRs. The fitted results are given in Fig. 3d, e, which show that the $E_a$ of the 0.06PSN–0.61PMN–0.33PT crystals is considerably higher than that of 0.67PMN–0.33PT crystals (~0.024

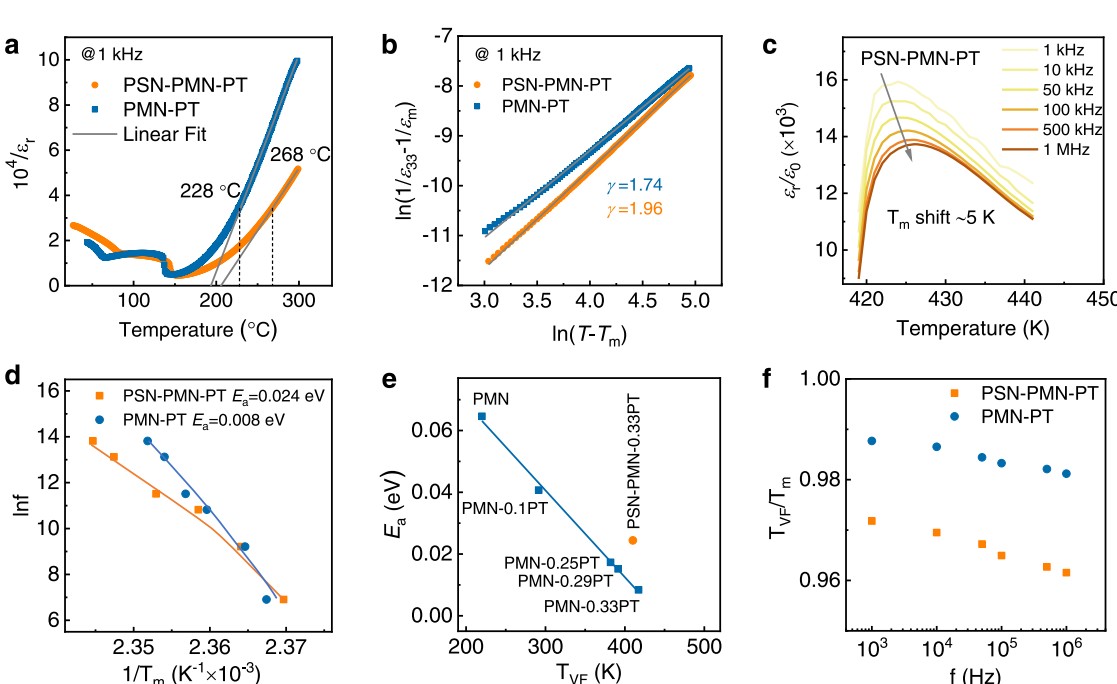

**Fig. 3 Relaxor behavior. a** Temperature dependence of the reciprocal of dielectric permittivity, from which $T_B$ is obtained by fitting with the modified Curie–Weiss law. **b** Modified Curie–Weiss law fitting results, from which $\gamma$ can be obtained. **c** High-temperature dielectric property of 0.06PSN–0.61PMN–0.33PT crystals measured at various frequencies from 1 kHz to 1 MHz. **d** Vogel–Fulcher fitting results on the data shown in Fig. 3c and Fig. S7, from which the activation energy $E_a$ and static freezing temperature $T_{VF}$ can be determined. **e** Summary of $E_a$ for various pure PMN, PMN-PT, and PSN–PMN–PT. Data are from refs. [25, 68, 69]. and this work. **f** $T_{VF}/T_m$ value as a function of frequency for 0.67PMN–0.33PT and 0.06PSN–0.61PMN–0.33PT.

vs. 0.008 eV), signifying stronger interactions among PNRs in the ternary crystals. Considering the higher $T_B$, larger $\gamma$, shorter range order, higher $E_a$, and lower $T_{VF}/T_m$ (Fig. 3f), we reasonably deduce that 0.06PSN–0.61PMN–0.33PT exhibits more relaxor and diffused characteristics with much significantly stronger interactions between adjacent PNRs than 0.67PMN–0.33PT.

**Highly dispersed local heterogeneous structure with considerable tetragonal phase.** The strong polar cluster interaction is directly related to the structural instability and finally contributes to the material functionality[26,27]. This motivated us to further resolve the local microstructure of the 0.06PSN–0.61PMN–0.33PT crystals, and investigate its possible phase coexistence and complex crystallographic symmetry, which are crucial for understanding why PSN–PMN–PT crystals simultaneously afford ultrahigh possesses giant piezoelectricity and a high coercive field.

Figure 4a shows an aberration-corrected high-angle annular dark-field scanning transmission electric microscopy (HAADF-STEM) image. From the image we determined the polarization vector $P_S$ of each unit cell column based on the atomic positions. Note that a dispersed polar state with multiphase, including rhombohedral (R) and/or orthorhombic (O), tetragonal (T), and monoclinic (M), was detected. Moreover, 0.06PSN–0.61PMN–0.33PT exhibited considerably smaller (2–4 nm) than 0.67PMN–0.33PT (8–20 nm), suggesting a higher density of domain walls/phase interfaces and abundant local heterogeneous structure (Supplementary Note 8 and Figs. S11 and S12).

The abundant tetragonal phase in 0.06PSN–0.61PMN–0.33PT determined via HAADF-STEM is notable. This behavior was further evidenced in our high-resolution X-ray diffraction (XRD) characterizations, from which a detailed analysis of the peak positions and intensities was conducted and the volume fraction of the tetragonal component was estimated to be 34.5% (Fig. 4b), which is significantly higher than that of 0.67PMN–0.33PT, 13.7% (Supplementary Note 9, Table S2 and Fig. S13). Electric-field- and temperature-dependent structural evolutions were also conducted, and the relationship of structure–dielectric/piezo-electric properties were studied (Supplementary Note 10, Table S3 and Figs. S14–S17).

Furthermore, we calculated the distances between A-site cations on a per-unit cell basis via HAADF-STEM and estimated the local lattice anisotropy by determining the local $c/a$ ratio. As shown in Fig. 4c, the standard deviation of the lattice parameter is significantly larger for 0.06PSN–0.61PMN–0.33PT than that for 0.67PMN–0.33PT, demonstrating a higher fluctuation in the sublattice parameter. Additionally, the local $c/a$ ratios for PSN–PMN–PT varied more than those for PMN–PT, indicating a much larger tetragonality and a more dispersive behavior. These observations are consistent with the XRD data (Table S2).

Then, we conducted geometric phase analysis (GPA) on the HAADF-STEM images of 0.06PSN–0.61PMN–0.33PT and 0.67PMN–0.33PT crystals, from which we derived the variations of local strain $S_{xx}$ along $[001]_C$ were (Fig. 4d, e). Notably, the ternary crystals possess significantly higher local strain (~3%) than their binary counterparts (1.5%), suggesting a large tetragonal lattice deformation $c/a$ ratio.

This microstructure of the PSN–PMN–PT crystals has not been previously reported for any other ultrahigh-piezoelectric materials. It features a highly dispersed local heterogeneous structure with abundant tetragonal phases, markedly differing from the usual behavior of binary relaxor-PT crystals, where the high piezoelectric activity is only found in the rhombohedral-side MPB compositions. The phase-field calculations well match our experimental discoveries, verifying the experimentally observed microstructure and functionality. As shown in Fig. 4f, pure PMN–PT exhibits rhombohedral characteristics with a large domain size. When some tetragonal nanosized phases are introduced into this matrix (similar to the PSN–PMN–PT case), a dispersed domain structure with decreased domain size forms, and multiphase coexistence becomes inevitable, well agreeing with the transmission electron microscopy (TEM) results shown in Fig. 4a. We calculated the magnitudes of $E_C$ and $d_{33}$ of these two systems, from which we observed significant $E_C$ enhancement without piezoelectricity reduction due to appropriate doping of the tetragonal phase (Supplementary Note 11, Table S4 and Fig. S18).

**Discussion**

Based on the experimental and phase-field simulation results, the ultrahigh piezoelectricity and extremely large $E_C$ in the PSN–PMN–PT system can be explained in the mesoscale. Previous studies have demonstrated that the introduction of cations into the B-site of PMN–PT can afford a high level of charge inhomogeneity[28–30], consequently yielding strong relaxor behavior, as confirmed in this study (Fig. 3 and Supplementary Note 12). Compared to PMN–PT, it is considerably more difficult for PSN–PMN–PT to establish a ferroelectric long-range order with only short-range ordering between neighboring clusters, accounting for symmetry breaking; thus, we observed a highly dispersed local heterogeneous structure (Fig. 4). The dispersed micropolar state with multiphase coexistence is strongly correlated with abundant local heterogeneity (Fig. 4a), which can significantly flatten the free energy profile, significantly contributing to the ultrahigh piezoelectric activity[12]. In addition to the highly dispersed characteristic, 0.06PSN–0.61PMN–0.33PT crystal also features a considerable tetragonal phase component with a relatively large $c/a$ ratio (Fig. 4b–d). Previous studies on the structure–property relation of relaxor-PT showed that tetragonal-rich crystals usually exhibit large $E_C$[31], where the large $c/a$ ratio is directly related to the high potential barrier between different polar states. After Sc doping into PMN–PT, nanosized tetragonal domains are highly dispersed into the matrix (Fig. 4a). These tetragonal polar regions strongly interact with their neighboring clusters (Fig. 3e, f), acting as "frozen seeds" in the entire matrix and pining the $P_S$ switch via a cooperative effect, consequently yielding unparalleled coercive field enhancement. The pinning effect of the tetragonal polar regions on domain switch is associated with the difficult domain nucleation and growth process[32,33], which is supported by the large activation electric field in 0.06PSN–0.61PMN–0.33PT (Supplementary Note 13 and Figs. S19 and S20).

Although an ultrahigh piezoelectric response and a large $E_C$ are generally exclusive in a single ferroelectric material (Fig. 1a), the enhancement of $E_C$ in our PSN–PMN–PT crystal is notably not at the expense of the piezoelectric activity, which can be explained from its particular microstructure. The dispersed microstructure is strongly correlated to a slush-like polar state with coexisting multiphases, including O/R, T, and M, signifying the instability of the ferroelectric phases, and intrinsically contributing substantially to the ultrahigh piezoelectric activity. Moreover, the interfacial energies can be manipulated using the abundant local structure heterogeneity arising from the tortuous interfaces between adjacent clusters, and the small domain size[12], further improving the piezoelectric response. Note that due to the existence of the tetragonal phase, fully (001) poled crystals may contain considerable single-domain components[34] and possess large shear piezoelectric activity that stems from the easy polarization rotation (Supplementary Note 14 and Figs. S21 and S22), which partially explains the two times larger $d_{15}$ in 0.06PSN–0.61PMN–0.33PT than that in 0.67PMN–0.33PT.

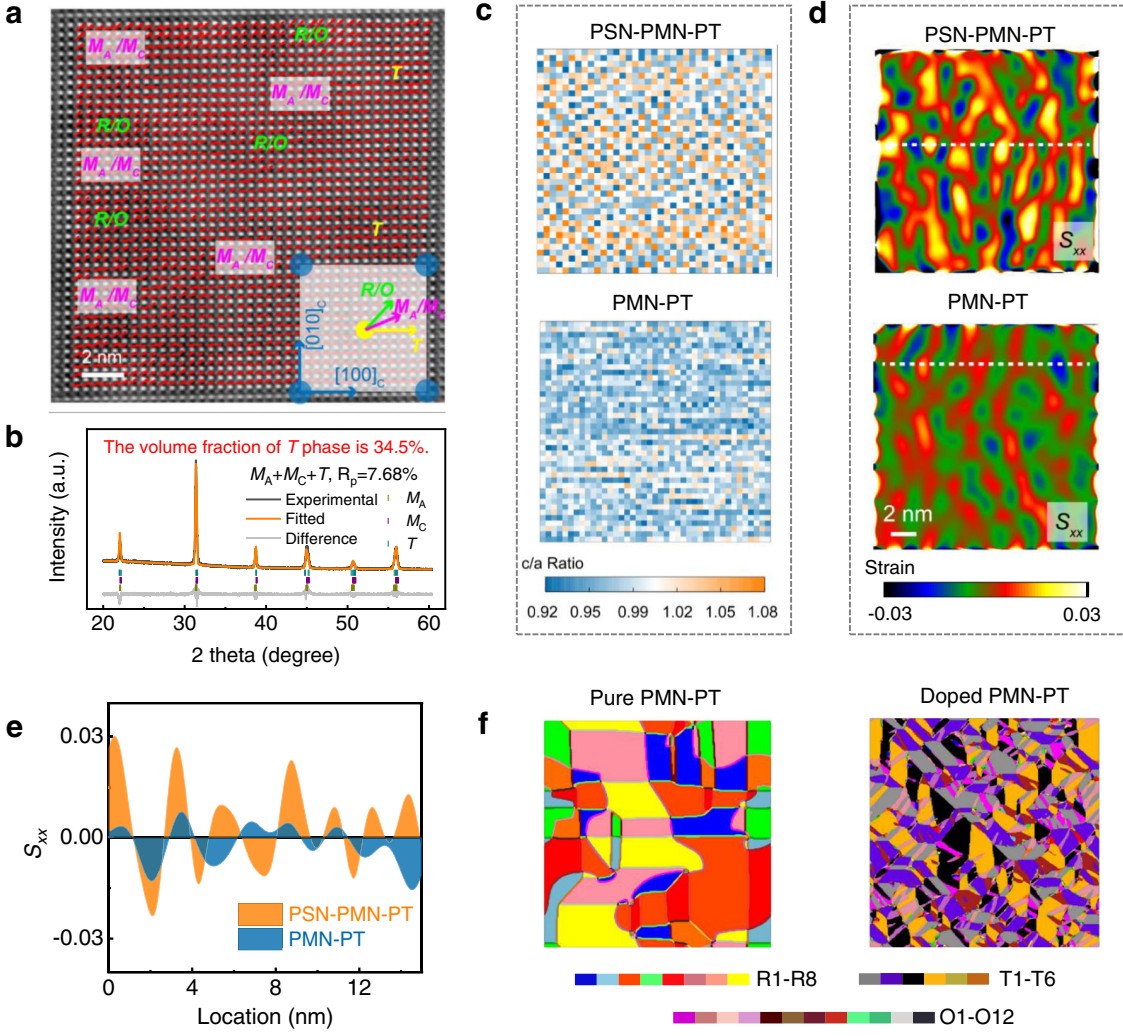

**Fig. 4 Microstructure. a** Atomic-resolution TEM images of the [001]$_C$-oriented 0.06PSN–0.61PMN–0.33PT crystals, where the $P_S$ directions are given for each unit-cell column. The possible phase structures can be deduced using the $P_S$ directions (R rhombohedra, O orthorhombic, $M_A$/$M_C$ monoclinic, and T tetragonal). **b** High-resolution XRD pattern and the optimal refinement results for 0.06PSN–0.61PMN–0.33PT. **c** Unit cell $c/a$ ratios for 0.06PSN–0.61PMN–0.33PT and 0.67PMN–0.33PT derived from the TEM characterizations. **d** Local strain $S_{xx}$ mapping extracted from the HAADF-STEM lattice image of 0.06PSN–0.61PMN–0.33PT in (**a**) and 0.67PMN–0.33PT in Fig. S9 via GPA, and the data along the white dotted lines are extracted and shown in (**e**). **f** Phase-field simulations of the domain structures of pure PMN-PT and that doped with tetragonal phase. PMN–PT exhibits pure R characteristics; after the introduction of the tetragonal phase, it exhibits multi-phase including R, O, and T characteristic with a reduced domain size. Different phases and various $P_S$ directions in the same phase are denoted by different colors.

Based on the above discussion, we propose a thermodynamic understanding of the inherent correlation between the macrostructure and materials properties: as regulated by Landau theory, the dispersed heterogeneous structure with multiphase coexistence makes the free energy extremely flat; meanwhile the highly tetragonal polar regions pin the domain switch, indicating the enhancement the potential barriers, resulting in a "flat and deep" potential well Fig. 5. This peculiar free energy profile causes a difficult polarization switch and an easy polarization rotation; consequently, an ultrahigh piezoelectric response and extremely large $E_C$ are simultaneously achieved. Both hard (large $E_C$) and soft (high piezoelectric response) doping properties are affording using this strategy, successfully addressing the longstanding challenge that excellent sensitivity and high stability of dipoles are generally exclusive, and achieving a striking enhancement of the overall performance.

To verify the importance of the delicate optimization of the composition, our study is compared with previous studies based on similar material systems. In PSN–PMN–PT crystals with

PSN:PMN ratio of 1:3, 1:1, or 3:1, Wang et al. [35] determined the piezoelectric constant as 1200–1600 pC/N and the coercive field as 4–6 kV/cm, which are significantly inferior to those of our 0.06PMN–0.61PMN–0.33PT sample (PSN:PMN ratio ~1:10). It has been demonstrated that the introduction of PSN into PMN–PT can yield a high level of lattice anisotropy (Supplementary Note 1)[36,37], favoring a tetragonal phase. Thus, if a superfluous amount of Sc is introduced, the tetragonal clusters may become too large for dispersal into the entire ferroelectric matrix, destroying the desirable local heterogeneous microstructure; additionally, a phase separation could occur, causing functional degradation. Moreover, the importance of MPB composition in the piezoelectric response should be emphasized. Xi et al. [7,38] reported that in 0.06PSN–0.63PMN–0.31PT single crystals, although with a large coercive field of 8.17 kV/cm, the maximum piezoelectric constant is only ~1200 pC/N, which is similar to that of PZT ceramics or lead-free crystals. Guo et al. [39] reported that 0.05PSN–0.63PMN–0.32PT single crystals in R phase exhibit inferior performance with $d_{33}$ = 1200 pC/N and

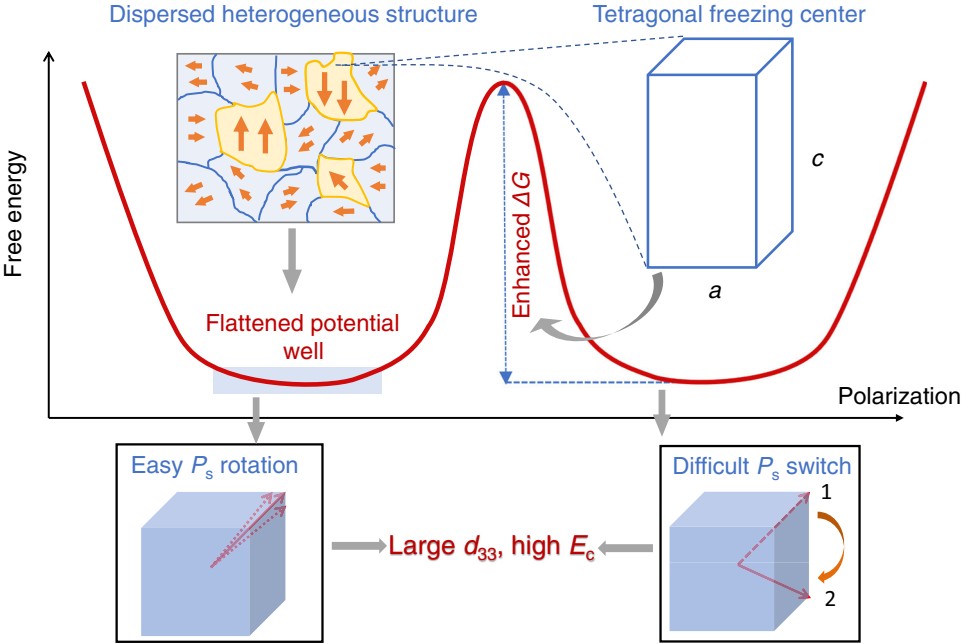

**Fig. 5 Simultaneous achievement of ultrahigh piezoelectricity and extremely large $E_C$.** The particular local structure and relatively large $c/a$ ratio in the PSN–PMN–PT are used to manipulate the free energy profile in different ways, affording a "deep and flat" free energy landscape. The PSN–PMN–PT system exhibits various characteristics including high level of local structure heterogeneity, slush-like multiphase coexistence, small domain size, and high density of PNRs, accounting for a flat potential well. Moreover, the lattice of PSN–PMN–PT system contains a considerable tetragonal phase component, demonstrating large anisotropy, which consequently contributes to a deep well, namely an enhanced $\Delta G$. The flattened potential well facilitates the polarization ($P_S$) rotation/elongation around the equilibrium position, enhancing the piezoelectric response, and the enhanced energy barrier makes the $P_S$ switch more difficult, contributing to a large $E_C$.

$\varepsilon_{33} \sim 3500$. Compared to previous studies, our 0.06PSN–0.61PMN–0.33PT single crystals exhibit substantially superior overall performance (Supplementary Note 15, Table S5 and Fig. S23). Therefore, the appropriate balance between various effects, including relaxor and long-range order and tetragonal and other phases, need to be considered for materials-by-design. This strategy should be converted to an atomistic model to understand the contribution of each atom to the free energy profile in a complex material system, to ultimately realize high-performance and/or high-power applications.

The natural question arises that "why such simultaneous improvement has not been obtained in previous studies?" Currently, the most conventional strategy employed to enlarge the coercive field in relaxor-PT systems is hard doping[8,9]. In hard doping, a small amount (<2 mol%) of acceptor ions such as $Mn^{2+/3+}$, is substituted into the B-sites of perovskite lattices, yielding acceptor–oxygen vacancy defect dipoles. These defect dipoles occupy energetically preferred sites in the lattice and align themselves along a preferential direction within a ferroelectric domain, and then, they move to the highly stressed areas of domain walls[40]. These defect dipoles pin the domain walls and stabilize the domain, establishing a parallel arrangement of defect dipole and local ferroelectric polarization, causing an offset of $P$–$E$ behavior that is experimentally characterized as internal bias, which effectively increases the $E_C$ by 30% compared to that of undoped materials[41]. Such a significantly reduced degree of switchable polarization is accompanied by suppressed domain wall mobility, inevitably resulting in an inferior piezoelectric response. Alternatively, based on the $P$–$E$ characterization results (Fig. 2c), an internal bias was not observed in the PSN–PMN–PT crystals, signifying the presence of a distinct mechanism associated with the intrinsically high lattice strain rather than the domain clamping effect observed for Mn-doped crystals. Therefore, the piezoelectric coefficients did not decrease with Sc

doping, due to the no-loss or even enhanced extrinsic piezoelectric contribution. These easily removable domain walls are further suggested by Rayleigh analysis (Supplementary Note 16, and Figs. S24 and S25), showing that a large Rayleigh parameter $\alpha$ is afforded at both the room temperature and the temperature near $T_{F–F}$, which is not favored in hard doping[8,41].

In conclusion, by employing a precise microstructure-by-design, we successfully addressed the long-sought-after materials with simultaneous ultrahigh piezoelectricity and unparalleled enhancement of $E_C$. Within the theoretical framework, we proposed a thermodynamic understanding of the inherent correlation between the free-energy landscape and material properties, where a "flat and deep" potential well is derived. Our dataset is the confirmation of the existence of extremely large $E_C$ in an ultrahigh piezoelectric material. Furthermore, although the relaxor–PT solid solution is employed herein, our proposed strategy is likely a universal and effective method for designing high-performance functional materials with both high tolerance and sensitivity to the external field.

## Methods

**Crystal growth.** The PSN–PMN–PT single crystals were grown using the Bridgman technique. High-purity $Sc_2O_3$ (99.99%), $Nb_2O_5$ (99.95%), $(MgCO_3)_4 \cdot Mg(OH)_2 \cdot 5H_2O$ (>99.0%), PbO (>99.0%), and $TiO_2$ (>99.0%) were used as raw materials. The precursors $MgNb_2O_6$ and $ScNbO_4$ were synthesized in advance to avoid the impurity phase formation. Then, $MgNb_2O_6$, $ScNbO_4$, $TiO_2$ and PbO powders were mixed and placed in a Pt crucible with a sealed $Al_2O_3$ crucible. The crucible was placed in a computer-controlled Bridgman furnace, which was heated from 600 to 1400 °C at a rate of 10 °C/min and maintained at 1400 °C for 10 h, and a stable temperature gradient of 30–50 °C/cm was formed. The crucible was descended at a rate of 0.2–0.4 mm/h, and the PSN–PMN–PT single crystals gradually grew via spontaneous nucleation.

**Sample preparation and electrical property measurements.** All the samples used herein were $[001]_C$-oriented with $x//[100]_C$, $y//[010]_C$ and $z//[100]_C$ via XRD. After cutting and polishing, all the samples were annealed at 600 °C for 1 h to

eliminate the stress generated during sample preparation. Gold electrodes were sputtered on the parallel $(001)_C$ faces of crystals. The temperature dependence of the dielectric constants was measured using an LCR meter (Agilent, 4284A) with a 2 °C/min step. $P$–$E$ loops were obtained using a Precision Premier II tester (Radiant Technologies, Albuquerque). After being poled by a DC $E$-field of 10 kV/cm at room temperature, the longitudinal piezoelectric coefficient $d_{33}$ was recorded using a quasi-static $d_{33}$ meter (Institute of Acoustics, ZJ-4A) and shear coefficient $d_{15}$ was measured using the resonance method. The resonance and anti-resonance frequencies were obtained using an Agilent 4294A impedance-phase gain analyzer, based on which the electromechanical coupling factors $k_{33}$ and $k_{15}$ were obtained.

**PFM measurements and the autocorrelation function technique**. For the PFM measurements employed in this work, the samples were ground to ensure a flat surface using the $Al_2O_3$ grinding powder and subsequently polished using polycrystalline diamond suspensions with abrasive particles of 9, 3, 1 μm, and 20 nm (MetaDi Supreme, Buehler). The PFM studies were performed using a Cypher ES (Asylum Research) in DART mode using Ir/Pt-coated conductive tips (Nanoworld, EFM). The autocorrelation images were obtained based on the PFM domain images via the following transformation[42,43]:

$$C(r_1, r_2) = \sum_{x,y} D(x,y)D(x+r_1, y+r_2) \tag{3}$$

where $D(x,y)$ is the piezoelectric signal and the autocorrelation function $C(r_1,r_2)$ is the two-dimensional polarization–polarization correlation function. Furthermore, $\langle C(r) \rangle = \sigma^2 \exp[-(r/\xi)^{2h}]$ is the averaged autocorrelation function $C(r_1, r_2)$ over all in-plane directions.

**Scanning transmission electron microscopy (STEM) experiments**. The TEM samples were prepared using a Tescan LYRA-3 XUM Model focused ion beam instrument. The selected area electron diffraction patterns and morphology of the crystals in Fig. S8 were characterized using TEM FEI Talos F200. STEM images were acquired on a spherical aberration-corrected FEI Titan G2 microscope operated at 300 kV using a HAADF detector. All STEM images were Fourier-filtered using a lattice mask to remove noise. The strain analyses in HAADF-STEM images were obtained through GPA using the custom scripts in the Gatan Digital-Micrograph software[44]. The polar vector for each unit cell was determined as the B-site cation displacement relative to its four nearest A-site neighbor cations by fitting atom positions as two-dimensional Gaussian peaks[45], which are mapped in the HADDF-STEM images of Figs. 4a and S9.

**Phase-field simulations**. In the phase-field simulations, the polarization $P_i(r,t)$ $(x,y,z)$ denotes the order parameter, which describes the ferroelectric polarization evolution. The temporal evolution of the polarization can be described by the time-dependent Ginzburg–Landau equation:

$$\frac{\partial P_i(\boldsymbol{r}, t)}{\partial t} = -L \frac{\delta F_P}{\delta P_i(\boldsymbol{r}, t)}, \, (i = x,y,z) \tag{4}$$

where $t$ is the simulation time, $L$ is the kinetic coefficient, $\boldsymbol{r}$ is the spatial position, and $F_P$ is the total free energy of the system that is denoted as follows[46]:

$$F_P = \iiint (f_{bulk}(P_i) + f_{elas}(P_i, \varepsilon_{ij}) + f_{elec}(P_i, E_i) + f_{grad}(P_{i,j}))dV \tag{5}$$

where $f_{bulk}$, $f_{elas}$, $f_{elec}$ and $f_{grad}$ represent the Landau bulk, elastic, electrostatic, and gradient energy densities, respectively. A stress-free boundary condition is adopted. The bulk energy density $f_{bulk}$ can be described as a six-order polynomial:

$$\begin{aligned} f_{bulk} &= \alpha_1(P_x^2 + P_y^2 + P_z^2) + \alpha_{11}(P_x^4 + P_y^4 + P_z^4) \\ &+ \alpha_{12}(P_x^2 P_y^2 + P_x^2 P_z^2 + P_z^2 P_y^2) + \alpha_{112}\Big[P_x^4(P_y^2 + P_z^2) \\ &+ P_y^4(P_y^2 + P_x^2) + P_z^4(P_y^2 + P_x^2)\Big] + \alpha_{111}(P_x^6 + P_y^6 + P_z^6) \\ &+ \alpha_{123}P_x^2 + P_y^2 + P_z^2 \end{aligned} \tag{6}$$

where $\alpha_1$, $\alpha_{11}$, $\alpha_{12}$, $\alpha_{111}$, $\alpha_{112}$ and $\alpha_{123}$ are the Landau energy coefficients. Among which, only $\alpha_1$ is temperature-dependent, $\alpha_1 = (T - T_C)/(2\varepsilon_0 C_0)$, where $T$ is the temperature, $T_C$ is the Curie temperature, $C_0$ is the Curie constant, and $\varepsilon_0 = 8.85 \times 10^{-12}$ is the permittivity of vacuum[47]. The Landau coefficients of PMN–0.3PT and PMN–0.42PT were taken from ref. [48].

The gradient energy density can be expressed as

$$f_{grad} = \frac{1}{2} G_{ijkl} \frac{\partial P_i}{\partial r_j} \frac{\partial P_k}{\partial r_l} \tag{7}$$

where $G_{ijkl}$ is the gradient energy coefficient. The electrostatic energy density can be written as

$$f_{elec} = -\frac{1}{2} \varepsilon_0 K_{ij}^b E_i E_j - E_i P_i \tag{8}$$

where $K_{ij}^b$ is the background relative permittivity and $E_i$ is the electric field, which

can be calculated as

$$E_i = -\frac{\partial \varphi}{\partial r_i} \tag{9}$$

The electric potential $\varphi$ can be obtained by solving the electrostatic equilibrium equation

$$\varepsilon_0 K_{ij}^b \frac{\partial^2 \varphi}{\partial r_i \partial r_j} = -\frac{\partial P_i}{\partial r_i} \tag{10}$$

Equations were numerically solved via the semi-implicit Fourier-spectral method[49].

## Data availability

The data that support the findings of this study are available from a public repository at https://doi.org/10.6084/m9.figshare.19346039.v1.

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

## Acknowledgements

L.M.Z. acknowledges the support from the National Natural Science Foundation of China (Grant No. 52072218), the National Key Research and Development Program of China (Grant No. 2021YFB3601504) and the Primary Research & Development Plan of Shandong Province (Grant No. 2019JZZY010313). Z.Z.X. acknowledges the support from the National Natural Science Foundation of China (Grant No. 51772235). W.M.L. acknowledges the support from the National Natural Science Foundation of China (Grant No. 12074149). F.F.G. acknowledges the support from the National Natural Science Foundation of China (Grant No. 11704249). L.Y.Y. appreciates the support from the Natural Science Foundation of Henan Province in China (Grant No. 212300410124). G.L. acknowledges the support from the National Natural Science Foundation of China (Grant No. U2032129) and the National Oversea Youth Talent project.

## Author contributions

The idea and project were conceived by L.Z., W.L., and G.L. L.Z. designed the experiment. L.Y. performed the electrical property and PFM measurements; H.H. and S.X performed the phase-field simulations; Z.X. and F.G. grew the crystals; G.T., Y.Z., and L.K. assisted in the PFM measurements and result analysis; Y.W. and G.L. performed the XRD experiments; M.Z. and H.Z. assisted in the result analysis and provided important suggestions during the preparation of the manuscript; L.Y., L.Z. and G.L. wrote the manuscript.

## Competing interests

The authors declare no competing interests.
