## [Peer review file · Nature Communications]

REVIEWER COMMENTS

Reviewer #1 (Remarks to the Author):

PMN-PT based ferroelectrics are important piezoelectric materials for many applications, but the stability of their piezoelectric constant is a problem for their applications under high voltage or high temperature. The important contribution of this paper is a new material with a high piezoelectric constant and high E_c which is related to good stability. The paper can be accepted after necessary revision.

High tetragonal distortion is possibly helpful to increase the coercive field E_c as the author suggested, but E_c is not an intrinsic property of ferroelectrics. Authors need to consider other parameters to explain the high E_c in this work.

1, Normally E_c is dominated by domain wall density/structures and point defects including oxygen vacancies. Authors already discussed the effect of oxygen vacancies on E_c in hard PZT, but it is also necessary discuss the effect of domain walls on E_c . Poling and depoling is related to voltage induced domain switching which includes nucleation of new domains at a defect site (normally near domain walls) and domain growth. Compared to domain growth, the nucleation is the critical step, which control the domain switching, see discussions in papers from Dragon at EPFL and James Scott at university of Cambridge

2, The E_c is related to measured frequency. At low frequency, domain switching is much easy, which is related to domain switching under very low field and long holding time of applied fields, so it is necessary to provide PE and IE loops at 0.1Hz, 1Hz and 10 Hz to discuss their different E_c . There are clear contributions of conductivity in PE loops in Fig S1 sample 3. Author need always provide related IE loops to support discussions on PE loops.

3, In Fig 2d, is the sample for permittivity vs temperature poled or unpoled? There is a clear peak at 85 C in Fig2d, but there is no change of P_r and P_s near 85C in Fig S3b. it is clear the sample after PE test is poled, which indicates the field induced domain switching and phase transition from PNRs to domains. It is necessary to compare the dielectric data vs temperature of poled and unpoled samples. It is also necessary to provide XRD data of poled and unpoled samples at different temperatures to discuss the change of lattice structures, distortions, and different phase concentrations from room temperature to 200C with 10 or 20-degree step. Specifically make a link between d_{33} and structures of a poled sample because d_{33} is from a poled sample.

Reviewer #2 (Remarks to the Author):

This manuscript aims at solving a long-standing puzzle in the physics of strongly disordered ionic crystals, which is well-known as the so-called relaxor enigma. This has been deciphered only in the last ~10 years by virtue of the charge-disorder-induced random field concept. The manuscript of Prof. Gang Liu et al. on "Simultaneously achieving giant piezoelectricity and record coercive field enhancement in relaxor-based ferroelectric crystals" describes the successful preparation of Lead-containing ternary compositions with record-high values of both the coercive electric field, E_C , and the piezoelectric activities, d_{33} and d_{15} .

The authors from eight different Chinese institutions claim to first describe a novel ferroelectric material with both an extremely large coercive field, $E_C \sim 8.2$ kV/cm, and a giant piezoelectric activity, $d_{33}=2630$ pC/N and $d_{15}=490$ pC/N, in a scandium (Sc) doped ternary $\text{Pb}(\text{Sc}_{1/2}\text{Nb}_{1/2})\text{O}_3$ - $\text{Pb}(\text{Mg}_{1/3}\text{Nb}_{2/3})\text{O}_3$ - PbTiO_3 crystal with composition 0.06 PSN-0.49 PMN-0.51 PT.

However, the claim of novelty is not fully justified. Already 20 years ago Y P Guo et al. from the Shanghai Institute of Ceramics reported in *J. Crystal Growth* 226 (2001) 111 [uncited] on the growth and electrical properties of single crystalline 0.05 PSN-0.63PMN-0.32PT. They also found high values such as $d_{33}=1200$ pC/N and $\epsilon_{33} \sim 3500$ and predicted "a great potential for application in ultrasonic transducers and high strain actuators".

Only late in this manuscript the authors are taking notice of another earlier paper by Z Wang et al. in *Cryst. Growth Des.* 18 (2018) 145, which also deals with PSN-PMN-PT crystals under the title "In Situ Di-, Piezo-, Ferroelectric Properties and Domain Configurations of PSN-PMN-PT Ferroelectric Crystals". Their samples with slightly higher Sc content (> 10%) are described as "promising ferroelectrics, which are good candidates for high temperature and power applications".

Only reluctantly the authors develop plausible reasons for the efficiency of their related low-Sc compounds, which are prone to stabilize larger tetragonal microregions in close vicinity of extremely small polar nanoregions (PNR) (Fig. 5). In addition the present reviewer strongly recommends to explicitly mention the basic importance of inherent charge disorder and their quenched random electric fields [48, 49, 55] in the relaxor-type ternary systems.

Clearly, repairing of these deficiencies demands for a more focused discussion in the revised version of this manuscript. On this occasion also numerous linguistic and grammar shortcomings must be removed, e.g. on

line 39: ... a considerable amount of tetragonal phase is confirmed)...

line 44: ... This updating investigation ...

line 49: ... wide variety of ferroelectric device types exist...

line 90: The resulting ternary ...

line 94: ... while the ultrahigh ...

line 95: ... significance is that the EC was successfully increased...

lines 101 – 114: Figure 1. The dilemma of ferroelectrics: enhanced EC is usually achieved at the expense of piezoelectricity. (a) ... (b) ...

This “extended figure caption” contains both information on the figure and the physics behind. For sake of readability both of these elements must be separated into a concise caption and extra text for its interpretation.

line 133: ... the coercive field of the resulting)

line 147 ... are found to be in the order of ...

line 253: ... These observations are consistent with the data obtained

line 262: ... with a considerably enhanced content of tetragonal phase ...

line 309: ... with coexisting multi-phases ...

line 312: ... further improving the piezoelectric response...

line 321-322: ... polarization switch, while an easy polarization rotation ...

line 328 – 329: ... As reported by Wang et al. ⁵¹, in PSN-PMN-PT crystals ...

line 360: ... walls are further suggested by Rayleigh analysis

line 363: ... present reviewer employing a precise microstructure-by-design technique ...

line 380: ... The flattened potential well facilitates the polarization ...

line 381: ... response while the enhanced energy barrier ...

line 407: ... by Al₂O₃ grinding powder

After carefully amending the manuscript along the above remarks it should be acceptable for publication in NComms.

Reviewer #3 (Remarks to the Author):

The authors report on discovery of relaxor-ferroelectric solid solution which exhibits a large piezoelectric coefficient and a large coercive field. This is indeed desirable but rare property, because in most materials these two properties are anticorrelated. Having both properties large is of interest for piezoelectric devices working under high driving conditions.

The approach which the authors have chosen to accomplish their goal is to use a material with a large chemical heterogeneity (on mesoscale) with inclusions that have high coercive fields for complete

switching. They claim that such material possesses free energy where chemical heterogeneity provides flat landscape for polarization rotation under weak to moderate external fields while tetragonal inclusions provide very large barrier to complete polarization switching.

The authors have employed several characterization techniques to give support to their claims. They also performed modelling by phase-field computations of a generic system with inclusions to simulate their material.

The approach is in fact based on recent results by Li et al (Ref. 31) where it was shown that large enhancement of electro-mechanical properties may be expected in chemically disordered systems and a previous idea by the same authors that immovable inclusions within ferroelectric domains may inhibit motion of domain walls. (Li, F.; et al. The Origin of Ultrahigh Piezoelectricity in Relaxor-Ferroelectric Solid Solution Crystals. Nature Communications 2016, 7, 13807.)

The authors investigated difficult and complex problems and one can easily imagine a separate effort on each topic addressed in this work to obtain solid enough results that can serve as a definite proof of various claims made in the paper. While the paper is plausible, most of the arguments taken separately are rather hand waiving and vague, although taken together they do give credibility to authors' conjectures. I believe that the results are publishable – in principle. The authors certainly present a useful final result. The decision whether it is appropriate for this journal is, I think, dependent on editorial policy.

In my opinion the concept, material, approach...are not entirely original, as mentioned above. The earlier papers that opened up the path to authors are also vague and speculative, but those were the first papers opening up the field and indicating directions of further research. I think that in subsequent publications one should get confirmations of original ideas in form of strong evidence. Otherwise we just get a series of papers based on speculations and conjectures, and the present paper follows that trend. This may be acceptable if the main result is exceptionally important. In the case of this paper the result is of a rather limited general interest.

The paper is not well written. The authors use word "convince" several times and the meaning is never correct. Here are some examples:

- considerable amount of tetragonal phase is convinced.
- further convincing the experimentally observed microstructure and functionality.
- are further convinced by Rayleigh analysis
- convinced the experimental design.

There are other errors in style that distract from reading, some of which are given below. Also, I have few technical comments and questions:

Authors write: " Both a large coercive field (EC) and a giant piezoelectric activity are essential for the ferroelectric materials used in high-drive electromechanical applications, especially the case for greater power efficiency, smaller size, lighter weight, and lower overall cost "

What does cost has to do with E_c ?

The word "respectively" is not used used appropriately in Abstract.

"hard doping of manganese (Mn) " ...should be "by" manganese

"...without sacrificing, or even improving their giant piezoelectricity" is probably misstated

..."the latest piezoelectric activity over 4000 pC/N has been achieved" should probably be "...the recent studies achieved piezoelectric activity of over 4000 pC/N"

"...polarization switch behavior " should be "polarisation switching behavior"

"It is of interest to note that the ternary crystals possess significantly higher local strain (~3%) as compared to that of their binary counterparts (1.5%), also suggesting a higher level of tetragonal lattice deformation c/a . "

This is in fact rather close. What is the error or standard deviation for local strain values?

"Based on the above dissection,..." this should rather be "discussion"

"This strategy takes the advantages of both "hard" (large EC) and "soft" (high piezoelectric response) doping, ..."

There is only one doping or modification. It's better to write that "...strategy accomplishes both hard ...and soft...properties"

"After being poled by a DC E-field of 10 kV/cm ..." could authors specify crystallographic direction of the field during poling?

Response Letter

Journal: *Nature Communications*

Manuscript ID: NCOMMS-21-26075A-Z

Authors: Liya Yang, Houbing Huang, Zengzhe Xi, Limei Zheng, Shiqi Xu, Gang Tian, Yuzhi Zhai, Feifei Guo, Lingping Kong, Yonggang Wang, Weiming Lü, Long Yuan, Minglei Zhao, Haiwu Zheng, and Gang Liu

Title: Simultaneously achieving giant piezoelectricity and record coercive field enhancement in relaxor-based ferroelectric crystals

Dear Editor,

Thank you very much for handling our manuscript, we also appreciate the reviewers for their good suggestions and valuable comments. We have made the revisions based on the comments. Please see our detailed point-to-point response as follows. All changes made accordingly in the manuscript are highlighted in red. We hope all your concerns have been well addressed, and the quality of our paper has been greatly improved after the revisions.

REVIEWER COMMENTS

Reviewer #1 (Remarks to the Author):

Comment: PMN-PT based ferroelectrics are important piezoelectric materials for many applications, but the stability of their piezoelectric constant is a problem for their applications under high voltage or high temperature. The important contribution of this paper is a new material with a high piezoelectric constant and high E_C which is related to good stability. The paper can be accepted after necessary revision.

Reply: We greatly appreciate your positive overall comments and publication recommendation.

Comment: High tetragonal distortion is possibly helpful to increase the coercive field E_C as the author suggested, but E_C is not an intrinsic property of ferroelectrics. Authors need to consider other parameters to explain the high E_C in this work.

Normally E_C is dominated by domain wall density/structures and point defects including oxygen

vacancies. Authors already discussed the effect of oxygen vacancies on E_C in hard PZT, but it is also necessary to discuss the effect of domain walls on E_C . Poling and depoling is related to voltage induced domain switching which includes nucleation of new domains at a defect site (normally near domain walls) and domain growth. Compared to domain growth, the nucleation is the critical step, which controls the domain switching, see discussions in papers from Dragon at EPFL and James Scott at university of Cambridge.

Reply: Thanks a lot. It is indeed a very valuable comment. As mentioned by the reviewer, domain switching can be divided into two key processes: domain nucleation and domain growth, which is supported by a paper from Dr. James Scott (*Adv. Mater.* **22**, 5315 (2010)). As suggested by reviewer, we have performed additional experiments to investigate the domain nucleation and domain growth process by using PFM technique, and we experimentally confirm that both domain nucleation and domain growth are not favored under small electric field, therefore resulting in a greatly enhanced E_C in our 0.06PSN-0.61PMN-0.33PT crystal sample. The activation electric field E_A required for domain switching of 0.06PSN-0.61PMN-0.33PT is 6 times higher than that of 0.67PMN-0.33PT (138 kV/cm vs. 23 kV/cm), corresponding well with the enhanced potential barrier of the ternary system (please see main text Figure 5). Please see the following content for our detailed experimental results and discussion.

PFM technique was employed to explore the domain dynamic behavior. An area of $6 \times 6 \mu\text{m}^2$ is pre-poled by a tip voltage of -20 V to form an upward region. Then nanoscale domains in this region were reversed downward with different positive tip voltages and pulse durations. Figure R1 shows the out-of-plane phase image of the domains. For the 0.67PMN-0.33PT single crystal, only one nucleated domain grows till to complete a domain switching. The domain diameter increases with the increase of both tip voltage and pulse duration, and the domain growth is the predominant effect. These results are consistent with previous results on some other ferroelectrics crystals and films (*Nat. Commun.*, **11**, 394, (2020); *Adv. Mater.* **32**(4): 1907036 (2020); *J. Mater. Chem. C*, **5**, 2459 (2017)). However, for 0.06PSN-0.61PMN-0.33PT, the domain switching is expedited by an increased amount of nucleation sites. This is corroborated by the irregular shape of the switched area arising from many individually nucleated domains, which may coalesce during the growth. The multi-site domain nucleation phenomena has been observed in ferroelectric $\text{Pb}(\text{Zr},\text{Ti})\text{O}_3$ film, which contains a large amount of defect pinning centers (*Adv. Funct. Mater.* **27**, 1605196 (2017)). As such, it is reasonably deduced that the

domain growth in 0.06PSN-0.61PMN-0.33PT is severely inhibited by the high concentration of pinning centers, *e.g.*, tetragonal nanoclusters, and domain nucleation is the dominant effect for domain dynamics.

In addition, domain nucleation in the 0.06PSN-0.61PMN-0.33PT is also more difficult than in the 0.67PMN-0.33PT. The domains in the yellow rectangle are fabricated at different pulse durations with a fixed tip voltage of 7 V. As can be seen, domain nucleation occurs with a smaller pulse duration in the binary system than in the ternary one. On the other hand, with a fixed pulse duration of 3 s, the domain nucleation also happens at a lower tip voltage (domains in the red rectangle) for the 0.67PMN-0.33PT. These results further support the observed higher E_C in ternary crystals.

Figure R1. Domain switching behavior under different tip voltage and pulse duration.

We then consider the dynamics of domain switching. The activation electric field E_A required for domain switch is determined by Merz's law. Figure R2a shows the P - E hysteresis loops measured at different frequencies f . The frequency dependent E_C can be depicted by Merz's law (*Adv. Funct. Mater.* **22**, 2148 (2012); *Acta Mater.* **157**, 355 (2018); *J. Mater. Chem. C* **9**, 2426 (2021)):

$$\tau \propto 1/f \propto \exp(E_A/E_C) \quad (\text{R1})$$

where τ is the switching time. Figure R2b gives the fitting results of the experimental data. A larger value of $E_A=138$ kV/cm was derived for 0.06PSN-0.61PMN-0.33PT, in comparison of 23 kV/cm for

0.67PMN-0.33PT. The higher E_A of ternary system means a more difficult domain reverse process, further verifying the proposed deep potential barrier ΔG (Figure 5).

Figure R2. Ferroelectric domain switching dynamics. (a) Frequency dependence of P - E hysteresis loops for the 0.06PSN-0.61PMN-0.33PT single crystal. (b) $1/E_C$ vs. $\ln f$ of 0.06PSN-0.61PMN-0.33PT and 0.67PMN-0.33PT crystals. The solid lines are the fitting results of Merz’s law.

In the revised manuscript, the papers from Dr. Dragan Damjanovic and Dr. James Scott are cited as Refs 52, 53 and 38, 39 in the Main text, respectively; and in the Supplementary as Refs 19, 20 and 21, 23, respectively. The above experimental results and discussions are presented in the Supplementary as “**Note 13. Domain switching**” in Page 18. Also, in the Main text (Line 20 on Page 6 and Line 2 on Page 16), we added the discussions:

“Note that E_C is not an intrinsic property of ferroelectrics^{38,39}. Poling/de-poling is related to voltage-induced domain switching, including nucleation of new domains at a defect site (normally near domain walls) and domain growth. Thus, we considered the dynamics of domain switching by measuring the E_C at various frequencies. As is well known, domain switching is considerably easy at low frequencies, which is related to domain switching under a very low field and long holding time of the applied fields. For the 0.06PSN–0.61PMN–0.33PT crystals, a large E_C of $\sim 7.5 \text{ kV}/\text{cm}$ is maintained at frequencies as low as 0.1 Hz, and 11.8 kV/cm is afforded at 100 Hz (Supplementary Note 2 and Figure S2). Moreover, we observed a low conductive current during the entire poling process, confirming the high quality of the crystals (Supplementary Note 3).”

“The pinning effect of the tetragonal polar regions on domain switch is associated with the difficult

domain nucleation and growth process^{52,53}, which is supported by the large activation electric field in 0.06PSN–0.61PMN–0.33PT (Supplementary Note 13).”

Comment: The E_C is related to measured frequency. At low frequency, domain switching is much easy, which is related to domain switching under very low field and long holding time of applied fields, so it is necessary to provide P - E and I - E loops at 0.1 Hz, 1 Hz and 10 Hz to discuss their different E_C . There are clear contributions of conductivity in P - E loops in Fig S1 sample 3. Author need always provide related I - E loops to support discussions on P - E loops.

Reply: Thank you very much for this valuable comment. We measured P - E hysteresis loops of 0.06PSN-0.61PMN-0.33PT at various frequencies (from 0.1 Hz to 100 Hz). We found that the crystal still exhibits a relatively high E_C of 7.5 kV/cm at lowest frequency, 0.1 Hz. We also provided I - E loops of 4 different samples. All the samples except Sample 3 in Figure S1 demonstrate excellent insulation behavior. The slight leakage current in Sample 3 is caused by the partial breakdown due to the repeated high electric field applied on the sample. Please see the following content for our detailed experimental results and discussion.

Figure R2a shows the P - E hysteresis loops of 0.06PSN-0.61PMN-0.33PT at the frequency (f) of 0.1-100 Hz. The frequency dependence of E_C was summarized in Figure R3, from which one can see E_C increases from 7.5 kV/cm to 11.8 kV/cm as f increases from 0.1 Hz to 100 Hz. The frequency dependent E_C can be described by the theoretical model from Ishibashi and Orihara (*Integr. Ferroelectr.***12**, 71(1996); *Phys. Status Solidi B* **252**, 833 (2015)):

$$E_C = Kf^\beta \quad (R2)$$

And our experimental data in Figure R3 can be well described by Equation (R2) with derived parameters of $K=2.51$ and $\beta=0.07$. According to relationship, the coercive field at other frequencies can be estimated.

Figure R4a shows the I - E loops of Sample 4 (the sample used in Figure R2) measured at different frequencies. For each frequency, a current peak can be observed around E_C , corresponding to the displacement current originated from domain switching. The maximum displacement current increases with frequency for the shortened switching time. Figure R4b summarizes I - E loops of 4 different samples measured at a fixed frequency of 1 Hz (Sample 1-3 correspond to the 3 samples in

Supplementary Figure S1). For all the samples, the conductive current is negligible in comparison with the displacement current. The enlarged I - E curves are provided as the inset of Figure R4b. Only Sample 3 demonstrates a detectable leakage current while other samples show excellent insulation effect.

Figure R3. Frequency dependent E_C for 0.06PSN-0.61PMN-0.33PT. Fitting the experimental data by Eq. (R2) gives $E_C=2.51f^{0.07}$.

Figure R4. I - E loops of the PSN-PMN-PT single crystals. (a) Frequency dependence of I - E loops for Sample 4. (b) I - E loops of 4 different samples measured at 1 Hz.

Therefore, we can conclude that, as reviewer said, E_C is related to measured frequency. At low frequency, domain switching is much easy, which is related to domain switching under very low field

and long holding time of applied fields. In addition, our additional experiments give a quantitative description of this behavior for PSN-PMN-PT crystals.

The above experimental results and discussions have been added into the Supplementary document as “**Note 2. Repeatability and frequency dependence of coercive field**” (Page 3) and “**Note 3. Current-electric field (*I-E*) loops**” (Page 4). In the main text (Line 20 on Page 6), we added the text below:

“Note that E_C is not an intrinsic property of ferroelectrics^{38,39}. Poling/de-poling is related to voltage-induced domain switching, including nucleation of new domains at a defect site (normally near domain walls) and domain growth. Thus, we considered the dynamics of domain switching by measuring the E_C at various frequencies. As is well known, domain switching is considerably easy at low frequencies, which is related to domain switching under a very low field and long holding time of the applied fields. For the 0.06PSN–0.61PMN–0.33PT crystals, a large E_C of ~7.5 kV/cm is maintained at frequencies as low as 0.1 Hz, and 11.8 kV/cm is afforded at 100 Hz (Supplementary Note 2 and Figure S2). Moreover, we observed a low conductive current during the entire poling process, confirming the high quality of the crystals (Supplementary Note 3).”

Comment: In Fig 2d, is the sample for permittivity vs temperature poled or unpoled? There is a clear peak at 85 °C in Fig 2d, but there is no change of P_r and P_s near 85 °C in Fig S3b. It is clear the sample after *P-E* test is poled, which indicates the field induced domain switching and phase transition from PNRs to domains. It is necessary to compare the dielectric data vs temperature of poled and unpoled samples. It is also necessary to provide XRD data of poled and unpoled samples at different temperatures to discuss the change of lattice structures, distortions, and different phase concentrations from room temperature to 200 °C with 10 or 20-degree step. Specifically make a link between d_{33} and structures of a poled sample because d_{33} is from a poled sample.

Reply: We would like to thank the referee for inspiring us for more detailed structural studies. Please see the following content for our detailed experimental results and discussion on this comment.

The comparison of dielectric constant vs. temperature curve of poled and unpoled samples:

In Figure 2d, the sample for permittivity vs temperature is poled. In Figure R5 we compare the dielectric permittivity vs. temperature curves between poled and unpoled samples. It is well accepted

that during poling process, PNRs transformed into macro ferroelectric domains (*Phys. Rev. Appl.* **11**, 044032, (2019); *J. Eur. Ceram. Soc.* **36**, 515 (2016); *Ceram. Int.* **42**,18631 (2016); *J. Phys. D: Appl. Phys.* **49**, 175301 (2016)), thus after poling the sample has macro ferroelectric domains at room temperature. The dielectric anomaly at $T_{F-F}=85$ °C of the poled samples corresponds to the temperature-driven ferroelectric to ferroelectric phase transition, along with the partial depoling of the $[001]_C$ poled domain structures. For the unpoled sample, however, the PNR state is always maintained, and no apparent phase structure change occurs below T_m (for details please see the data shown in Figure R7c), thus the dielectric anomaly corresponding to the ferroelectric-ferroelectric phase transition is not detectable. Other relaxor ferroelectrics also demonstrate similar dielectric behavior (*J. Appl. Phys.* 108, 034112 (2010)).

Figure R5. Comparison of dielectric permittivity vs. temperature curve of poled and unpoled 0.06PSN-0.61PMN-0.33PT single crystals.

The inconspicuous change of P_r and P_s near 85 °C in Figure S3b:

The inconspicuous change of P_r and P_s near T_{F-F} should mainly ascribed to the multi-phase coexistence and the slow changes of phase structures newer T_{F-F} (for details please see the data shown in Figure R7d). It is known that the influence of phase transitions on the P_r (or P_s) is complicated, and it depends on both the direction of the applied electric field and the polar vectors of the various

ferroelectric phases. For example, when measured along $[111]_C$, the rhombohedral-tetragonal phase transition would cause reduction of P_r (or P_s) as polar vector changes from $[111]_C$ to $[001]_C$. When measured along $[001]_C$, however, P_r (or P_s) increases around this phase transition (*CrystEngComm* **21**, 348 (2019)). In addition, as the temperature approaching Curie temperature, P_r (or P_s) should decrease with temperature without ferroelectric-ferroelectric phase transition. Thus, the variation of P_r (or P_s) at T_{F-F} is affected by combined effects of temperature-induced P_s reduction and the change of the polar direction. In this work, 0.06PSN-0.61PMN-0.33PT single crystal contains M_A , M_C and tetragonal phases below 140 °C. The phase concentrations change gently rather than abruptly at T_{F-F} (Figure 7d). Thus, the temperature-induced P_s reduction should be the dominant effect, therefore giving an inconspicuous change of P_s and P_r near 85 °C.

Phase structure difference between poled and unpoled samples:

For $[001]_C$ oriented crystal, power XRD characterizations only give information of $(00m)$ ($m=1, 2, 3\dots$) Bragg peaks, thus it is difficult to determine the phase component and lattice parameters using a single crystal sample. Alternatively, we employed polycrystalline ceramics with the same composition as crystal for a detailed temperature-dependent structural study. The main difference between single crystal and ceramics arises from extrinsic factors such as grain boundary, while XRD patterns generally reflect the distortion of the lattice structure, which is an intrinsic factor. Although there may exist a slight difference in structures between single crystal and ceramics, the influence of electric field and temperature should be dominated factor for intrinsic structures here. Thus we believe the results based on ceramics can provide us with quotable information.

Figure R6 demonstrates the XRD patterns of the poled and unpoled samples. At $2\theta\sim 45^\circ$, the high-angle peak is strengthened after poling, indicating that the domains switch to directions close to the E -field. The narrower diffraction peaks in the poled sample demonstrate more uniform domain structure. The phase concentrations and lattice parameters for each phase are listed in Table R1. After poling, the component of M_C phase and tetragonal (T) phase increase while the M_A phase decreases.

Figure R6. XRD patterns for the poled and unpoled samples.

Table R1. Phase component and lattice parameters for the poled and unpoled samples.

PSN-PMN-0.32PT	Unpoled				Poled			
	a (Å)	b (Å)	c (Å)	Volume fraction	a (Å)	b (Å)	c (Å)	Volume fraction
M_A	5.732	5.687	4.009	43.9%	5.719	5.667	4.000	17.5%
M_C	4.065	3.973	4.032	22.7%	4.037	3.993	4.054	44.3%
T	4.001	4.001	4.053	33.4%	4.018	4.018	4.046	38.2%

Temperature dependent phase structures

The XRD patterns as a function of temperature for unpoled and poled samples are shown in Figures R7a and R7b, respectively. The phase fractions are also determined according to our refinements, as shown in Figure R7c and R7d, respectively. With the increase of temperature, M_C phase fraction gradually increases. For the poled samples, the phase fraction changes obviously around T_{F-F} , corresponding well with remarkable dielectric anomaly in Figure R5. The cubic phase begins to appear at around 140 °C, and becomes the dominant phase as the temperature increases.

The relationship between piezoelectric response and structure

Above T_{F-F} , the piezoelectric reduces significantly for the partial depoling during the phase transition. So we focus on the piezoelectric response below T_{F-F} and discuss the relationship between piezoelectric activity and phase structure.

The total piezoelectric response is the integral effect from all ferroelectric phases: T, M_A and M_C . The tetragonal phase is associated with low d_{33} along $[001]_C$ (*Appl. Phys. Lett.* **117**, 052904 (2020); *Appl. Phys. Lett.* **113**, 102903 (2018)), and high d_{33} in the 0.06PSN-0.61PMN-0.33PT origins mainly from M_A and M_C phases where P_S rotation contributes greatly. The total volume fraction of M_A+M_C increases with temperature (Figure R8a). In addition, the lattice distortion strengthens with temperature, as demonstrated by the increased β (the angle between lattice a and c) value (Figure R8b). The larger β corresponds to a stronger deviation of P_S vector from $[001]_C$, which is more responsive to the external electric field and facilitates piezoelectricity. The enhanced monoclinic phase fraction, the increased lattice distortion, together with the softened crystal lattice at high temperature give a monotonously enhanced d_{33} with temperature (Figure S4)

Figure R7. Temperature dependent XRD patterns and calculated phase fractions for (a) and (c) unpoled and (b) and (d) poled samples.

The above experimental results and discussion have been added in the Supplementary as “**Note 10. Electric field- and temperature-dependent structural evolutions**” on Page 13. In the revised manuscript (Line 16, Page 12), we added “Electric-field- and temperature-dependent structural evolutions were also conducted, and the structure–piezoelectricity relation was studied (Supplementary Note 10).”.

Figure R8. Temperature dependent (a) monoclinic phase fraction and (b) β of M_A and M_C .

Reviewer #2 (Remarks to the Author):

Comment: This manuscript aims at solving a long-standing puzzle in the physics of strongly disordered ionic crystals, which is well-known as the so-called relaxor enigma. This has been deciphered only in the last ~10 years by virtue of the charge-disorder-induced random field concept. The manuscript of Prof. Gang Liu et al. on "Simultaneously achieving giant piezoelectricity and record coercive field enhancement in relaxor-based ferroelectric crystals" describes the successful preparation of Lead-containing ternary compositions with record-high values of both the coercive electric field, E_C , and the piezoelectric activities, d_{33} and d_{15} .

The authors from eight different Chinese institutions claim to first describe a novel ferroelectric material with both an extremely large coercive field, $E_C \sim 8.2$ kV/cm, and a giant piezoelectric activity, $d_{33}=2630$ pC/N and $d_{15}=490$ pC/N, in a scandium (Sc) doped ternary $\text{Pb}(\text{Sc}_{1/2}\text{Nb}_{1/2})\text{O}_3\text{-Pb}(\text{Mg}_{1/3}\text{Nb}_{2/3})\text{O}_3\text{-PbTiO}_3$ crystal with composition 0.06PSN-0.49 PMN-0.51 PT.

Reply: We thank the reviewer for considering our manuscript as “the successful preparation of Lead-containing ternary compositions with record-high values of both the coercive electric field, E_C , and the piezoelectric activities, d_{33} and d_{15} ” and “first describe a novel ferroelectric material with both an extremely large coercive field, $E_C \sim 8.2$ kV/cm, and a giant piezoelectric activity”. These properties achieved in this work is indeed a breakthrough for the bottle-neck issue and dilemma in ferroelectrics shown in Figure 1, namely, an enhanced E_C is usually achieved at the expense of piezoelectricity.

Comment: However, the claim of novelty is not fully justified. Already 20 years ago Y P Guo et al. from the Shanghai Institute of Ceramics reported in J. Crystal Growth 226 (2001) 111 [uncited] on the growth and electrical properties of single crystalline 0.05PSN–0.63PMN-0.32PT. They also found high values such as $d_{33}=1200$ pC/N and $\epsilon_{33}\sim 3500$ and predicted “a great potential for application in ultrasonic transducers and high strain actuators”.

Only late in this manuscript the authors are taking notice of another earlier paper by Z Wang et al. in Cryst. Growth Des. 18 (2018) 145, which also deals with PSN-PMN-PT crystals under the title “In Situ Di-, Piezo-, Ferroelectric Properties and Domain Configurations of PSN–PMN–PT Ferroelectric Crystals”. Their samples with slightly higher Sc content ($> 10\%$) are described as “promising ferroelectrics, which are good candidates for high temperature and power applications”.

Reply: Thank you very much for this comment. This comment is very valuable for us to improve the quality of manuscript and especially, to justify the novelty of this work clearer and more comprehensive. The works by Y P Guo *et al* and Z Wang *et al* are undoubtedly pioneer results on PSN-PMN-PT ternary system, providing basic characterizations in this system. [The paper from Y P Guo (*J. Crystal Growth* 226 (2001) 111) has been added in revised manuscript.] Although the systems studied by Y P Guo *et al* and Z Wang *et al* are similar as ours, we note that in their works, we still have to face a formidable materials science challenge, namely, there is a trade-off between E_C and piezoelectricity. In their works, either piezoelectricity or E_C , or both, is relatively low. Therefore, we must explore a novel design route and new physical mechanism to further improve both properties, and our experimental result presented in current work has demonstrated a success. We would like verify the novelty of our work by comparing our result with two above mentioned works as follows.

(1) Novelty about material properties (performance): Our 0.06PSN-0.61PMN-0.33PT crystal simultaneously achieves large E_C and high piezoelectric response, while other two works do not. Table R2 lists the main performance and physical parameters of the PSN-PMN-PT system from all three studies: our work, Work 1 (Y. P. Guo, *J. Crystal Growth* 226, 111 (2001)) and Work 2 (Z Wang, *Cryst. Growth Des.* 18, 145 (2018)). One can clearly see that our work demonstrates superior properties: d_{33} ~2630 pC/N is 2 times higher than those in Work 1 (~1200 pC/N) and Work 2 (1260-1550 pC/N), and meanwhile E_C ~8.2 kV/cm is much larger than those in Works 1 and 2 (4-6 kV/cm). In addition, we observed that the electromechanical coupling factor (k_{33} ~0.9) and dielectric constant (ϵ_{33} ~5950) of our crystals are also superior. The PSN-PMN-PT crystals in Works 1 and 2 demonstrate relatively low d_{33} and moderate E_C , comparable to the PIN-PMN-PT:Mn and PSN-PT systems, at an average level among all the data shown in Figure 1a. Similar to other works, the enhancement of E_C in Works 1 and 2 is at the expense of piezoelectricity d_{33} . Alternatively, our crystal achieves simultaneous ultrahigh piezoelectricity and extremely large coercive field, far beyond the boundary (see the red dash line in Figure 1a) of other systems, getting access the previously “no data” region.

(2) Novelty about material structure (mechanism): The special microstructure, highly dispersed heterogeneous structures with considerable content of tetragonal pinning centers are formed in our 0.06PSN-0.61PMN-0.33PT, which is the critical factor to achieve synergetic enhancement of piezoelectricity and coercive field. Obviously, this microstructure is not formed in Works 1 and 2. The diffuseness factor γ for the sample in Work 1 is 1.72, and 1.65-1.82 in Work 2 (Table R2 and Figure R9),

much lower than the value in our work ($\gamma=1.94$). In addition, the crystals in Work 2 demonstrate a much larger domain size. Both small γ value and large domain size verify less dispersed micro polar structure, therefore giving low piezoelectric properties.

One may wonder why this special micro structure is formed only in our crystal while not in others. Two factors are particularly important for the formation of this microstructure: 1) the MPB composition. In the vicinity of MPB, both micro domain structure and functional properties depends greatly on the composition, for the phase transforms from rhombohedral phase to MPB and further to tetragonal phase in a small composition range of ~5% PbTiO_3 content. For the 0.05PSN-0.63PMN-0.32PT crystal in Work 1, the sample is in the pure rhombohedral phase, thus the dispersed heterogeneous structures with considerable amount of tetragonal component are hardly formed; 2) the highly disordered state. The low PSN content is favorable for a disordered state, while high PSN content may favors ordered state (see detailed explanation in the response to the next comment, Page 16 of this letter). For the samples in Work 2, although the material composition is located in MPB region, the high content of PbTiO_3 and PSN results in a high level of ordered state, and therefore a larger domain size and less dispersed microstructure is established. Although enhanced coercive field in Works 1 and 2 is observed in comparison with PMN-PT system, however, this enhancement arises from the high coercive nature of PSN, different from the mechanism with our work, and resulting in an inferior piezoelectricity.

As discussed above, our work not only fabricates a high-performance ferroelectric material but also provides an appealing and simple route for improving overall functionalities in ferroelectrics. We believe the present work will rekindle the confidence and research enthusiasm of the PbTiO_3 -based relaxor ferroelectrics, and greatly promote the development of this field. The work by Y P Guo has been cited as Ref 57 in the revised manuscript. In the main text of revised manuscript (Line 16 on Page 17), we added:

“Guo et al.⁵⁹ reported that 0.05PSN–0.63PMN–0.32PT single crystals in R phase exhibit inferior performance with $d_{33} = 1200$ pC/N and $\epsilon_{33} \sim 3500$. Compared to previous studies, our 0.06PSN–0.61PMN–0.33PT single crystals exhibit substantially superior overall performance (Supplementary Note 15).”

The above discussions have been added into the Supplementary as “**Note 15. Property comparison among various PSN-PMN-PT crystals**” on Page 22.

Table R2. Comparison of the main performance for PSN-PMN-PT system.

	This work	Work 1	Work 2
Composition	0.06PSN-0.61PMN-0.33PT	0.05PSN- 0.63PMN-0.32PT	yPSN-z0.63PMN-xPT $x=0.12-0.13, z=0.52-0.48;$ $x=0.26-28, z=0.48-0.435;$ $x=0.3675-0.3975, z=0.51-0.47$
Phase structure	MPB	R	R, MPB, T
d_{33} (pC/N)	2630	1200	1260-1550
E_C (kV/cm)	8.2	4-6 ^b	4-6
ϵ_{33} (ϵ_0)	5950	3500	1700-2000
k_{33}	0.9	-	~ 0.75
γ	1.96	1.73 ^a	1.65-1.82 ^a
T_{F-F} (°C)	85	70	120-180
T_C (°C)	152	162	200-240
Domain size (nm)	~200	-	~ 1000

^a The diffuseness factor γ is not provided in the reference, we calculated this value based on the dielectric permittivity vs. temperature curves, as shown in Figure R9.

^b E_C value is not provided in the paper. Generally, the ferroelectrics are poled by a field above twice of coercive field. Considering that the poling electric field is 10 kV/cm, E_C is estimated to be 4-6 kV/cm.

Figure R9. Diffuse factor γ for the samples studied in Work 1 and 2.

Comment: Only reluctantly the authors develop plausible reasons for the efficiency of their related low-Sc compounds, which are prone to stabilize larger tetragonal microregions in close vicinity of extremely small polar nanoregions (PNR) (Fig. 5). In addition, the present reviewer strongly recommends to explicitly mention the basic importance of inherent charge disorder and their quenched random electric fields [48, 49, 55] in the relaxor-type ternary systems.

Clearly, repairing of these deficiencies demands for a more focused discussion in the revised version of this manuscript.

Reply: We appreciate this very valuable comment. We would like to emphasize that the introduction of small amount of PSN is critical to the formation of the particular microstructure. And we re-consider the

importance of random electric fields. Please see the following content for our detailed experimental results and discussion on this comment.

First, the introduction of a small amount of trivalent Sc³⁺ ions facilitates a highly dispersed microstructure and therefore strong random field. It is known that the inherent charge disorder of the relaxor-PbTiO₃ ferroelectrics is due to the random distribution of various B-site ions such as Mg²⁺, Nb⁵⁺ and Ti⁴⁺. It offers uncorrelated and quenched random electric fields at the sites of the ferroelectric-active ions. These random fields favor the formation of PNRs. The dynamic PNRs begin to appear at Burn temperature T_B . The interaction between PNRs and quenched random electric fields results in a local mesoscale phase transition at T^* , below which the static PNR appears, coexisting with dynamic ones. On further cooling, the slowdown of PNR dynamics result in a remarkable dielectric relaxation. Finally, below freezing temperature, T_{VF} , the PNR dynamics ultimately slow down into a totally static glassy-like state (*Phys. Rev. Lett.* **68**, 847 (1992); *Phys. Rev. B* **77**, 054105 (2008); *Phys. Status Soli. B*, **251**, 1993 (2014); *Phys. Rev. Lett.* **97**,137601(2006)).

In our work, the introduction of a small amount of trivalent Sc³⁺ ions make the B-site ions more disordered than the PMN-PT system, thus a stronger random field formed. As a result, the dynamic PNRs appear at a higher temperature, as verified by the experimental results that the T_B of 0.06PSN-0.61PMN-0.33PT is 40 °C higher than the 0.67PMN-0.33PT (Figure 3a). The highly disordered state and strong random field can be further confirmed by the higher diffused factor γ (Figure 3b), the lower freezing temperature T_{VF} (Figure 3f), the larger T_m shift (Figure 3c and Figure S10), the smaller domain size and the shorter correlation length ζ (Figure S8).

It should be noted that to obtain highly disordered state, the amount of doped Sc³⁺ should be small. If the system contains a large amount of PSN, the chemically ordered regions with regular Sc³⁺-Nb⁵⁺ ordering could be easily established, and a weak but homogeneous local field is formed at the chemically ordered regions, giving raise to preferential local order ferroelectric domain (*Phys. Rew. Lett.* **97**, 137601 (2006)). Thus, the PSN-PMN-PT with a high amount of PSN demonstrates weak relaxor behavior, large domain size, and inferior performance (as shown in Work 2, for data please see Table R2).

Second, the introduction of PSN facilities the formation of tetragonal phase. Besides the highly charge disorder, the introduction of PSN favors large lattice deformation (Supplementary Note 1), and therefore

strong random local stress is also formed. The random stress is supposed to provoke the formation of highly distorted ferroelectric regions, such as tetragonal domains, which act as pinning centers in the PSN-PMN-PT system. The larger fractal dimension $d=2.31$ (Figure S6b) for the 0.06PSN-0.61PMN-0.33PT in comparison with 0.67PMN-0.33PT ($d=2.07$) refers to high domain wall roughness, where local pinning is strong and difficult to be overcome.

We further carry out the autocorrelation analysis as a function of temperature to understand the distorted state at various temperatures, results are shown in Figures R10a and R10b. The temperature dependent correlation radius ζ and fractal dimension d are summarized in Figures R10c and R10d, respectively. Initially, both ζ and d change slightly with temperature, demonstrating a stable domain structure and random fields with temperature. As approaching freezing temperature T_{VF} , ζ increases by 2 times (from 85 nm to 171 nm) and d decreases dramatically, indicating that the local structure becomes more ordered. Above T_{VF} , some of the static PNRs transform into dynamic ones, the system becomes more disordered, resulting in the decrease of ζ and increase of d . Nevertheless, at 180 °C, 25 °C above T_m , the $\zeta \sim 30$ nm is still not negligible, denoting the existence of quasi-static PNRs. With temperature further increasing above 180 °C, the quasi-static PNRs are not detectable, only dynamic PNSs exist.

Figure R10. Temperature dependent autocorrelation analysis. (a) The autocorrelation images at various temperatures. (b) The temperature evolution of the averaged autocorrelation function and fitting results. (c) and (d) average autocorrelation length ζ and fractional dimension d as a function of temperature,

respectively.

The discussions on the charge disorder and random electric fields are presented in the Supplementary (Page 17) as “**Note 12. Highly disordered structure and random fields**”. The temperature dependent autocorrelation analysis is combined with the room temperature autocorrelation analysis in Supplementary (Page 9) as “**Note 6. Autocorrelation function of the polar regions**”.

Comments: On this occasion also numerous linguistic and grammar shortcomings must be removed, e.g. on

line 39: ... a considerable amount of tetragonal phase is **confirmed**)...

line 44: ... This **updating** investigation ...

line 49: ... wide variety of ferroelectric device types **exist**...

line 90: The **resulting** ternary ...

line 94: ... **while** the ultrahigh ...

line 95: ... significance is **that** the EC was successfully increased...

lines 101 – 114: Figure 1. The dilemma of ferroelectrics: enhanced EC is usually achieved at the expense of piezoelectricity. (a) ... (b) ...

This “extended figure caption” contains both information on the figure and the physics behind. For sake of readability both of these elements must be separated into a concise caption and extra text for its interpretation.

line 133: ... the coercive field of **the resulting**)

line 147 ... are found to be **in** the order of ...

line 253: ... These observations are **consistent** with the data obtained

line 262: ... with **a considerably enhanced content of** tetragonal phase ...

line 309: ... with **coexisting** multi-phases ...

line 312: ... further **improving** the piezoelectric response...

line 321-322: ... polarization switch, **while** an easy polarization rotation ...

line 328 – 329: ... As reported by **Wang et al.**⁵¹, in PSN-PMN-PT crystals ...

line 360: ... walls are further **suggested** by Rayleigh analysis

line 363: ... present reviewer employing a **precise** microstructure-by-design technique ...

line 380: ... The flattened potential well **facilitates** the polarization ...

line 381: ... response **while** the enhanced energy barrier ...

line 407: ... by Al₂O₃ **grinding** powder

After carefully amending the manuscript along the above remarks **it should be acceptable for publication in NComms.**

Reply: We would like to show our great appreciation for such a careful reading, and we are greatly encouraged by the comments “After carefully amending the manuscript along the above remarks it should be acceptable for publication in NComms”. The linguistic and grammar shortcomings are corrected accordingly. Also, the caption of Figure 1 has been simplified as:

“Figure 1 Dilemma in ferroelectrics: enhanced E_C is usually achieved at the expense of piezoelectricity. (a) d_{33} vs. E_C for various relaxor-PT single crystals. The red dashed line denotes the tendency of most relaxor-PT crystals. Generally, E_C enhancement is accompanied by inferior piezoelectricity. Alternatively, our 0.06PSN–0.61PMN–0.33PT (red star) affords highly remarkable results. Data from Refs. 9, 11–30 and this work. (b) Schematic of the different free energy landscapes and the corresponding macroscopic performances.”

We also have asked AIP publishing to provide professional language service, therefore linguistic and grammar shortcomings have been removed. Figure R11 gives a document certifies that the manuscript was subject matter expert language edited by Author Services. Their highly qualified, native English-speaking editors corrected English language usage, grammar, punctuation, and spelling.

EDITORIAL CERTIFICATE

This document certifies that the following manuscript was subject matter expert language edited by Author Services from AIP Publishing. Our highly qualified, native English-speaking editors corrected English language usage, grammar, punctuation, and spelling.

Paper Title:

Simultaneously achieving giant piezoelectricity and record coercive field enhancement in relaxor-based ferroelectric crystals

Author:

丽雅 杨

Date certificate issued:

March 10, 2022

authorservices.aip.org

Figure R11 AIP editorial certificate

Reviewer #3 (Remarks to the Author):

Comment: The authors report on discovery of relaxor-ferroelectric solid solution which exhibits a large piezoelectric coefficient and a large coercive field. This is indeed desirable but rare property, because in most materials these two properties are anticorrelated. Having both properties large is of interest for piezoelectric devices working under high driving conditions.

The approach which the authors have chosen to accomplish their goal is to use a material with a large **chemical heterogeneity** (on mesoscale) with inclusions that have high coercive fields for complete switching. They claim that such material possesses free energy where chemical heterogeneity provides flat landscape for polarization rotation under weak to moderate external fields while tetragonal inclusions provide very large barrier to complete polarization switching.

The authors have employed several characterization techniques to give support to their claims. They also performed modelling by phase-field computations of a generic system with inclusions to simulate their material.

Reply: We greatly thank the reviewer for pointing out the simultaneous achievement of large piezoelectricity and large coercive field “is indeed desirable but rare property, because in most materials these two properties are anticorrelated”, and especially he/she mentioned that “Having both properties large is of interest for piezoelectric devices working under high driving conditions.” Specially, the observed E_C enhancement (from ~2.5 to 8.2 kV/cm) is the largest for ultrahigh-piezoelectric ferroelectrics, far exceeding any previously reported enhancements induced by the doping effects and other techniques.

Comment: The approach is in fact based on recent results by Li et al (Ref. 31) where it was shown that large enhancement of electro-mechanical properties may be expected in chemically disordered systems and a previous idea by the same authors that immovable inclusions within ferroelectric domains may inhibit motion of domain walls. (Li, F.; et al. The Origin of Ultrahigh Piezoelectricity in Relaxor-Ferroelectric Solid Solution Crystals. Nature Communications 2016, 7, 13807.)

The authors investigated difficult and complex problems and one can easily imagine a separate effort on each topic addressed in this work to obtain solid enough results that can serve as a definite proof of various claims made in the paper. While the paper is plausible, most of the arguments taken separately

are rather hand waiving and vague, although taken together they do give credibility to authors' conjectures. I believe that the results are publishable – in principle. The authors certainly present a useful final result. The decision whether it is appropriate for this journal is, I think, dependent on editorial policy.

In my opinion the concept, material, approach...are not entirely original, as mentioned above. The **earlier papers** that opened up the path to authors are also vague and speculative, but those were the first papers opening up the field and indicating directions of further research. I think that in subsequent publications one should get confirmations of original ideas in form of strong evidence. Otherwise we just get a series of papers based on speculations and conjectures, and the present paper follows that trend. This may be acceptable if the main result is exceptionally important. In the case of this paper the result is of a rather limited general interest.

Reply: We appreciate the comment from the reviewer with “I believe that the results are publishable – in principle. The authors certainly present a useful final result”. Also, we thank the novelty concern of the reviewer and would like to rephrase the novelty of this research and the difference between our work and previous studies, furthermore we provide more strong evidence to support our conception. To address the concern of present reviewer, we would like to clarify the novelty and originality of this work in terms of “the concept”, “material” and “approach” as mentioned. We also would like to discuss the relationship and difference between our work and earlier paper by Li *et al* as mentioned.

The novelty and originality of this work:

(1) The concept. In this work, we aimed at the simultaneous enhancement of the piezoelectric response and E_C in relaxor-based ferroelectrics, which is a **debut phenomenon surpassing the conventional anticorrelated relationship** between piezoelectric response and E_C . So we believe it is exceptionally important for various high-drive applications. From the view of physics, it is extremely difficult to simultaneously achieve high piezoelectricity and large coercive field, because high piezoelectricity requires easy P_S rotation and domain switch, while large E_C demands difficult P_S rotation and domain switch. A new concept should be proposed to improve the overall performance. We propose a special “flat and deep” free energy landscape to simultaneously realize easier P_S rotation and difficult domain switch. The flat potential well gives an easier P_S rotation under weak external field, while the enhanced potential barrier makes complete domain reverse difficult. On the basis of this peculiar free energy

profile, the unordinary property enhancement with both giant piezoelectric response and extremely high E_C could be achieved.

(2) The material. The low E_C is one of the most concerned issues for traditional PMN-PT crystals which greatly limits its applications. Many alternative systems have been developed in the past two decades to overcome these issues. However, the E_C improvement always accompanies the sacrifice piezoelectric activity. The higher the E_C , the lower d_{33} is. In our work, according to the though mentioned in last paragraph, an ultrahigh piezoelectric activity ($d_{33}\sim 2630$ pC/N, $d_{15}\sim 490$ pC/N) and a record enhancement of coercive field (8.2 kV/cm, over 3 times of that for the PMN-PT) are simultaneously achieved. This is an exceptional improvement in the ferroelectrics field, and the performance is far beyond the E_C vs. d_{33} boundary (red dash line) of other systems in Figure 1a.

(3) The approach. The “flat and deep” free energy landscape is achieved by introducing small amount of Sc^{3+} to form special microstructure: the highly dispersed heterogeneous structure with considerable content of tetragonal pinning centers. The dispersed local heterogeneous structure is formed due to the charge disorder in the B-site arises a strong random field, which makes it extremely difficult to establish ferroelectric long-range order (see detailed discussions in response to Reviewer #2, Page 16 of this letter). In addition, the 0.06PSN-0.61PMN-0.33PT is in the deep MPB composition with multiphase coexist (Figure 4a and b). The dispersed local heterogeneous structure together with multi-phase coexistence greatly decrease the P_S anisotropy, resulting in a flat free energy profile, which makes the P_S rotation much easier and contributes greatly to the high piezoelectric activity. The existence of considerable amount of tetragonal pinning centers is also caused by the introduction of PSN. The PSN favors large lattice deformation (Supplementary Note 1), and therefore strong random local stress is formed. The random stress is supposed to provoke the formation of highly distorted ferroelectric regions, such as tetragonal domains (see detailed discussions in Response to Reviewer #2, Page 17 of this letter). This tetragonal clusters disperses in the matrix, and strongly interact with neighboring polar clusters, acting as “pinning centers” in the ferroelectrics, resulting in difficult domain switch and enhanced E_C in the 0.06PSN–0.61PMN-0.33PT.

It is also worth noting that the “deep and flat” profile requires cautious manipulation of the microstructure. Only doping specific ions with optimum concentration could induce such microstructure (see detailed discussions in Response to Reviewer #2, Pages 13-14 of this letter). That is why the simultaneous enhancement of piezoelectric response and coercive field has not been observed in other

relaxor-PbTiO₃ systems including PMN-PT:Sm, PIN-PMN-PT, PFN-PYN-PT (Figure 1a), and the PSN-PMN-PT system with other compositions (Table R2).

The difference and relationship between our work and earlier papers by Li *et al*:

(1) Our research purpose is different from Li *et al*. They focus only on the piezoelectric response enhancement, and the coercive field E_C is out of consideration. In their works the high piezoelectric response embraced in the frame of E_C sacrifice, where they observed a low E_C ~2.5 kV/cm for both Sm-doped PMN-PT crystal and ceramic. In present work, we would like to solve a long-term puzzle: if there is a possibility to highly enhance the coercive field, and more importantly, without sacrificing their giant piezoelectricity.

(2) The mechanism of property enhancement is not same, although in the work by Li *et al*, the piezoelectricity enhancement is also achieved on the basis of free energy landscape manipulation. By Sm doping, the local polar regions with orthorhombic symmetry are embedded in the tetragonal matrix. Competition between the bulk and interfacial energy lead to a highly flattened free-energy profile, resulting in a highly responsive polarization and domain dynamics, so ultrahigh piezoelectric activity is obtained. However, this mechanism inevitably results in a low E_C . In our work, we make further progress in the free energy manipulation to form a “deep and flat” free-energy profile to achieve both easier P_S rotation and difficult domain reverse, which benefits both d_{33} and E_C .

(3) Our 0.06PSN-0.67PMN-0.33PT crystals demonstrate other advantages. The 0.06PSN-0.67PMN-0.33PT exhibits enhanced phase transition temperature, which guarantees improved thermal stability. In contrary, the introduction of Sm in Li *et al*'s work leads to significant reduction of phase transition temperature. Moreover, our work also verify that high-quality PSN-PMN-PT crystal is easy to grow in a large size scale in comparison with other new relaxor-PT ferroelectrics.

More strong evidence to support our conception:

To give more experimental evidence of the enhanced potential barrier, we experimentally determine activation electric field E_A required for domain switch, as shown in Figure R2. E_A of 0.06PSN-0.61PMN-0.33PT is almost 6 times higher than that of 0.67PMN-0.33PT, which is direct evidence for the deep potential barriers between adjunct P_S states.

Furthermore, the strong pinning effect of tetragonal nanoclusters are also confirmed by the

investigations of the domain dynamics using PFM technique, as shown in Figure R1. Figure R1 reveals that the domain-wall motions of 0.06PSN-0.61PMN-0.33PT is strongly inhibited by pinning centers, and domain nucleation is the dominant effect for dynamics. Moreover, the domain nucleation is also more difficult in the 0.06PSN-0.61PMN-0.33PT than in the 0.67PMN-0.33PT (see detailed discussions in Response to Reviewer #1, Page 2 of this letter).

As discussed above, we believe that the present work not only presents a remarkable and important property enhancement, but also provides a novel and original approach to design and optimize the overall performance of ferroelectrics. As such, this work could rekindle the confidence and research enthusiasm of relaxor-based ferroelectrics, and promote the development of next-generation piezoelectric ceramics.

Comments: The paper is not well written. The authors use word "convince" several times and the meaning is never correct. Here are some examples:

- considerable amount of tetragonal phase is convinced.
- further convincing the experimentally observed microstructure and functionality.
- are further convinced by Rayleigh analysis
- convinced the experimental design.

Reply: Thank the reviewer for his/her careful reading. The word "convince" has been changed as follows:

- a considerable amount of tetragonal phase is **confirmed**.
- a strong relaxor behavior as **confirmed** in current work
- are further **suggested** by Rayleigh analysis
- conducted** the experimental design.

Comments: There are other errors in style that distract from reading, some of which are given below. Also, I have few technical comments and questions:

Reply: Thanks for the valuable comments. Please see the following content for our detailed experimental results and discussion on this comment.

Authors write: "Both a large coercive field (E_C) and a giant piezoelectric activity are essential for the ferroelectric materials used in high-drive electromechanical applications, especially the case for greater power efficiency, smaller size, lighter weight, and lower overall cost ", What does cost has to do with E_C ?

The cost is unrelated to E_C . In the revised manuscript, we delete the sentence "and lower overall cost".

The word "respectively" is not used appropriately in Abstract.

In the revised manuscript, the word "respectively" is removed. And the sentence changes into "Consequently, both elevated energy barriers and flattened potential wells are thermodynamically achieved, affording a large E_C of 8.2 kV/cm (greater than that of $\text{Pb}(\text{Mg}_{1/3}\text{Nb}_{2/3})\text{O}_3\text{-PbTiO}_3$ by a factor of three) and ultrahigh piezoelectricity ($d_{33} = 2630$ pC/N; $d_{15} = 490$ pC/N)."

"hard doping of manganese (Mn) " ...should be "by" manganese

We have corrected this error in the revised manuscript.

"...without sacrificing, or even improving their giant piezoelectricity" is probably misstated

Thanks for pointing out this misstated. The sentence has been changed as "is there a possibility to highly enhance the coercive field without sacrificing their ultrahigh piezoelectricity?"

..."the latest piezoelectric activity over 4000 pC/N has been achieved" should probably be "...the recent studies achieved piezoelectric activity of over 4000 pC/N"

"...polarization switch behavior " should be "polarization switching behavior"

Thanks a lot. We have revised the sentences accordingly.

"It is of interest to note that the ternary crystals possess significantly higher local strain (~3%) as compared to that of their binary counterparts (1.5%), also suggesting a higher level of tetragonal lattice deformation c/a . "

This is in fact rather close. What is the error or standard deviation for local strain values?

We extract S_{xx} along 20 different horizontal lines in Figure 4d. The upper and lower boundaries of the yellow and blue regions in Figure R11 demonstrate the maximum and minimum strain values at each location, respectively. It is clear that 0.06PSN-0.61PMN-0.33PT crystal exhibits much higher local

strain than 0.67PMN-0.33PT. We calculated the standard deviation of local strain values, as shown by the red lines in Figure R12. The large standard deviation further confirming the high level of local strain in the ternary system.

Figure R12. The maximum, minimum and standard deviation of local strain S_{xx} derived from 20 different horizontal lines in Figure 4d.

"Based on the above dissection, " this should rather be "discussion".

The word has been changed accordingly.

"This strategy takes the advantages of both “hard” (large E_C) and “soft” (high piezoelectric response) doping, ..."

There is only one doping or modification. It's better to write that "...strategy accomplishes both hard ...and soft...properties".

The sentence has been revised according to the reviewer’s suggestions.

"After being poled by a DC E -field of 10 kV/cm ..." could authors specify crystallographic direction of the field during poling?

All the samples are $[001]_C$ orientate with $x//[100]_C$, $y//[010]_C$ and $z//[001]_C$, and the samples are poled along $[001]_C$ (the subscript C refers to the pseudocubic unit cells). We have made revisions accordingly in revised manuscript.

References:

1. Scott J F. Switching of ferroelectrics without domains. *Adv Mater* **22**, 5315-5317 (2010).
2. Rose Li Y, Halliwill K D, Adams C J, et al. Mutational signatures in tumours induced by high and low energy radiation in Trp53 deficient mice. *Nat Commun* **11**(1): 1-15 (2020).
3. Bakaul S R, Kim J, Hong S, et al. Ferroelectric Domain Wall Motion in Freestanding Single-Crystal Complex Oxide Thin Film. *Adv Mater* **32**(4): 1907036 (2020).
4. He W, Li Q, Sun Y, et al. Investigation of piezoelectric property and nanodomain structures for PIN–PZ–PMN–PT single crystals as a function of crystallographic orientation and temperature. *J Mater Chem C* **5**(9): 2459-2465 (2017).
5. McGilly L J, Sandu C S, Feigl L, et al. Nanoscale defect engineering and the resulting effects on domain wall dynamics in ferroelectric thin films. *Adv Funct Mater* **27**(15): 1605196 (2017).
6. Jiang A Q, Chen Z H, Hui W Y, et al. Subpicosecond domain switching in discrete regions of $\text{Pb}(\text{Zr}_{0.35}\text{Ti}_{0.65})\text{O}_3$ thick films. *Adv Funct Mater* **22**(10): 2148-2153 (2012).
7. Schultheiß J, Liu L, Kungl H, et al. Revealing the sequence of switching mechanisms in polycrystalline ferroelectric/ferroelastic materials. *Acta Mater* **157**: 355-363 (2018).
8. Li K, Sun E, Zhang Y, et al. High piezoelectricity of Eu^{3+} -doped $\text{Pb}(\text{Mg}_{1/3}\text{Nb}_{2/3})\text{O}_3$ - 0.25PbTiO_3 transparent ceramics. *J Mater Chem C* **9**(7): 2426-2436 (2021).
9. Scott J F. Models for the frequency dependence of coercive field and the size dependence of remanent polarization in ferroelectric thin films. *Integr Ferroelectr* **12**: 71-81 (1996).
10. Nomura Y, Tachi T, Kawae T, et al. Temperature dependence of ferroelectric properties and the activation energy of polarization reversal in (Pr, Mn)-codoped BiFeO_3 thin films. *Phys Status Solidi B* **252**: 833-838 (2015).
11. Li J, Li J, Qin S, et al. Effects of long-and short-range ferroelectric order on the electrocaloric effect in relaxor ferroelectric ceramics. *Phys Rev Appl* **11**: 044032 (2019).
12. Zuo R, Li F, Fu J, et al. Electric field forced c-axis oriented growth of polar nanoregions and rapid switching of tetragonal domains in BNT-PT-PMN ternary system. *J Eur Ceram Soc* **36**: 515-525

(2016).

13. Zhou X, Jiang C, Luo H, et al. Enhanced piezoresponse and electric field induced relaxor-ferroelectric phase transition in NBT-0.06BT ceramic prepared from hydrothermally synthesized nanoparticles. *Ceram Int* **42**(16): 18631-18640 (2016).
14. Zaman A, Hussain A, Malik R A, et al. Dielectric and electromechanical properties of LiNbO₃-modified (BiNa)TiO₃–(BaCa)TiO₃ lead-free piezoceramics. *J Phys D: Appl Phys* **49**(17): 175301 (2016).
15. Lin D, Li Z, Zhang S, et al. Electric-field and temperature induced phase transitions in Pb(Mg_{1/3}Nb_{2/3})O₃–0.3PbTiO₃ single crystals. *J Appl Phys* **108**(3): 034112 (2010).
16. Qi X, Sun E, Lü W, et al. Dynamic characteristics of defect dipoles in Mn-doped 0.24Pb(In_{1/2}Nb_{1/2})O₃–0.47Pb(Mg_{1/3}Nb_{2/3})O₃–0.29PbTiO₃ single crystal. *CrystEngComm* **21**(2): 348-355 (2019).
17. Wang L, Zhai Y, Zheng L, et al. Intrinsic piezoelectricity in (K, Na)NbO₃-based lead-free single crystal: Piezoelectric anisotropy and its evolution with temperature. *Appl Phys Lett* **117**(5): 052904 (2020).
18. Zheng L, Jing Y, Lu X, et al. Temperature dependent piezoelectric anisotropy in tetragonal 0.63Pb(Mg_{1/3}Nb_{2/3})-0.37PbTiO₃ single crystal. *Appl Phys Lett* **113**(10): 102903 (2018).
19. Guo Y, Xu H, Luo H, et al. Growth and electrical properties of Pb(Sc_{1/2}Nb_{1/2})O₃–Pb(Mg_{1/3}Nb_{2/3})O₃–PbTiO₃ ternary single crystals by a modified Bridgman technique. *J Cryst Growth* **226**(1): 111-116 (2001).
20. Wang Z, He C, Qiao H, et al. In situ di-, piezo-, ferroelectric properties and domain configurations of Pb(Sc_{1/2}Nb_{1/2})O₃–Pb(Mg_{1/3}Nb_{2/3})O₃–PbTiO₃ ferroelectric crystals. *Cryst Growth Des* **18**(1): 145-151 (2018).
21. Westphal VV, Kleemann W, Glinchuk MD. Diffuse phase transitions and random-field-induced domain states of the "relaxor" ferroelectric Pb(Mg_{1/3}Nb_{2/3})O₃. *Phys Rev Lett* **68**, 847-850 (1992)
22. Kleemann W. Relaxor ferroelectrics: Cluster glass ground state via random fields and random bonds. *Phys Status Solidi B* **251**, 1993-2002 (2014).

23. Shvartsman VV, Kleemann W, Łukasiewicz T, Dec J. Nanopolar structure in $\text{Sr}_x\text{Ba}_{1-x}\text{Nb}_2\text{O}_6$ single crystals tuned by Sr/Ba ratio and investigated by piezoelectric force microscopy. *Phys Rev B* **77**, 054105 (2008).
24. Tinte S, Burton B P, Cockayne E, et al. Origin of the relaxor state in $\text{Pb}(\text{B}_x\text{B}_{1-x})\text{O}_3$ perovskites. *Phys Rev Lett* **97**(13): 137601 (2006).

We thank editor and reviewers again for the revision suggestions and all above useful questions and comments. Hopefully the manuscript has been improved by taking into account all above comments and addressing all above questions.

Sincerely yours,

Gang Liu

Staff Scientist, Center for High Pressure Science and Technology Advanced Research

REVIEWERS' COMMENTS

Reviewer #1 (Remarks to the Author):

I am happy with the revised version and the authors' response. My suggestion is to accept the paper.

Reviewer #2 (Remarks to the Author):

The authors have carefully taken account all of my preceding recommendations, added additional clarifying text and as a result submitted a largely improved manuscript. Practically all recommendations have been taken into account and were recomposed in the new manuscripts of the Word files attached below - "NCOMMS-21-26075B_article revised" and "NCOMMS-21-26075B_supp revised". New text has been inserted in RED color. Only few minor deficiencies have remained left, which I corrected in BLUE color.

I deem the manuscript now acceptable as is.

Reviewer #3 (Remarks to the Author):

The authors have answered my comments and have made appropriate changes in the manuscript. The authors show large E_c and d_{33} , but their interpretation of those properties in terms of a flat and deep energy barrier for polarization switching and rotation remains hand-waving. The arguments given by the authors are obvious (a large d_{33} in a nonpolar direction does mean that energy for polarization rotation is flat, and large E_c and activation energy for domain switching are trivially related, as have been shown by many authors). However, this trivial correlation does not tell us anything about the atomic and microstructural mechanisms. This is fine, I understand that interpretation is difficult. What I object is that authors write their paper as though the proposed "explanation" is the definite interpretation, whereas this interpretation (via tetragonal regions rich with Sc) is only plausible, but still hand-waving. The sentence in Abstract "...dispersed local heterogeneity comprises abundant tetragonal phases. Consequently, both elevated energy barriers and flattened potential wells are thermodynamically achieved,..." is a pure conjecture. The paper does describe a material that has an interesting application potential, but that's all.

Supplementary Information

Simultaneously achieving giant piezoelectricity and record coercive field enhancement in relaxor-based ferroelectric crystals

Liya Yang,^{#1,2,3} Houbing Huang,^{#4} Zengzhe Xi,⁵ Limei Zheng,^{1,*} Shiqi Xu,⁴ Gang Tian,¹ Yuzhi Zhai,¹ Feifei Guo,⁵ Lingping Kong,⁶ Yonggang Wang,⁶ Weiming Lü,^{7,*} Long Yuan,⁸ Minglei Zhao,¹ Haiwu Zheng,² and Gang Liu^{6,*}

¹School of Physics, Shandong University, Jinan 250100, China

²International Joint Research Laboratory of New Energy Materials and Devices of Henan Province, School of Physics and Electronics, Henan University, Kaifeng 475004, China

³Condensed Matter Science and Technology Institute, School of Instrumentation Science and Engineering, Harbin Institute of Technology, Harbin 150080, China

⁴School of Materials Science and Engineering & Advanced Research Institute of Multidisciplinary Science, Beijing Institute of Technology, Beijing 100081, China

⁵School of Materials and Chemical Engineering, Xi'an Technological University, Xi'an 710032, China

⁶Center for High Pressure Science and Technology Advanced Research, Shanghai 201203, China

⁷Spintronics Institute, School of Physics and Technology, University of Jinan, Jinan 250022, China

⁸Key Laboratory of Functional Materials Physics and Chemistry of the Ministry of Education, Jilin Normal University, Changchun 130103, China

*Corresponding authors:

zhenglm@sdu.edu.cn (L.Z.);

sdy.lvwm@ujn.edu.cn (W.L.);

liugang@hpstar.ac.cn (G.L.)

Note 1. Lattice anisotropy of $\text{Pb}(\text{Sc}_{1/2}\text{Nb}_{1/2})\text{O}_3$ system

The lattice parameters of $\text{Pb}(\text{Sc}_{1/2}\text{Nb}_{1/2})\text{O}_3$ (PSN) and $\text{Pb}(\text{Mg}_{1/3}\text{Nb}_{2/3})\text{O}_3$ (PMN) are listed in Table S1, where the shear lattice deformation is estimated by $(90^\circ-\alpha)/2$. Both materials are rhombohedral (R) structures at room temperature. In comparison with PMN (ICSD #161663), PSN (ICSD #90501) has larger lattice parameters, and exhibits ~3 times larger of the shear deformation. Thus, the introduction of PSN into Pb-based ferroelectrics results in a high level of lattice anisotropy. For instance, the lattice deformation (estimated by $c/a-1$) of PSN-0.42PT are ~5 times higher than that of PMN-0.32PT (both compounds are with MPB composition). As such, we may consider introduce PSN into the PMN-PT system to establish local structure heterogeneity with strong tetragonality and large lattice deformation $c/a-1$.

Table S1. Lattice parameters and lattice deformation of various PSN- and PMN-based perovskites.

	Phase	Lattice parameter	Lattice deformation	Deformation Ratio ^a
PSN	Rhombohedral	$a=b=c=4.082 \text{ \AA}$ $\alpha=\beta=\gamma=89.914^\circ$	$(90^\circ-\alpha)/2=0.043^\circ$	3
PMN	Rhombohedral	$a=b=c=4.045 \text{ \AA}$ $\alpha=\beta=\gamma=89.971^\circ$	$(90^\circ-\alpha)/2=0.015^\circ$	
PSN-0.42PT ¹	Monoclinic Pm	$a=4.036 \text{ \AA}$ $b=3.987 \text{ \AA}$ $c=4.091 \text{ \AA}$ $\beta=90.190^\circ$	$c/a-1=1.36\%$	6
PMN-0.32PT ²	Monoclinic Pm	$a=4.018 \text{ \AA}$ $b=4.005 \text{ \AA}$ $c=4.028 \text{ \AA}$ $\beta=90.146^\circ$	$c/a-1=0.23\%$	

^a Deformation ratio is the ratio of lattice deformation of PSN system to PMN system. For the pure PSN and PMN, shear deformation is adopted; for the binary PSN-PT and PMN-PT, normal lattice deformation is considered.

Note 2. Repeatability and frequency dependence of coercive field

To verify the repeatability of the extraordinary large coercive field E_C , we measured P - E hysteresis loops of three different $[001]_C$ oriented 0.06PSN-0.61PMN-0.33PT crystals. All samples exhibit similar E_C values, ~8.0 kV/cm (Figure S1), indicating that the large coercive field is an intrinsic behavior rather than accidental phenomena.

Figure S1. Repeatability of the extraordinary large coercive field convinced in three different 0.06PSN-0.61PMN-0.33PT samples, showing E_C values of 8.1, 8.2, and 7.8 kV/cm, respectively.

Figure S2a shows the P-E hysteresis loops of 0.06PSN-0.61PMN-0.33PT at frequencies $0.1 \leq f \leq 100$ Hz. The frequency dependence of E_C was summarized in Figure S2b, from which one can see E_C increases from 7.5 kV/cm to 11.8 kV/cm as f increases from 0.1 Hz to 100 Hz. The frequency dependent E_C can be described by the theoretical model from Ishibashi and Orihara^{3,4}

$$E_C = Kf^\beta . \quad (S1)$$

And our experimental data in Figure S2b can be well described by Equation (S1) with derived parameters of $K=2.51$ and $\beta=0.07$. According to relationship, the coercive field at other frequencies can be estimated.

Figure S2. Frequency dependence of P-E hysteresis loops (a) and E_C (b) for 0.06PSN-0.61PMN-0.33PT single crystal. Fitting the experimental data by Eq. S1 gives $E_C=2.51f^{0.07}$.

Note 3. Current vs. electric field (I - E) loops

Figure S3a shows the I - E loops of the crystal sample (used in Figure S2) measured at different frequencies. For each frequency, a current peak can be observed around E_C , corresponding to the displacement current originated from domain switching. The maximum displacement current increases with frequency for the shortened switching time. Figure S3b summarizes I - E loops of 4 different samples measured at a fixed frequency of 1 Hz (Samples 1-3 correspond to the 3 samples in Supplementary Figure S1). For all the samples, the conductive current is negligible in comparison with the displacement current. The enlarged I - E curves are provided as the inset of Figure S3b. Only Sample 3 demonstrates a detectable leakage current while other samples show excellent insulation effect.

Note 4. Temperature dependence of material properties of PSN-PMN-PT

Figure S4 shows the normalized temperature dependence of d_{33} , d_{15} , k_{33} and k_{15} of 0.06PSN-0.61PMN-0.33PT single crystals below T_{F-F} . k_{33} and k_{15} is almost temperature independent with variation below 3% and 13%, respectively, manifesting excellent thermal stability. Shear piezoelectric coefficient d_{15} increases slightly by 30%, while d_{33} increases dramatically by 140% as approaching phase transition temperature T_{F-F} .

Figure S3. I - E loops of the 0.06PSN-0.61PMN-0.33PT single crystals. (a) Frequency dependence of I - E loops for Sample 4. (b) I - E loops of 4 different samples measured at 1 Hz.

Figure S4. Normalized d_{33} , d_{15} , k_{33} and k_{15} as a functions of temperature.

Figure S5a shows the temperature dependent P - E hysteresis loops for 0.06PSN-0.61PMN-0.33PT single crystal. The maximum polarization P_{\max} , remnant polarization P_r , and the coercive field E_C as a function of temperature are summarized in Figure S5b. E_C maintains a high value above 6 kV/cm till temperature increases to T_{F-F} . Below 150 °C, well-saturated P - E loops can be obtained, from which one can see P_{\max} and P_r decrease gradually with temperature. As approaching Curie temperature, the P - E loop shrinks obviously and P_r reduces dramatically. High polarization values with $P_{\max}=12 \mu\text{C}/\text{cm}^2$ and $P_r=3.5 \mu\text{C}/\text{cm}^2$ can even be achieved at 210°C, 50°C above Curie temperature T_C .

Figure S5. Temperature dependence of ferroelectric properties for 0.06PSN-0.61PMN-0.33PT single crystal. (a) P - E hysteresis loops and (b) remnant polarization P_r , maximum polarization P_{\max} and E_C as a function of temperature. All the loops are measured at 1 kHz. E_C above 6.2 kV/cm is maintained below T_{F-F} .

Note 5. Domain switching and local piezoelectric response of PSN-PMN-PT

We carried out piezoelectric force microscopy (PFM) to *in situ* map the ferroelectric domain patterns and investigate the local piezoelectric response. We employed a gradually increasing tip-voltage V_{dc} from 0 to 10 V on the crystal surface, enabling us to pole a local area of $3 \times 3 \mu\text{m}^2$ in a smooth manner. Figure S6a shows the evolutions of the out-of-plane PFM amplitude and phase images, from which the amplitude and phase response along the white dash line were derived and quantitatively analyzed (Figure S6b). The 180° phase contrast between the recording area measured at 5 V and 8 V reveals the complete polarization reversal of the ferroelectric domains, which is further supported by switching spectroscopy PFM (SS-PFM) characterization also displaying a 180° phase contrast (Figure S6c). According to the fact that the enhanced domain switching and mobility contribute significantly to the giant piezoelectric response, such a fully polarization switch is critically necessary for high performance of ferroelectrics, especially for materials with a large E_C . As shown in Figure S6d, the PSN-PMN-0.33PT crystal features a well-defined local piezoelectric response loop, presenting direct evidence for a superior piezoelectric response from a microscopic perspective. Since the vertical vibration signal is directly related to the piezoelectric response, the macroscopic piezoelectric property can be viewed as a collective effect of the microscopic piezoelectric response.

Figure S6. Domain structures and local domain dynamics measured by PFM. (a) Evolution of vertical PFM amplitude and phase structures under different tip voltage V_p . The amplitude and phase response along the white dash are extracted and shown in (b). (c) Local amplitude and phase and (d) piezoresponse as a function of tip voltage measured by SS-PFM. The local piezoresponse is calculated by $PR=A \times \cos(\varphi)$, where PR , A , and φ indicate local piezoresponse, amplitude, and phase angle, respectively.

Figures S7a-l show the domain evolution with temperature for 0.06PSN-0.61PMN-0.33PT single crystals. The homogeneous labyrinth-like nanodomain structures show slight changes below T_{F-F} . Above T_{F-F} , however, the labyrinth domains expand dramatically with temperature, and the stripe-like domains with domain walls along $[010]_C$ direction appears, demonstrating that the crystal changes from MPB to T phase. As temperature further approaches T_C , both amplitude and phase contrast between different nano-domains attenuated, corresponding to degradation of ferroelectric domain structure.

Figure S7. Domain evolution with temperature for 0.06PSN-0.61PMN-0.33PT single crystals. (a-l) The PFM amplitude and phase images at various temperatures.

Note 6. Autocorrelation function of the polar regions.

Figure S8a shows the PFM images of $[001]_c$ oriented 0.06PSN-0.61PMN-0.33PT and 0.67PMN-0.33PT crystals. Ternary 0.06PSN-0.61PMN-0.31PT consists of smaller domains than the binary system (~ 200 nm for PSN-PMN-PT *vs.* ~ 1300 nm for PMN-PT). We further quantify the agglomerates of the local polar regions by an averaged autocorrelation function over all in-plane directions as:⁵⁻⁷

$$\langle C(r) \rangle = \sigma^2 \exp \left[- \left(\frac{r}{\xi} \right)^{2h} \right]. \quad (1)$$

Here σ is a constant and ξ is short-range correlation length that indicates the degree of polarization correlation, and h ($0 < h < 1$) describes the roughness of the polarization interface, and the fractal dimension of an interface is $d = 3 - h$. The best fitting results are shown in Figure S8b. $\xi = 70$ nm is obtained in 0.06PSN-0.61PMN-0.33PT crystal, much smaller than those from analysis on 0.67PMN-0.33PT ($\xi = 450$ nm), demonstrating it is much more difficult for the ternary crystal to establish homogeneous polarization order but only short-range order between neighboring clusters, echoing to the stronger relaxor behavior shown in Figures 3b and 3c.

Figure S8. (a) Amplitude, phase and autocorrelation images and (b) the averaged autocorrelation function for the two crystals and the best fitting of the experimental points.

Figures S9a and b shows the temperature dependent autocorrelation analysis, and in Figure S9c and d we extracted the temperature dependent correlation radius ξ and fractal dimension d , respectively. Initially, both ξ and d change slightly with temperature, demonstrating a stable domain structure and random fields with temperature. As approaching freezing temperature T_{VF} , ξ increases by 2 times (from 85 nm to 171 nm) and d decreases dramatically, indicating that the local structure becomes more ordered. Above T_{VF} , some of the static PNRs transform into dynamic ones, the system becomes more disordered, resulting in the decrease of ξ and increase of d . Nevertheless, at 180 °C, 25 °C above T_m , ξ (~30 nm) is still not negligible, denoting the existence of quasi-static PNRs. With temperature further increasing above 180 °C, the quasi-static PNRs are not detectable, only dynamic PNRs exist.

Note 7. Temperature dependence of dielectric constant for 0.67PMN-0.33PT single crystal.

Figure S10 shows the temperature dependent dielectric constant around T_m for the unpoled 0.67PMN-0.33PT single crystals, from which Vogel-Fulcher relationship can be fitted. In addition, we observed a 2.5 K of T_m shift, which is much smaller than that of ternary crystals with same PT content.

Figure S9. Temperature dependent autocorrelation analysis. (a) The autocorrelation images at various temperatures. (b) The temperature evolution of the averaged autocorrelation function and fitting results. (c) and (d) average autocorrelation length ξ and fractional dimension d as a function of temperature, respectively.

Figure S10. High-temperature dielectric property for unpoled 0.67PMN-0.33PT single crystal.

Note 8. Local structure heterogeneity.

Figures S11a-c and S11e-g represent the selected area electron diffraction (SAED) patterns along $[001]_c$, $[1\bar{2}0]_c$, and $[1\bar{1}\bar{3}]_c$ axis for 0.67PMN-0.33PT and 0.06PSN-0.61PMN-0.33PT, respectively. These two crystals exhibit similar crystalline structure. Figures S11d and S11h are the dark-field TEM images of the two crystals, respectively, where the long range order ferroelectric domain structures cannot be observed, in accordance with the results observed by PFM (Figure S8). Moreover, the PSN-PMN-PT system exhibits a more complicated local structure heterogeneity.

Figure S11. Selected area electron diffraction (SAED) patterns of 0.06PSN-0.61PMN-0.33PT and 0.67PMN-0.33PT single crystals. (a-c) are the images along $[001]_c$, $[1\bar{2}0]_c$, and $[1\bar{1}\bar{3}]_c$ axis for PMN-PT, and (e-g) are those for PSN-PMN-PT. (d) and (h) is the local microstructures observed by HRTEM.

Figure S12 is the HAADF-STEM image of 0.67PMN-0.33PT crystal, from which the polarization vector P_S of each unit cell column was determined. Arising from its MPB composition, 0.67PMN-0.33PT also exhibits a multiphase coexistence, *e.g.*, O, T, and M phases, yet the polar regions are larger than that in 0.06PSN-0.61PMN-0.33PT.

Figure S12. HAADF-STEM image of $[001]_C$ oriented 0.67PMN-0.33PT single crystal, the P_S directions are given for each unit-cell column and the possible phase structure have been labeled.

Note 9. Phase structure and lattice parameters determined by high-resolution XRD.

High-resolution XRD pattern of 0.67PMN-0.33PT is shown in Figure S13, which demonstrates multiphase coexistence: monoclinic M_A (space group Cm), monoclinic M_C (space group Pm) and tetragonal (space group $P4mm$) phases, consisting with the STEM results (Figure S8). Details phase constitute of 0.06PSN-0.61PMN-0.33PT and 0.67PMN-0.33PT are listed in Table S2. The ternary system exhibits a larger lattice parameter than the binary system due to the larger ionic radius of Sc^{3+} (0.745 Å) than Ti^{4+} (0.605 Å) and Nb^{5+} (0.640 Å). Moreover, the 0.06PSN-0.61PMN-0.33PT contains more tetragonal component than 0.67PMN-0.33PT.

Figure S13. High-resolution XRD pattern and Rietveld refinement of PMN-0.33PT.

Table S2. Phase constitute and lattice parameters for the two crystals basis by high resolution XRD.

Phase	Space Group	0.06PSN-0.61PMN-0.33PT				0.67PMN-P0.33T			
		a (Å)	b (Å)	c (Å)	volume fraction	a (Å)	b (Å)	c (Å)	volume fraction
M_A	Cm	5.715	5.704	4.029	43.2%	5.700	5.681	4.010	55.8%
M_C	Pm	4.021	4.010	4.032	22.3%	4.017	4.005	4.029	30.5%
T	$P4mm$	4.011	4.011	4.051	34.5%	4.015	4.015	4.044	13.7%

Note 10. Electric field- and temperature-dependent structural evolutions

In Figure S14 we compare the dielectric permittivity vs. temperature curves between poled and unpoled 0.06PSN-0.61PMN-0.33PT samples. It is well accepted that during poling process, PNRs transformed into macro ferroelectric domains⁸⁻¹¹, thus after poling the sample has macro ferroelectric domains at room temperature. The dielectric anomaly at $T_{F-F}=85$ °C of the poled samples corresponds to the temperature-driven ferroelectric to ferroelectric phase transition, along with the partial depoling of the $[001]_C$ poled domain structures. For the unpoled sample, however, the PNR state is always maintained, and no apparent phase structure change occurs below T_m (for details please see the data

shown in Figure S16c), thus the dielectric anomaly corresponding to the ferroelectric-ferroelectric phase transition is not detectable. Other relaxor ferroelectrics also demonstrate similar dielectric behavior¹².

Figure S14. Comparison of dielectric permittivity vs. temperature curves of poled and unpoled 0.06PSN-0.61PMN-0.33PT single crystals.

For $[001]_c$ oriented crystal, power XRD characterizations only give information of $(00m)$ ($m=1, 2, 3\dots$) Bragg peaks, thus it is difficult to determine the phase component and lattice parameters using a single crystal sample. Alternatively, we employed polycrystalline ceramics with the same composition as crystal for a detailed temperature-dependent structural study. The main difference between single crystal and ceramics arises from extrinsic factors such as grain boundary, while XRD patterns generally reflect the distortion of the lattice structure, which is an intrinsic factor. Although there may exist a slight difference in structures between single crystal and ceramics, the influence of electric field and temperature should be dominated factor for intrinsic structures here. Thus we believe the results based on ceramics can provide us with quotable information.

Figure S15 demonstrates the XRD patterns of the poled and unpoled samples. At $2\theta \sim 45^\circ$, the high-angle peak is strengthened after poling, indicating that the domains switch to directions close to the E -field. The narrower diffraction peaks in the poled sample demonstrate more uniform domain structure.

The phase concentrations and lattice parameters for each phase are listed in Table S3. After poling, the component of M_C phase and tetragonal (T) phase increase while the M_A phase decreases.

Figure S15. XRD patterns for the poled and unpoled samples.

Table S3. Phase component and lattice parameters for the poled and unpoled samples.

PSN-PMN-0.32PT	Unpoled				Poled			
	a (Å)	b (Å)	c (Å)	volume fraction	a (Å)	b (Å)	c (Å)	volume fraction
M_A	5.732	5.687	4.009	43.9%	5.719	5.667	4.000	17.5%
M_C	4.065	3.973	4.032	22.7%	4.037	3.993	4.054	44.3%
T	4.001	4.001	4.053	33.4%	4.018	4.018	4.046	38.2%

The XRD patterns as a function of temperature for unpoled and poled samples are shown in Figures S16a and S16b, respectively. The phase fractions are also determined according to our refinements, as shown in Figures S16c and S16d, respectively. With the increase of temperature, M_C phase fraction gradually increases. For the poled samples, the phase fraction changes obviously around T_{F-F} , corresponding well with remarkable dielectric anomaly in Figure S14. The cubic phase begins to appear at around 140 °C, and becomes the dominant phase as the temperature increases.

Figure S16. Temperature dependent XRD patterns and calculated phase fractions for (a) and (c) unpoled and (b) and (d) poled samples.

Then we discuss the variations of piezoelectric activity with temperature from the aspects of phase structure changes. The total piezoelectric response is the integral effect from all ferroelectric phases: T, M_A and M_C . The tetragonal phase is associated with low d_{33} along $[001]_C$ ^{13,14}, and high d_{33} in the 0.06PSN-0.61PMN-0.33PT origins mainly from M_A and M_C phases where P_S rotation contributes greatly. The total volume fraction of M_A+M_C increases with temperature (Figure S17a). In addition, the lattice distortion strengthens with temperature, as demonstrated by the increased β (the angle between lattice a and c) value (Figure S17b). The larger β corresponds to a stronger deviation of P_S vector from $[001]_C$, which could be more sensitive to the external electric field and facilitates piezoelectricity. The

enhanced monoclinic phase fraction, the increased lattice distortion, together with the softened crystal lattice at high temperature give a monotonously enhanced d_{33} with temperature (Figure S4).

Figure S17. Temperature dependent (a) monoclinic phase fraction and (b) β of M_A and M_C .

Note 11. Phase-field simulations.

In the simulations, we doped tetragonal PMN-0.42PT PNRs into high piezoelectric PMN-0.30PT matrix to theoretically reproduce the situations in PSN-PMN-PT system. Here we chose PMN-0.3PT with pure rhombohedral phase as the parent matrix for simplicity, although differs slightly from the experimental composition, the simulations can still provide guidelines that greatly help to understand the experimental results. The ferroelectrics ceramic of PMN-PT is discretized at grid size is $128\Delta x \times 128\Delta x \times 16\Delta x$, with stress-free boundary conditions, Δx is set to 1 nm. For PMN-0.3PT and PMN-0.42PT, the material coefficients and shown in Table S3.¹⁵ Figure S18a shows the calculated piezoelectric constant d_{33} of PMN-PT with and without tetragonal component doping. d_{33} of pure PMN-PT is 729 pC/N, enhancing to 1641 pC/N by introducing tetragonal component. Besides, the strong piezoelectric in the doped PMN-PT is highly homogenous (Figure S18c). The P - E hysteresis loops of these two systems were also calculated (Figure S18b), from which we observe a significant enhancement magnitude of E_C due to the appropriate doping of tetragonal component.

Table S4. Materials coefficients for PMN-0.3PT and PMN-0.42PT used in this work.

Coefficient	PMN-0.3PT	PMN-0.42PT
$\alpha_1 (10^5 \text{ C}^{-2} \text{ m}^2 \text{ N})$	$2.295 \times T - 935.9$	$2.583 \times T - 1204$
$\alpha_{11} (10^5 \text{ C}^{-4} \text{ m}^6 \text{ N})$	$-0.3775 \times T + 457.7$	$-0.3775 \times T + 304.2$
$\alpha_{12} (10^7 \text{ C}^{-4} \text{ m}^6 \text{ N})$	6.075	10.85
$\alpha_{111} (10^9 \text{ C}^{-6} \text{ m}^{10} \text{ N})$	2.57	2.57
$\alpha_{112} (10^9 \text{ C}^{-6} \text{ m}^{10} \text{ N})$	6.95	6.95
$\alpha_{123} (10^9 \text{ C}^{-6} \text{ m}^{10} \text{ N})$	13.13	13.13
$S_{11} (10^{-12} \text{ m}^2/\text{N})$	52	9.43
$S_{12} (10^{-12} \text{ m}^2/\text{N})$	-18.9	-1.68
$S_{44} (10^{-12} \text{ m}^2/\text{N})$	14	35.09
$Q_{11} (\text{m}^4/\text{C}^2)$	0.084	0.084
$Q_{12} (\text{m}^4/\text{C}^2)$	-0.025	-0.025
$Q_{44} (\text{m}^4/\text{C}^2)$	0.035	0.035

Figure S18. Phase-field simulations of (a) Piezoelectric constants d_{33} , (b) P - E loops and (c) distribution of d_{33} for pure and doped PMN-PT.

Note 12. Highly disordered structure and random fields

The relaxor-PbTiO₃ ferroelectrics are charge-disordered relaxors. The inherent charge disorder is due to the random distribution of various B-site ions such as Mg²⁺, Nb⁵⁺ and Ti⁴⁺. It offers uncorrelated and quenched random electric fields at the sites of the ferroelectric-active ions. These random fields favor the formation of PNRs. The dynamic PNRs begin to appear at Burn temperature T_B . The interaction between PNRs and quenched random electric fields results in a local mesoscale phase transition at T^* , below which the static PNR appears, coexisting with dynamic ones. On further cooling, the slowdown of PNR dynamics result in a remarkable dielectric relaxation. Finally, below freezing temperature, T_{VF} , the PNR dynamics ultimately slow down into a totally static glassy-like state^{6,16-18}.

The introduction of a small amount of trivalent Sc³⁺ ions make the B-site ions more disordered than the PMN-PT system, thus a stronger random field formed. As a result, the dynamic PNRs appear at a higher temperature, as verified by the experimental results that the T_B of 0.06PSN-0.61PMN-0.33PT is 40 °C higher than the 0.67PMN-0.33PT (Figure 3a). The highly disordered state and strong random field can be further confirmed by the higher diffused factor γ (Figure 3b), the lower freezing temperature T_{VF} (Figure 3f), the larger T_m shift (Figure 3c and Figure S10), the smaller domain size and the shorter correlation length ζ (Figure S8).

It should be noted that to obtain highly disordered state, the amount of doped Sc³⁺ should be small. If the system contains a large amount of PSN, the chemically ordered regions with regular Sc³⁺-Nb⁵⁺ ordering could be easily established, and a weak but homogeneous local field is formed at the chemically ordered regions, giving raise to preferential local order ferroelectric domain¹⁶.

Note 13. Domain switching

PFM technique was employed to explore the domain dynamic behavior. An area of 6×6 μm² is pre-poled by a tip voltage of -20 V to form an upward region. Then nanoscale domains in this region were reversed downward with different positive tip voltages and pulse durations. Figure S19 shows the out-of-plane phase image of the domains. For the 0.67PMN-0.33PT single crystal, only one nucleated domain grows till to complete a domain switching. The domain diameter increases with the increase of both tip voltage and pulse duration, and the domain growth is the predominant effect. These results are consistent with previous results on some other ferroelectrics crystals and films¹⁹⁻²¹. However, for 0.06PSN-0.61PMN-0.33PT, the domain switching is expedited by an increased amount of nucleation

sites. This is corroborated by the irregular shape of the switched area arising from many individually nucleated domains, which may coalesce during the growth. The multi-site domain nucleation phenomena has been observed in ferroelectric Pb(Zr,Ti)O₃ film, which contains a large amount of defect pinning centers^{22,23}. As such, it is reasonably deduced that the domain growth in 0.06PSN-0.61PMN-0.33PT is severely inhibited by the high concentration of pinning centers, *e.g.*, tetragonal nanoclusters, and domain nucleation is the dominant effect for domain dynamics.

In addition, domain nucleation in the 0.06PSN-0.61PMN-0.33PT is also more difficult than in the 0.67PMN-0.33PT. The domains in the yellow rectangle are fabricated at different pulse durations with a fixed tip voltage of 7 V. As can be seen, domain nucleation occurs with a smaller pulse duration in the binary system than in the ternary one. On the other hand, with a fixed pulse duration of 3 s, the domain nucleation also happens at a lower tip voltage (domains in the red rectangle) for the 0.67PMN-0.33PT. These results further support the observed higher E_C in ternary crystals.

Figure S19. Domain switching behavior under different tip voltage and pulse duration.

We then consider the dynamics of domain switching. The activation electric field E_A required for domain switch is determined by Merz's law. Figure S2a shows the P - E hysteresis loops measured at different frequencies f . The frequency dependent E_C can be depicted by Merz's law²⁴⁻²⁶:

$$\tau \propto 1/f \propto \exp(E_A/E_C), \quad (\text{S2})$$

where τ is the switching time. Figure S20 gives the fitting results of the experimental data. A larger value of $E_A=138$ kV/cm was derived for 0.06PSN-0.61PMN-0.33PT, in comparison of 23 kV/cm for 0.67PMN-0.33PT. The higher E_A of ternary system means a more difficult domain reverse process, further verifying the proposed deep potential barrier ΔG (Figure 5).

Figure S20. $1/E_c$ vs. $\ln f$ of 0.06PSN-0.61PMN-0.33PT and 0.67PMN-0.33PT crystals. The solid lines are the fitting results of Merz's law.

Note 14. Longitudinal and shear piezoelectric response.

Longitudinal piezoelectric coefficient d_{33} and shear coefficient d_{15} correspond to different kinds of lattice deformation in response to electric stimulate. For the 33-mode, the measured electric field is applied along $[001]_C$, leading to a longitudinal lattice deformation through electromechanical coupling (Figure S21a), whereas for the 15-mode, shear lattice deformation occurs under external field along $[100]_C$ (Figure S21b). Figure S22 demonstrates the piezoelectric lattice deformation of d_{33} and d_{15} modes for various engineered domain configurations. Generally, the 4R and 4O domain structures exhibit large longitudinal piezoelectric coefficient d_{33} while the 1T domain structure shows high shear response d_{15} for the easier polarization rotation.²⁷

Figure S21. Diagrams of (a) longitudinal and (b) shear lattice deformation for $[001]_c$ poled samples. The vectors P demonstrate total polarization along $[001]_c$ rather than the spontaneous polarization.

Figure S22. Piezoelectric deformation of crystal lattice with various engineered domain configurations: (a) d_{33} and (b) d_{15} mode of 4R ($[001]_c$ poled rhombohedral phase), 4O ($[001]_c$ poled orthorhombic phase), and 1T ($[001]_c$ poled tetragonal phase). The vectors P demonstrate spontaneous polarizations for each domain structures.

Note 15. Property comparison among various PSN-PMN-PT crystals

In Table S5 we listed the main performance of our 0.06PSN-0.61PMN-0.33PT single crystal in comparison with previous work.^{28,29} One can clearly see that our work demonstrates superior properties: d_{33} ~2630 pC/N is 2 times higher than those in Work 1 (~1200 pC/N) and Work 2 (1260-1550 pC/N), and meanwhile E_C ~8.2 kV/cm is much larger than those in Works 1 and 2 (4-6 kV/cm). In addition, we observed that the electromechanical coupling factor (k_{33} ~0.9) and dielectric constant (ϵ_{33} ~5950) of our crystals are also superior. The PSN-PMN-PT crystals in Works 1 and 2 demonstrate relatively low d_{33} and moderate E_C , comparable to the PIN-PMN-PT:Mn and PSN-PT systems, at an average level among all the data shown in Figure 1a. Similar to other works, the enhancement of E_C in Works 1 and 2 is at the expense of piezoelectricity d_{33} . Alternatively, our crystal achieves simultaneous ultrahigh piezoelectricity and extremely large coercive field, far beyond the boundary (see the red dash line in Figure 1a) of other systems, getting access the previously “no data” region.

Table S5. Comparison of the main performance for PSN-PMN-PT system.

	This work	Work 1 (ref. 28)	Work 2 (ref. 29)
Composition	0.06PSN-0.61PMN-0.33PT	0.05PSN-0.63PMN-0.32PT	yPSN-z0.63PMN-xPT $x=0.12-0.13, z=0.52-0.48;$ $x=0.26-28, z=048-0.435;$ $x=0.3675-0.3975, z=0.51-0.47$
Phase structure	MPB	R	R, MPB, T
d_{33} (pC/N)	2630	1200	1260-1550
E_C (kV/cm)	8.2	4-6 ^b	4-6
ϵ_{33} (ϵ_0)	5950	3500	1700-2000
k_{33}	0.9	-	~ 0.75
γ	1.96	1.73 ^a	1.65-1.82 ^a
T_{F-F} (°C)	85	70	120-180
T_C (°C)	152	162	200-240
Domain size (nm)	~200	-	~ 1000

^a The diffuseness factor γ , which was originally lacking in this manuscript, has been calculated from the dielectric permittivity vs. temperature curves, as shown in Figure S23; ^b The E_C value is not provided in ref. 28. Generally, the ferroelectrics is poled by a field exceeding $2E_C$. Considering that the poling electric field is 10 kV/cm, E_C is estimated to be 4-6 kV/cm.

The diffuseness factor γ for the sample in Work 1 is 1.72, and 1.65-1.82 in Work 2 (Table S5 and Figure S23), much lower than the value in our work ($\gamma=1.94$). In addition, the crystals in Work 2 demonstrate a much larger domain size. Both small γ value and large domain size verify less dispersed micro polar structure, therefore gives low piezoelectric properties.

Figure S23. Diffuse factor γ for the samples studied in Works 1 and 2.

Note 16. Domain wall mobility and lattice deformation under external stimuli

Rayleigh analysis was performed to estimate the domain wall mobility under the external E field.^{30,31}

$$\epsilon_{33}(E_0) = \epsilon_{rev} + \alpha \cdot E_0 \quad (2)$$

$$P(E) = \epsilon_0 [(\epsilon_{rev} + \alpha \cdot E_0)E \pm \alpha(E_0^2 - E^2)/2]. \quad (3)$$

Where $\varepsilon_{33}(E_0)$ is the dielectric permittivity under an AC electric field with amplitude E_0 , αE_0 describes irreversible dielectric contributions from domain wall motions, and α is the Rayleigh parameter. ε_{rev} is the reversible dielectric response, which mainly comes from lattice deformation. Figure S24 shows the Rayleigh behavior of $[001]_C$ poled 0.06PSN-0.61PMN-0.33PT and 0.67PMN-0.33PT crystals. α of PSN-PMN-0.33PT is 3185 cm/kV, 3 times higher than that for 0.67PMN-0.33PT (902 cm/kV), corresponding to a high level of domain wall mobility. ε_{rev} of 0.06PSN-0.61PMN-0.33PT is 7628, 60% higher than that of PMN-PT (4596), demonstrating an easier lattice deformation under external stimuli.

Figure S24. Rayleigh analysis of 0.67PMN-0.33PT and 0.06PSN-0.61PMN-0.33PT single crystals. (a) ε_{33} as a function of E_0 and the linear fitting results. (b) Comparison between the measured and calculated P - E hysteresis loops of 0.67PMN-0.33PT crystal. (c) Comparison between the measured and calculated P - E hysteresis loops of 0.06PSN-0.61PMN-0.33PT crystal.

Rayleigh analysis as a function of temperature was also carried out (Figure S25a) and the reversible dielectric constant ε_{rev} and Rayleigh parameter α as a function of temperature are summarize in Figure S25b. ε_{rev} dramatically enhances as the phase transition temperature T_{F-F} is approached. The total incensement is around 2 times, from 7628 at room temperature to 15215 around T_{F-F} . The enhancement

of α is much more distinct in comparison with ϵ_{rev} , which grows exponentially by 7 times from 3185 cm/kV to 24127 cm/kV, corresponding to the greatly improved domain wall motions.

Figure S25. Temperature dependent Rayleigh behavior of 0.06PSN-0.61PMN-0.33PT single crystal. (a) ϵ_{33} and (b), α and ϵ_{rev} at various temperatures.

Supplementary references:

1. Rajasekaran SV, Achary SN, Patwe SJ, Jayavel R, Mangamma G & Tyagi AK. Phase transformation in relaxor-ferroelectric single crystal $0.58\text{Pb}(\text{Sc}_{1/2}\text{Nb}_{1/2})\text{O}_3\text{-}0.42\text{PbTiO}_3$. *J Mater Res* **29**, 1054 (2014).
2. Singh AK, Pandey D. Evidence for M_B and M_C phases in the morphotropic phase boundary region of $(1-x)\text{Pb}(\text{Mg}_{1/3}\text{Nb}_{2/3})\text{O}_3\text{-}x\text{PbTiO}_3$: A Rietveld study. *Phys Rev B* **67**, 064102 (2003).
3. Scott J F. Models for the frequency dependence of coercive field and the size dependence of remanent polarization in ferroelectric thin films. *Integr Ferroelectr* **12**: 71-81 (1996).
4. Nomura Y, Tachi T, Kawae T, et al. Temperature dependence of ferroelectric properties and the activation energy of polarization reversal in (Pr, Mn)-codoped BiFeO_3 thin films. *Phys Status Solidi B* **252**: 833-838 (2015).
5. Tai CW, Baba-Kishi KZ. Relationship between dielectric properties and structural long-range order in $x\text{Pb}(\text{In}_{1/2}\text{Nb}_{1/2})\text{O}_3\text{-}(1-x)\text{Pb}(\text{Mg}_{1/3}\text{Nb}_{2/3})\text{O}_3$ relaxor ceramics. *Acta Mater* **54**, 5631-5640 (2006).
6. Shvartsman VV, Kleemann W, Łukasiewicz T, Dec J. Nanopolar structure in $\text{Sr}_x\text{Ba}_{1-x}\text{Nb}_2\text{O}_6$ single crystals tuned by Sr/Ba ratio and investigated by piezoelectric force microscopy. *Phys Rev B* **77**, 054105 (2008).
7. Shvartsman VV, Dkhil B, Kholkin AL. Mesoscale domains and nature of the relaxor state by piezoresponse force microscopy. *Annu Rev Mater Res* **43**, 423-449 (2013).
8. Li J, Li J, Qin S, et al. Effects of long-and short-range ferroelectric order on the electrocaloric effect in relaxor ferroelectric ceramics. *Phys Rev Appl* **11**: 044032 (2019).
9. Zuo R, Li F, Fu J, et al. Electric field forced c-axis oriented growth of polar nanoregions and rapid switching of tetragonal domains in BNT-PT-PMN ternary system. *J Eur Ceram Soc* **36**: 515-525 (2016).
10. Zhou X, Jiang C, Luo H, et al. Enhanced piezoresponse and electric field induced relaxor-ferroelectric phase transition in NBT-0.06BT ceramic prepared from hydrothermally synthesized nanoparticles. *Ceram Int* **42**(16): 18631-18640 (2016).

11. Zaman A, Hussain A, Malik R A, et al. Dielectric and electromechanical properties of LiNbO₃-modified (BiNa)TiO₃–(BaCa)TiO₃ lead-free piezoceramics. *J Phys D: Appl Phys* **49**(17): 175301 (2016).
12. Lin D, Li Z, Zhang S, et al. Electric-field and temperature induced phase transitions in Pb(Mg_{1/3}Nb_{2/3})O₃–0.3PbTiO₃ single crystals. *J Appl Phys* **108**(3): 034112 (2010).
13. Wang L, Zhai Y, Zheng L, et al. Intrinsic piezoelectricity in (K, Na) NbO₃-based lead-free single crystal: Piezoelectric anisotropy and its evolution with temperature. *Appl Phys Lett* **117**: 052904 (2020).
14. Zheng L, Jing Y, Lu X, et al. Temperature dependent piezoelectric anisotropy in tetragonal 0.63Pb(Mg_{1/3}Nb_{2/3})O₃-0.37PbTiO₃ single crystal. *Appl Phys Lett* **113**(10): 102903 (2018).
15. Liu D, Zhao R, Jafri HM, et al. Phase-field simulations of surface charge-induced polarization switching. *Appl Phys Lett* **114**, 112903 (2019)
16. Tinte S, Burton B P, Cockayne E, et al. Origin of the relaxor state in Pb(B_xB_{1-x})O₃ perovskites. *Phys Rev Lett* **97**(13): 137601 (2006).
17. Westphal V, Kleemann W, Glinchuk M D. Diffuse phase transitions and random-field-induced domain states of the “relaxor” ferroelectric Pb(Mg_{1/3}Nb_{2/3})O₃. *Phys Rev Lett* **68**(6): 847 (1992).
18. Kleemann W. Relaxor ferroelectrics: Cluster glass ground state via random fields and random bonds. *Phys Status Solidi B* **251**(10):1993–2002 (2014).
19. Rose L Y, Halliwill K D, Adams C J, et al. Mutational signatures in tumours induced by high and low energy radiation in Trp53 deficient mice. *Nat Commun* **11**(1): 1-15 (2020).
20. Bakaul S R, Kim J, Hong S, et al. Ferroelectric Domain Wall Motion in Freestanding Single- Crystal Complex Oxide Thin Film. *Adv Mater* **32**(4): 1907036 (2020).
21. He W, Li Q, Sun Y, et al. Investigation of piezoelectric property and nanodomain structures for PIN–PZ–PMN–PT single crystals as a function of crystallographic orientation and temperature. *J Mater Chem C* **5**(9): 2459-2465 (2017).
22. McGilly L J, Sandu C S, Feigl L, et al. Nanoscale defect engineering and the resulting effects on domain wall dynamics in ferroelectric thin films. *Adv Funct Mater* **27**(15): 1605196 (2017).

23. Jakes P, Erdem E, Eichel R A, et al. Position of defects with respect to domain walls in Fe³⁺-doped Pb[Zr_{0.52}Ti_{0.48}]O₃ piezoelectric ceramics. *Appl Phys Lett* **98**, 072907 (2011).
24. Jiang A Q, Chen Z H, Hui W Y, et al. Subpicosecond domain switching in discrete regions of Pb (Zr_{0.35}Ti_{0.65})O₃ thick films. *Adv Funct Mater* **22**(10): 2148-2153 (2012).
25. Schultheiß J, Liu L, Kungl H, et al. Revealing the sequence of switching mechanisms in polycrystalline ferroelectric/ferroelastic materials. *Acta Mater* **157**: 355-363 (2018).
26. Scott J F. Switching of ferroelectrics without domains. *Adv Mater* **22**, 5315-5317 (2010).
27. Zhang S, Li F. High performance ferroelectric relaxor-PbTiO₃ single crystals: Status and perspective. *J Appl Phys* **111**, 2-27 (2012).
28. Guo Y, Xu H, Luo H, et al. Growth and electrical properties of Pb(Sc_{1/2}Nb_{1/2})O₃-Pb (Mg_{1/3}Nb_{2/3})O₃-PbTiO₃ ternary single crystals by a modified Bridgman technique. *J Cryst Growth* **226**(1): 111-116 (2001).
29. Wang Z, He C, Qiao H, et al. In situ di-, piezo-, ferroelectric properties and domain configurations of Pb(Sc_{1/2}Nb_{1/2})O₃-Pb(Mg_{1/3}Nb_{2/3})O₃-PbTiO₃ ferroelectric crystals. *Cryst Growth Des* **18**(1): 145-151 (2018).
30. Griggio F, Jesse S, Kumar A, et al. Substrate clamping effects on irreversible domain wall dynamics in lead zirconate titanate thin films. *Phys Rev Lett* **108**, 157604 (2012).
31. Garcia J E, Perez R, Ochoa DA, et al. Evaluation of domain wall motion in lead zirconate titanate ceramics by nonlinear response measurements. *J Appl Phys* **10**, 6445 (2008).

Simultaneously achieving giant piezoelectricity and record coercive field enhancement in relaxor-based ferroelectric crystals

Liya Yang,^{#1,2,3} Houbing Huang,^{#4} Zengzhe Xi,⁵ Limei Zheng,^{1,*} Shiqi Xu,⁴ Gang Tian,¹ Yuzhi Zhai,¹
Feifei Guo,⁵ Lingping Kong,⁶ Yonggang Wang,⁶ Weiming Lü,^{7,*} Long Yuan,⁸ Minglei Zhao,¹ Haiwu
Zheng,² and Gang Liu^{6,*}

¹School of Physics, Shandong University, Jinan 250100, China

²International Joint Research Laboratory of New Energy Materials and Devices of Henan Province,
School of Physics and Electronics, Henan University, Kaifeng 475004, China

³Condensed Matter Science and Technology Institute, School of Instrumentation Science and
Engineering, Harbin Institute of Technology, Harbin 150080, China

⁴School of Materials Science and Engineering & Advanced Research Institute of Multidisciplinary
Science, Beijing Institute of Technology, Beijing 100081, China

⁵School of Materials and Chemical Engineering, Xi'an Technological University, Xi'an 710032, China

⁶Center for High Pressure Science and Technology Advanced Research, Shanghai 201203, China

⁷Spintronics Institute, School of Physics and Technology, University of Jinan, Jinan 250022, China

⁸Key Laboratory of Functional Materials Physics and Chemistry of the Ministry of Education, Jilin
Normal University, Changchun 130103, China

*Corresponding authors: zhenglm@sdu.edu.cn (L.Z.); sdylvw@ujn.edu.cn (W.L.);
liugang@hpstar.ac.cn (G.L.)

Abstract

A large coercive field (E_C) and ultrahigh piezoelectric are essential for ferroelectric materials used in high-drive electromechanical applications, especially when high power efficiency, small size, and light weight are required. The discovery of relaxor-PbTiO₃ crystals is a recent breakthrough in ferroelectric materials; they currently afford the highest piezoelectricity, but usually with a low E_C of the order of only 2–3 kV/cm. Such performance deterioration occurs because high piezoelectricity is usually interlinked with an easy polarization rotation, subsequently favoring a dipole switch under small external stimuli. Therefore, the search for novel ferroelectrics with both a large E_C and ultrahigh piezoelectricity has become an imminent challenge. Herein, by adopting a microstructure design strategy with small amounts of scandium (Sc) doping, a novel ternary Pb(Sc_{1/2}Nb_{1/2})O₃–Pb(Mg_{1/3}Nb_{2/3})O₃–PbTiO₃ crystal is reported, wherein the dispersed local heterogeneity comprises abundant tetragonal phases. Consequently, both elevated energy barriers and flattened potential wells are thermodynamically achieved, affording a large E_C of 8.2 kV/cm (greater than that of Pb(Mg_{1/3}Nb_{2/3})O₃–PbTiO₃ by a factor of three) and ultrahigh piezoelectricity ($d_{33} = 2630$ pC/N; $d_{15} = 490$ pC/N). The observed E_C enhancement is the largest reported for ultrahigh-piezoelectric ferroelectric crystals, significantly exceeding the previously reported enhancements induced by the doping effects and other techniques. This study provides a simple, practical, and universal route for improving functionalities in ferroelectrics with an atomic-level understanding, which is critical for future material-by-design.

Introduction

Various ferroelectric device types exist. However, the same basic mechanism occurs in all devices: spontaneous polarization (P_S) changes under external stimuli and then converts mechanical to electrical energy, or vice versa; here, the polarization rotation, extension, and switch are critical¹⁻³. Although ferroelectric materials differ, the core task for applications is to always make the above sequence of events possible, favorable, and stable, under both small and large drives^{4,5}. Consequently, high piezoelectric response and a large coercive field (E_C) are of fundamental importance, enabling both high operation efficiency and a wide operational field range in numerous electromechanical applications, such as high-power transducers and high-field actuators^{6,7}. In the past 30 years, ultrahigh piezoelectric perovskites, relaxor-PbTiO₃ (relaxor-PT) single crystals, have been discovered and greatly developed; they are the driving force for emerging electromechanical applications⁸. However, compositional modification for the simultaneous enhancement of piezoelectricity and E_C is challenging. For example, inferior piezoelectricity d_{33} (~1100 pC/N) is afforded and consequently considerable degradation of electromechanical response occurs upon the hard doping by manganese (Mn) in Pb(Mg_{1/3}Nb_{2/3})O₃-PbTiO₃ (PMN-PT) crystals^{9,10}; further, E_C with a low magnitude (~2.4 kV/cm), which is unsuitable for high-power and high-field applications, is afforded when soft doping strategies are employed¹¹. Figure 1a summarizes the relation between the coercive field E_C and piezoelectric coefficient d_{33} for various relaxor-PT ferroelectric crystals, demonstrating that high piezoelectric activity is generally associated with a low coercive field. Thus, while ultrahigh-piezoelectric relaxor-PT crystals are revolutionizing the electromechanical community, the crucial question to naturally arise is “is there a possibility to highly enhance the coercive field without sacrificing their ultrahigh piezoelectricity?”

Over the past decade, remarkable progress has been made toward achieving ultrahigh piezoelectricity of in relaxor-PT by introducing an additional structural heterogeneity and a slush-like polar state to manipulate interfacial energies and/or expand the phase coexistence region³¹⁻³³, which can further flatten the energy landscape and consequently enhance the piezoelectric response (Figure 1b). However, a remarkable high E_C has still not been achieved, which requires a high potential barrier to make the dipole switch difficult (Figure 1b)³⁴. For example, although recent studies have achieved piezoelectric activity of over 4000 pC/N in Sm-doped PMN–PT single crystals¹¹, the crystals afford a low E_C (~2.4 kV/cm). Furthermore, some studies demonstrated that E_C of over 10 kV/cm can be achieved by doping relaxor-PT with Yb and Ho, but a weak piezoelectric response (~1100 pC/N) is inevitably afforded^{10,35}. To date, to our knowledge, no study has reported a method for simultaneously achieving ultrahigh piezoelectricity and large E_C for relaxor-PT crystals, and the fundamental mechanism of this issue is not yet fully understood.

Herein, the thermodynamics and microstructure of the relaxor-PT system are re-scrutinized. As shown in Figure 1b, the piezoelectric activity and E_C could be simultaneously improved by making the potential wells flatter and enhancing the barrier between adjunct polar states; the former could be realized via nanoscale inhomogeneity and the latter is usually correlated to large tetragonality in the perovskite lattice⁸. Thus, if a high-piezoelectric parent matrix comprises involves some strongly tetragonal polar nano-regions (PNRs), large E_C enhancement with improved piezoelectricity could be achieved. Note that an obvious difference exists in the lattice constants between PMN and another relaxor $\text{Pb}(\text{Sc}_{1/2}\text{Nb}_{1/2})\text{O}_3$ (PSN), (Supplementary Note 1), which is critical for creating a highly anisotropic microstructure with large tetragonality in a relaxor-PT system. Thus, we studied the effect of scandium (Sc) substitution for B-site cations in a model ultrahigh-piezoelectric relaxor-PT perovskite, PMN–PT. The resulting ternary 0.06PSN–0.61PMN–0.33PT crystals demonstrate excellent piezoelectric activity and electromechanical

coupling response ($d_{33} = 2630$ pC/N, $k_{33} = 90.8\%$; $d_{15} = 490$ pC/N, $k_{15} = 54.7\%$), where the shear activity is **twice that of the** binary PMN–PT counterpart, **and** the ultrahigh longitudinal performance is maintained. **Notably**, E_C was successfully **improved** by over **three** times to 8.2 kV/cm. Such an enhancement of **the** coercive field is the largest reported **for** giant piezoelectric materials, far **exceeding** all experimentally observed results. **We aimed to echo** the proposed free energy landscape design, providing not only effective experimental routes but also vital theoretical guidelines for designing better ferroelectric materials.

Figure 1 Dilemma in ferroelectrics: enhanced E_C is usually achieved at the expense of piezoelectricity. (a) d_{33} vs. E_C for various relaxor-PT single crystals. The red dashed line denotes the tendency of most relaxor-PT crystals. Generally, E_C enhancement is accompanied by inferior piezoelectricity. Alternatively, our 0.06PSN–0.61PMN–0.33PT (red star) affords highly remarkable results. Data from Refs. 9, 11–30 and this work. (b) Schematic of the different free energy landscapes and the corresponding macroscopic performances.

Results

Materials properties. **Herein**, the composition selection of the PSN–PMN–PT solid solution is based on two considerations. **First**, the morphotropic phase boundary (MPB) compositions are definitely required to optimize the piezoelectric properties^{8,19}. From the composition-dependent phase diagram of

binary PMN–PT and PSN–PT, we deduced the MPB regions of the ternary PSN–PMN–PT system, as shown in the blue region in Figure 2a. Additionally, a low Sc content doping strategy was employed in the crystal design because the piezoelectric activity of PSN–PT is significantly weaker than that of PMN–PT^{36,37}. Subsequently, high-quality crack-free 0.06PSN–(0.94– x)PMN– x PT ($x=0.31$ – 0.35) crystals with a diameter of 25 mm were successfully grown. Figure 2b displays the photograph of the as-grown crystals. All samples were cut from the same thin crystal wafer with identified PT contents and then poled along [001]_C for domain-engineered configurations. The quantitative compositions of the samples were demonstrated to be 0.06PSN–0.61PMN–0.33PT via energy dispersive spectrometry, satisfying the material design requirements.

Compared to their PMN–PT binary counterpart, 0.06PSN–0.61PMN–0.33PT crystals exhibit high longitudinal piezoelectric activity as well as obviously higher shear piezoelectric activity, higher ferroelectric–ferroelectric phase transition temperature T_{F-F} , and a larger coercive field E_C . As shown in Figure 2c, the polarization–electric hysteresis loop (P – E loop) indicates that the 0.67PMN–0.33PT crystals afford a low E_C of 2.6 kV/cm, severely hampering its potential applications. Notably, after a little Sc substitution, the coercive field of the resulting 0.06PSN–0.61PMN–0.33PT crystals improved to 8.2 kV/cm, which is three times larger than that of PMN–PT. To the best of our knowledge, such an enhancement represents the most advanced enhancement reported to date for almost all investigated ferroelectrics with ultrahigh piezoelectric coefficients of over 2000 pC/N. To verify the repeatability of the extraordinary E_C values, we analyzed several different samples; all samples afforded E_C values of around 8 kV/cm (Figure S1). Note that E_C is not an intrinsic property of ferroelectrics^{38,39}. Poling/de-poling is related to voltage-induced domain switching, including nucleation of new domains at a defect site (normally near domain walls) and domain growth. Thus, we considered the dynamics of domain switching by measuring the E_C at various frequencies. As is well known, domain switching is

considerably easy at low frequencies, which is related to domain switching under a very low field and long holding time of the applied fields. For the 0.06PSN–0.61PMN–0.33PT crystals, a large E_C of ~7.5 kV/cm is maintained at frequencies as low as 0.1 Hz, and 11.8 kV/cm is afforded at 100 Hz (Supplementary Note 2 and Figure S2). Moreover, we observed a low conductive current during the entire poling process, confirming the high quality of the crystals (Supplementary Note 3).

The T_{F-F} observed in 0.06PSN–0.61PMN–0.33PT crystals is particularly interesting, ~85 °C, which are 13 °C higher than that observed for the PMN–PT crystals with similar PT contents (Figure 2d), which is promising for a wide temperature usage range and drive field stability. Figure S4 displays the temperature dependence of the piezoelectric response of the 0.06PSN–0.61PMN–0.33PT crystals, exhibiting a variation of 140% in d_{33} , which is considerably lower than that of PMN–PT crystals (200%-300%)⁸. Furthermore, a relatively high E_C of over 6.2 kV/cm can be maintained till the occurrence of the phase transition (Figure S5).

Remarkably, the piezoelectric coefficients d_{15} and electromechanical coupling constants k_{15} of the 0.06PSN–0.61PMN–0.33PT crystals were found to be 490 pC/N and 54.7%, respectively (Figure 2e), featuring a significantly larger shear piezoelectric response that is much greater than that of other [001]_C-poled relaxor-PT systems. Moreover, 0.06PSN–0.61PMN–0.33PT crystals exhibit almost the same ultrahigh longitudinal property ($d_{33} = 2630$ pC/N) as the 0.67PMN–0.33PT crystals²². We also conducted detailed piezoelectric force microscopy (PFM) characterizations to investigate the polarization switching behavior and the local piezoelectric deformations of the PSN–PMN–PT crystals; the results further support their superior piezoelectric response from a microscopic perspective (Figure S6). Below T_{F-F} , no obvious changes were observed in the domain morphology with increasing temperature (Figure S7), which well agrees with the weak variations of the piezoelectric response and

electromechanical properties, namely, a relatively strong thermal stability of functionality (Figure S4). Figure 2f presents a radar chart summarizing the critical properties, including d_{33} , k_{33} , d_{15} , k_{15} , E_C , and T_{F-F} of various $[001]_C$ -poled relaxor-PT crystals with MPB compositions. The figure shows that the 0.06PSN–0.61PMN–0.33PT crystals cover an extremely large area, thereby demonstrating their superior overall performance and greater efficiency for potential device applications^{22,38,39}.

Figure 2 Functional characterizations of PSN–PMN–PT single crystals. (a) Phase diagram of the PSN–PMN–PT ternary system. The blue area indicates the MPB region of the PSN–PMN–PT system. The orange point indicates the composition, 0.06PSN–0.61PMN–0.33PT, which is intensively investigated herein. (b) Photograph of the as-grown PSN–PMN–PT crystals, showing a large dimension of $\Phi 25 \times 35 \text{ mm}^3$ without cracks. (c) P – E loop of 0.06PSN–0.61PMN–0.33PT in comparison with that of 0.67PMN–0.33PT. (d) Temperature dependence of the relative dielectric permittivity of the two crystals. 0.06PSN–0.61PMN–0.33PT exhibits a comparable Curie temperature T_C but a markedly improved T_{F-F} than 0.67PMN–0.33PT. (e) Shear piezoelectric performance of various $[001]_C$ -poled relaxor-PT crystals with MPB composition, demonstrating that 0.06PSN–0.61PMN–0.33PT exhibits

significantly superior d_{15} and k_{15} . (f) Comparison of the essential parameters for various [001]_C-oriented PbTiO₃-based relaxor ferroelectric single crystals with MPB composition. The 0.06PSN–0.61PMN–0.33PT crystals cover the largest area, denoting a superior overall performance. Data from Refs. 22, 40, 41, and this work.

Relaxor behavior. A key feature of relaxor-ferroelectric solid solutions is the existence of local heterogeneity, such as PNRs, which contribute over 50% to the dielectric/piezoelectric response according to the recent cryogenic experimental measurements^{32,42}. Therefore, we investigated the relaxor behavior of the ternary 0.06PSN–0.61PMN–0.33PT crystals and explored the possible differences from their binary 0.67PMN–0.33PT counterparts, with the aim to determine why 0.06PSN–0.61PMN–0.33PT crystals simultaneously exhibit ultrahigh piezoelectric activity and extremely large E_C , which seems unusual in most ferroelectric solid solution systems, as summarized in Figure 1a.

Figure 3a shows the temperature-dependent reciprocal of the dielectric response of the 0.06PSN–0.61PMN–0.33PT and 0.67PMN–0.33PT crystals, indicating that the phase transitions proceed gradually rather than sharply with temperature. Such a diffuseness characteristic is a relaxor feature, causing the deviation from the Curie–Weiss law, where the Burns temperatures (T_B) of around 268 °C can be derived for 0.06PSN–0.61PMN–0.33PT, 40 °C higher than that of 0.67PMN–0.33PT (228 °C). Thus, we reasonably deduce that during paraelectric-to-ferroelectric transitions, PNRs appear earlier (at a higher temperature) in 0.06PSN–0.61PMN–0.33PT than in 0.67PMN–0.33PT, presenting polarized precursor clusters. Such a diffused characteristic is further supported by quantitative analysis via the modified Curie–Weiss law⁴³:

$$\frac{1}{\varepsilon} - \frac{1}{\varepsilon_m} = \frac{(T - T_m)^\gamma}{C} \quad (1)$$

where ε_m is the maximum dielectric constant at T_m , C is the Curie-like constant, and γ describes the degree of diffuseness. Linear fitting of $\ln(1/\varepsilon_{33}-1/\varepsilon_m)$ versus $\ln(T-T_m)$ data yields γ values of 1.96 and 1.74 for 0.06PSN–0.61PMN–0.33PT and 0.67PMN–0.33PT respectively (Figure 3b), suggesting a stronger relaxor nature in the ternary system. Further autocorrelation function analysis based on PFM characterizations demonstrates that it is much more difficult for the ternary crystal to establish a homogeneous polarization order than only short-range orders between neighboring clusters (Supplementary Note 6).

Figure 3c shows the frequency dependence of high-temperature dielectric properties of the ternary 0.06PSN–0.61PMN–0.33PT crystals, where the magnitude of the dielectric permittivity decreases and the dielectric maximum shifts to higher temperatures with increasing frequency, again demonstrating strong relaxor behavior. Notably, 0.06PSN–0.61PMN–0.33PT exhibits a “ T_m shift” (shift of the dielectric maxima temperature with frequency over the range of 1 kHz–1MHz) that is twice that of the 0.67PMN–0.33PT crystals (5 K vs. 2.5 K, Figure 3c and Figure S10), signifying strong interactions between PNRs and the development of local correlations⁴². Furthermore, the frequency dependence of the high-temperature dielectric data can be well fitted using the Vogel–Fulcher relation⁴⁴:

$$f = f_0 \exp[-E_a/(k_B(T_m - T_{VF}))] \quad (2)$$

where f_0 is the Debye frequency, T_m is the temperature of the permittivity maximum, and T_{VF} is the static freezing temperature, which can be deemed as T_m at 0 Hz. T_{VF}/T_m is a semi-quantitative parameter employed for evaluating PNR interactions⁴⁴. E_a represents the activation energy of the polarization fluctuation in an isolated cluster that stems from the development of a short-range order; thus, a larger E_a suggests stronger interactions between neighboring PNRs. The fitted results are given in Figures 3d and 3e, which show that the E_a of the 0.06PSN–0.61PMN–0.33PT crystals is considerably higher than

that of 0.67PMN–0.33PT crystals (~ 0.024 eV vs. 0.008 eV), **signifying** stronger **interactions** among PNRs in **the** ternary crystals. Considering the higher T_B , larger γ , shorter range order, higher E_a , and lower T_{VF}/T_m (Figure 3f), we reasonably deduce that 0.06PSN–0.61PMN–0.33PT exhibits more relaxor and diffused characteristics with much **significantly** stronger interactions between adjacent PNRs than **0.67PMN–0.33PT**.

Figure 3 Relaxor behavior. (a) Temperature dependence of the reciprocal of dielectric permittivity, and T_B is obtained by fitting **with the modified** Curie–Weiss law. (b) Modified Curie–Weiss law fitting results, from which γ can be obtained. (c) High-temperature dielectric property of 0.06PSN–0.61PMN–0.33PT crystals measured at various frequencies from 1 kHz to 1 MHz. (d) Vogel–Fulcher fitting results on the data shown in Figure 3c and Figure S7, from which the activation energy E_a and static freezing temperature T_{VF} can be **determined**. (e) Summary of E_a for various pure PMN, PMN–PT, and PSN–PMN–PT. Data are from **Refs.** 44–46 and this work. (f) T_{VF}/T_m value as a function of frequency for 0.67PMN–0.33PT and 0.06PSN–0.61PMN–0.33PT.

Highly dispersed local heterogeneous structure with considerable tetragonal phase. The strong polar cluster interaction is directly related to the structural instability and finally contributes to the material functionality^{47,48}. This motivated us to further resolve the local microstructure of the 0.06PSN–0.61PMN–0.33PT crystals, and investigate its possible phase coexistence and complex crystallographic symmetry, which are crucial for understanding why PSN–PMN–PT crystals simultaneously afford ultrahigh possesses giant piezoelectricity and a high coercive field.

Figure 4a shows an aberration-corrected high-angle annular dark-field scanning transmission electric microscopy (HAADF-STEM) image. From the image we determined the polarization vector P_s of each unit cell column based on the atomic positions. Note that a dispersed polar state with multiphase, including rhombohedral (R) and/or orthorhombic (O), tetragonal (T), and monoclinic (M), was detected. Moreover, 0.06PSN–0.61PMN–0.33PT exhibited considerably smaller (2–4 nm) than 0.67PMN–0.33PT (8–20 nm), suggesting a higher density of domain walls/phase interfaces and abundant local heterogeneous structure (Figures S11 and S12).

The abundant tetragonal phase in 0.06PSN–0.61PMN–0.33PT determined via HAADF-STEM is notable. This behavior was further evidenced in our high-resolution X-ray diffraction (XRD) characterizations, from which a detailed analysis of the peak positions and intensities was conducted and the volume fraction of the tetragonal component was estimated to be 34.5% (Figure 4b), which is significantly higher than that of 0.67PMN–0.33PT, 13.7% (Figure S13). Electric-field- and temperature-dependent structural evolutions were also conducted, and the structure–piezoelectricity relation was studied (Supplementary Note 10).

Furthermore, we calculated the distances between A-site cations on a per-unit cell basis via HAADF-STEM and estimated the local lattice anisotropy by determining the local c/a ratio. As shown

in Figure 4c, the standard deviation of the lattice parameter is significantly larger for 0.06PSN–0.61PMN–0.33PT than that for 0.67PMN–0.33PT, demonstrating a higher fluctuation in the sublattice parameter. Additionally, the local c/a ratios for PSN–PMN–PT varied more than those for PMN–PT, indicating a much larger tetragonality and a more dispersive behavior. These observations are consistent with the XRD data (Table S2).

Then, we conducted geometric phase analysis (GPA) on the HAADF-STEM images of 0.06PSN–0.61PMN–0.33PT and 0.67PMN–0.33PT crystals, from which we derived the variations of local strain S_{xx} along $[001]_C$ were (Figures 4d and 4e). Notably, the ternary crystals possess significantly higher local strain ($\sim 3\%$) than their binary counterparts (1.5%), suggesting a large tetragonal lattice deformation c/a ratio.

The novel microstructure of the PSN–PMN–PT crystals has not been previously reported for any other ultrahigh-piezoelectric materials. It features a highly dispersed local heterogeneous structure with abundant tetragonal phases, markedly differing from the usual behavior of binary relaxor-PT crystals, where the high piezoelectric activity is only found in the rhombohedral-side MPB compositions. The phase-field calculations well match our experimental discoveries, verifying the experimentally observed microstructure and functionality. As shown in Figure 4f, pure PMN–PT exhibits rhombohedral characteristics with a large domain size. When some tetragonal nanosized phases are introduced into this matrix (similar to the PSN–PMN–PT case), a dispersed domain structure with decreased domain size forms, and multiphase coexistence becomes inevitable, well agreeing with the transmission electron microscopy (TEM) results shown in Figure 4a. We calculated the magnitudes of E_C and d_{33} of these two systems, from which we observed significant E_C enhancement without piezoelectricity reduction due to appropriate doping of the tetragonal phase (Figure S18).

Figure 4 Microstructure. (a) Atomic-resolution TEM images of the $[001]_c$ -oriented 0.06PSN–0.61PMN–0.33PT crystals, where the P_s directions are given for each unit-cell column. The possible phase structures can be deduced using the P_s directions (R: rhombohedra; O: orthorhombic; M_A/M_C : monoclinic; and T: tetragonal). (b) High-resolution XRD pattern and the optimal refinement results for 0.06PSN–0.61PMN–0.33PT. (c) Unit cell c/a ratios for 0.06PSN–0.61PMN–0.33PT and 0.67PMN–

0.33PT derived from the TEM characterizations. (d) Local strain S_{xx} mapping extracted from the HAADF-STEM lattice image of 0.06PSN–0.61PMN–0.33PT in Figure 4a and 0.67PMN–0.33PT in Figure S9 via GPA, and the data along the white dotted lines are extracted and shown in (e). (f) Phase-field simulations of the domain structures of pure PMN–PT and that doped with tetragonal phase. PMN–PT exhibits pure R characteristics; after the introduction of the tetragonal phase, it exhibits multi-phase including R, O, and T characteristic with a reduced domain size. Different phases and various P_S directions in the same phase are denoted by different colors.

Discussion

Based on the experimental and phase-field simulation results, the ultrahigh piezoelectricity and extremely large E_C in the PSN–PMN–PT system can be explained in the mesoscale. Previous studies have demonstrated that the introduction of cations into the B-site of PMN–PT can afford a high level of charge inhomogeneity⁴⁹⁻⁵¹, consequently yielding strong relaxor behavior, as confirmed in this study (Figure 3 and Supplementary Note 12). Compared to PMN–PT, it is considerably more difficult for PSN–PMN–PT to establish a ferroelectric long-range order with only short-range ordering between neighboring clusters, accounting for symmetry breaking; thus, we observed a highly dispersed local heterogeneous structure (Figure 4). The dispersed micro polar state with multiphase coexistence is strongly correlated with abundant local heterogeneity (Figure 4a), which can significantly flatten the free energy profile, significantly contributing to the ultrahigh piezoelectric activity³¹. In addition to the highly dispersed characteristic, 0.06PSN–0.61PMN–0.33PT crystal also features a considerable tetragonal phase component with a relatively large c/a ratio (Figure 4b–d). Previous studies on the structure–property relation of relaxor-PT showed that tetragonal-rich crystals usually exhibit large E_C ³⁰, where the large c/a ratio is directly related to the high potential barrier between different polar states. After Sc doping into PMN–PT, nanosized tetragonal domains are highly dispersed into the matrix (Figure 4a). These tetragonal polar regions strongly interact with their neighboring clusters (Figures 3e

and 3f), acting as “frozen seeds” in the entire matrix and pinning the P_S switch via a cooperative effect, consequently yielding unparalleled coercive field enhancement. The pinning effect of the tetragonal polar regions on domain switch is associated with the difficult domain nucleation and growth process^{52,53}, which is supported by the large activation electric field in 0.06PSN–0.61PMN–0.33PT (Supplementary Note 13).

Although an ultrahigh piezoelectric response and a large E_C are generally exclusive in a single ferroelectric material (Figure 1a), the enhancement of E_C in our PSN–PMN–PT crystal is notably not at the expense of the piezoelectric activity, which can be explained from its particular microstructure. The dispersed microstructure is strongly correlated to a slush-like polar state with coexisting multiphases, including O/R, T, and M, signifying the instability of the ferroelectric phases, and intrinsically contributing substantially to the ultrahigh piezoelectric activity. Moreover, the interfacial energies can be manipulated using the abundant local structure heterogeneity arising from the tortuous interfaces between adjacent clusters, and the small domain size³¹, further improving the piezoelectric response. Note that due to the existence of the tetragonal phase, fully (001) poled crystals may contain considerable single-domain components⁵⁴ and possess large shear piezoelectric activity that stems from the easy polarization rotation (Figures S21 and S22), which partially explains the two times larger d_{15} in 0.06PSN–0.61PMN–0.33PT than that in 0.67PMN–0.33PT.

Based on the above discussion, we propose a thermodynamic understanding of the inherent correlation between the macrostructure and materials properties: the dispersed heterogeneous structure with multiphase coexistence makes the free energy extremely flat and the highly tetragonal polar regions pin the domain switch by enhancing the potential barriers, resulting in a “flat and deep” potential well Figure 5. This peculiar free energy profile causes a difficult polarization switch and an easy polarization

rotation; consequently, an ultrahigh piezoelectric response and extremely large E_C are simultaneously achieved. Both hard (large E_C) and soft (high piezoelectric response) doping properties are affording using this strategy, successfully addressing the longstanding challenge that excellent sensitivity and high stability of dipoles are generally exclusive, and achieving a striking enhancement of the overall performance.

To verify the importance of the delicate optimization of the composition, our study is compared with previous studies based on similar material systems. In PSN–PMN–PT crystals with PSN:PMN ratio of 1:3, 1:1, or 3:1, Wang et al.⁵⁵ determined the piezoelectric constant as 1200–1600 pC/N and the coercive field as 4–6 kV/cm, which are significantly inferior to those of our 0.06PMN–0.61PMN–0.33PT sample (PSN:PMN ratio ~1:10). It has been demonstrated^{56,57} that the introduction of PSN into PMN–PT can yield a high level of lattice anisotropy (Supplementary Note 1), favoring a tetragonal phase. Thus, if a superfluous amount of Sc is introduced, the tetragonal clusters may become too large for dispersal into the entire ferroelectric matrix, destroying the desirable local heterogeneous microstructure; additionally, a phase separation could occur, causing functional degradation. Moreover, the importance of MPB composition in the piezoelectric response should be emphasized. Xi et al.^{7,58} reported that in 0.06PSN–0.63PMN–0.31PT single crystals, although with a large coercive field of 8.17 kV/cm, the maximum piezoelectric constant is only ~1200 pC/N, which is similar to that of PZT ceramics or lead-free crystals. Guo et al.⁵⁹ reported that 0.05PSN–0.63PMN–0.32PT single crystals in R phase exhibit inferior performance with $d_{33} = 1200$ pC/N and $\epsilon_{33} \sim 3500$. Compared to previous studies, our 0.06PSN–0.61PMN–0.33PT single crystals exhibit substantially superior overall performance (Supplementary Note 15). Therefore, the appropriate balance between various effects, including relaxor and long-range order and tetragonal and other phases, need to be considered for materials-by-design. This strategy should be converted to an atomistic model to understand the contribution of each atom to the free energy

profile in a complex material system to ultimately realize high-performance and/or high-power applications.

The natural question arises that “why has such simultaneous improvement not been obtained in previous studies?” Currently, the most conventional strategy employed to enlarge the coercive field in relaxor-PT systems is hard doping^{8,9}. In hard doping, a small amount (<2 mol%) of acceptor ions such as Mn^{2+/3+}, is substituted into the B-sites of perovskite lattices, yielding acceptor–oxygen vacancy defect dipoles. These defect dipoles occupy energetically preferred sites in the lattice and align themselves along a preferential direction within a ferroelectric domain, and then, they move to the highly stressed areas of domain walls⁶⁰. These defect dipoles pin the domain walls and stabilize the domain, establishing a parallel arrangement of defect dipole and local ferroelectric polarization, causing an offset of P – E behavior that is experimentally characterized as internal bias, which effectively increases the E_C by 30% compared to that of undoped materials⁶¹. Such a significantly reduced degree of switchable polarization is accompanied by suppressed domain wall mobility, inevitably resulting in an inferior piezoelectric response. Alternatively, based on the P – E characterization results (Figure 2c), an internal bias was not observed in the PSN–PMN–PT crystals, signifying the presence of a distinct mechanism associated with the intrinsically high lattice strain rather than the domain clamping effect observed for Mn-doped crystals. Therefore, the piezoelectric coefficients did not decrease with Sc doping, due to the no-loss or even enhanced extrinsic piezoelectric contribution. These easily removable domain walls are further suggested by Rayleigh analysis (Figures S24 and S25), which shows that a large Rayleigh parameter α is afforded at both the room temperature and the temperature near T_{F-F} , which is not favored in hard doping^{8,61}.

In conclusion, by employing a **precise** microstructure-by-design technique, we successfully addressed the long-sought-after **crystals with simultaneous** ultrahigh piezoelectricity and **unparalleled** enhancement of E_C . Within the theoretical framework, **we proposed** a thermodynamic understanding of the inherent correlation between **the** free-energy landscape and material properties, where a “flat and deep” potential well is derived. **To our knowledge, our** dataset **is** the first confirmation of the existence of extremely large E_C in **an** ultrahigh piezoelectric **material**. **Furthermore**, although **the** PMN–PT solid solution is employed **herein**, our proposed strategy is likely a universal and effective method for **designing** high-performance functional materials with both high tolerance and sensitivity to the external field.

Figure 5. Simultaneous achievement of ultrahigh piezoelectricity and extremely large E_C . The particular local structure and relatively large c/a ratio in the PSN–PMN–PT are used to manipulate the free energy profile in different ways, **affording** a “deep and flat” free energy landscape. The PSN–PMN–PT system **exhibits** various characteristics including high level of local structure heterogeneity, slush-like multiphase coexistence, small domain size, and high density of PNRs, accounting for a flat

potential well. Moreover, the lattice of PSN–PMN–PT system contains a considerable tetragonal phase component, demonstrating large anisotropy, which consequently contributes to a deep well, namely an enhanced ΔG . The flattened potential well facilitates the polarization (P_S) rotation/elongation around the equilibrium position, enhancing the piezoelectric response, and the enhanced energy barrier makes the P_S switch more difficult, contributing to a large E_C .

Methods

Crystal growth. The PSN–PMN–PT single crystals were grown using the Bridgman technique. High-purity Sc_2O_3 (99.99%), Nb_2O_5 (99.95%), $(\text{MgCO}_3)_4 \cdot \text{Mg}(\text{OH})_2 \cdot 5\text{H}_2\text{O}$ (>99.0%), PbO (>99.0%), and TiO_2 (>99.0%) were used as raw materials. The precursors MgNb_2O_6 and ScNbO_4 were synthesized in advance to avoid the impurity phase formation. Then, MgNb_2O_6 , ScNbO_4 , TiO_2 and PbO powders were mixed and placed in a Pt crucible wrapped with a sealed Al_2O_3 crucible. The crucible was placed in a computer-controlled Bridgman furnace, which was heated from 600 to 1400 °C at a rate of 10 °C/min and maintained at 1400 °C for 10 h, and a stable temperature gradient of 30–50 °C/cm was formed. The crucible was descended at a rate of 0.2–0.4 mm/h, and the PSN–PMN–PT single crystals gradually grew via spontaneous nucleation.

Sample preparation and electrical property measurements. All the samples used herein were $[001]_C$ -oriented with $x//[100]_C$, $y//[010]_C$ and $z//[100]_C$ via XRD. After cutting and polishing, all the samples were annealed at 600 °C for 1 h to eliminate the stress generated during sample preparation. A gold electrode was sputtered on the parallel $(001)_C$ faces. The temperature dependence of the dielectric constants was measured using an LCR meter (Agilent, 4284A) with a 2 °C/min step. P – E loops were obtained using a Precision Premier II tester (Radiant Technologies Albuquerque). After being poled by a DC E -field of 10 kV/cm at room temperature, the longitudinal piezoelectric coefficient d_{33} was recorded using a quasi-static d_{33} meter (Institute of Acoustics, ZJ-4A) and shear coefficient d_{15} was measured

using the resonance method. The resonance and anti-resonance frequencies were obtained using an Agilent 4294A impedance-phase gain analyzer, based on which the electromechanical coupling factors k_{33} and k_{15} were obtained.

PFM measurements and the autocorrelation function technique. For the PFM observations, the samples were ground to ensure a flat surface using the Al_2O_3 grinding powder and subsequently polished using polycrystalline diamond suspensions with abrasive particles of 9 μm , 3 μm , 1 μm , and 20 nm (MetaDi Supreme, Buehler). The PFM studies were performed using a Cypher ES (Asylum Research) in DART mode using Ir/Pt-coated conductive tips (Nanoworld, EFM). The autocorrelation images were obtained based on the PFM domain images via the following transformation^{62, 63},

$$C(r_1, r_2) = \sum_{x,y} D(x, y)D(x + r_1, y + r_2) \quad (1)$$

where $D(x,y)$ is the piezoelectric signal and the autocorrelation function $C(r_1, r_2)$ is the two-dimensional polarization-polarization correlation function. Furthermore, $\langle C(r) \rangle = \sigma^2 \exp[-(r/\xi)^{2h}]$ is the averaged autocorrelation function $C(r_1, r_2)$ over all in-plane directions.

Scanning Transmission Electron Microscopy (STEM) experiments. The TEM samples were prepared using a Tescan LYRA-3 XUM Model focused ion beam instrument. The selected area electron diffraction patterns and morphology of the crystals in Figure S8 were characterized using TEM FEI Talos F200. STEM images were acquired on a spherical aberration-corrected FEI Titan G2 microscope operated at 300 kV using a HAADF detector. All STEM images were Fourier-filtered using a lattice mask to remove noise. The strain analyses in HAADF-STEM images were obtained through GPA using the custom scripts in the Gatan DigitalMicrograph software⁶⁴. The polar vector for each unit cell was determined as the B-site cation displacement relative to its four nearest A-site neighbor cations by fitting

atom positions as two-dimensional Gaussian peaks⁶⁵, which are mapped in the HADDF-STEM images of Figure 4a and Figure S9.

Phase-field simulations. In the phase-field **simulations**, the polarization $P_i(r, t)$ (x, y, z) **denotes** the order parameter, which describes the ferroelectric polarization evolution. The temporal evolution of the polarization can be described by the time-dependent Ginzburg–Landau equation:

$$\frac{\partial P_i(\mathbf{r}, t)}{\partial t} = -L \frac{\delta F_P}{\delta P_i(\mathbf{r}, t)}, \quad (i = x, y, z), \quad (2)$$

where t is the simulation time, L is the kinetic coefficient, \mathbf{r} is the **spatial** position, **and** F_P is the total free energy of the system **that is denoted as follows**⁶⁶

$$F_P = \iiint (f_{bulk}(P_i) + f_{elas}(P_i, \varepsilon_{ij}) + f_{elec}(P_i, E_i) + f_{grad}(P_{i,j})) dV, \quad (3)$$

where f_{bulk} , f_{elas} , f_{elec} and f_{grad} represent the Landau bulk, elastic, electrostatic, and gradient **energy densities**, respectively. A stress-free boundary condition is adopted. The bulk energy density f_{bulk} can be described as a six-order polynomial:

$$\begin{aligned} f_{bulk} = & \alpha_1 (P_x^2 + P_y^2 + P_z^2) + \alpha_{11} (P_x^4 + P_y^4 + P_z^4) \\ & + \alpha_{12} (P_x^2 P_y^2 + P_x^2 P_z^2 + P_z^2 P_y^2) + \alpha_{112} [P_x^4 (P_y^2 + P_z^2) \\ & + P_y^4 (P_y^2 + P_x^2) + P_z^4 (P_y^2 + P_x^2)] + \alpha_{111} (P_x^6 + P_y^6 + P_z^6) \\ & + \alpha_{123} P_x^2 P_y^2 P_z^2 \end{aligned}, \quad (4)$$

where α_1 , α_{11} , α_{12} , α_{111} , α_{112} and α_{123} are the Landau energy coefficients. Among **which**, only α_1 is **temperature-dependent**, $\alpha_1 = (T - T_C)/(2\varepsilon_0 C_0)$, where T is the temperature, T_C is the Curie temperature, C_0 is the Curie constant, and $\varepsilon_0 = 8.85 \times 10^{-12}$ is the permittivity of vacuum⁶⁷. The Landau coefficients of PMN–0.3PT and PMN–0.42PT were taken from **Ref. 68**.

The gradient energy density can be expressed as

$$f_{grad} = \frac{1}{2} G_{ijkl} \frac{\partial P_i}{\partial r_j} \frac{\partial P_k}{\partial r_l}, \quad (5)$$

where G_{ijkl} is the gradient energy coefficient. The electrostatic energy density can be written as

$$f_{elec} = -\frac{1}{2} \varepsilon_0 K_{ij}^b E_i E_j - E_i P_i, \quad (6)$$

where K_{ij}^b is the background relative permittivity and E_i is the electric field, which can be calculated as

$$E_i = -\frac{\partial \varphi}{\partial r_i}. \quad (7)$$

The electric potential φ can be obtained by solving the electrostatic equilibrium equation

$$\varepsilon_0 K_{ij}^b \frac{\partial^2 \varphi}{\partial r_i \partial r_j} = -\frac{\partial P_i}{\partial r_i}. \quad (8)$$

Equations were numerically solved via the semi-implicit Fourier-spectral method⁶⁹.

Data availability :

The data that support the findings of this study are available from a public repository at <https://doi.org/10.6084/m9.figshare.19346039.v1>.

References :

1. Fu H, Cohen RE. Polarization rotation mechanism for ultrahigh electromechanical response in single-crystal piezoelectrics. *Nature* **403**, 281-283 (2000).
2. Damjanovic D. Contributions to the Piezoelectric Effect in Ferroelectric Single Crystals and Ceramics. *J Am Ceram Soc* **88**, 2663-2676 (2005).
3. Tagantsev AK, Cross LE, Fousek J. *Domains in Ferroic Crystals and Thin Films*. (Springer press, New York, 2010).
4. Hao J, Li W, Zhai J, Chen H. Progress in high-strain perovskite piezoelectric ceramics. *Mater Sci Eng: R-Reports* **135**, 1-57 (2019).
5. Thong H-C, *et al.* Technology transfer of lead-free (K, Na)NbO₃-based piezoelectric ceramics. *Mater Today* **29**, 37-48 (2019).
6. Zhang S, Li F, Jiang X, Kim J, Luo J, Geng X. Advantages and Challenges of Relaxor-PbTiO₃ Ferroelectric Crystals for Electroacoustic Transducers- A Review. *Prog Mater Sci* **68**, 1-66 (2015).
7. Zheng T, Wu J, Xiao D, Zhu J. Recent development in lead-free perovskite piezoelectric bulk materials. *Prog Mater Sci* **98**, 552-624 (2018).
8. Zhang S, Li F. High performance ferroelectric relaxor-PbTiO₃ single crystals: Status and perspective. *J Appl Phys* **111**, 031301 (2012).
9. Zhang S, Lee SM, Kim DH, Lee HY, Shrout TR. Characterization of Mn-modified Pb(Mg_{1/3}Nb_{2/3})O₃-PbZrO₃-PbTiO₃ single crystals for high power broad bandwidth transducers. *Appl Phys Lett* **93**, 122908 (2008).
10. Zheng L, Sahul R, Zhang S, Jiang W, Li S, Cao W. Orientation dependence of piezoelectric properties and mechanical quality factors of 0.27Pb(In_{1/2}Nb_{1/2})O₃-0.46Pb(Mg_{1/3}Nb_{2/3})O₃-0.27PbTiO₃:Mn single crystals. *J Appl Phys* **114**, 104105 (2013).
11. Li F, *et al.* Giant piezoelectricity of Sm-doped Pb(Mg_{1/3}Nb_{2/3})O₃ -PbTiO₃ single crystals. *Science* **364**, 264 (2019).

12. Chen Z, Zhang Y, Li S, Lu XM, Cao W. Frequency dependence of the coercive field of $0.71\text{Pb}(\text{Mg}_{1/3}\text{Nb}_{2/3})\text{O}_3\text{-}0.29\text{PbTiO}_3$ single crystal from 0.01 Hz to 5 MHz. *Appl Phys Lett* **110**, 202904 (2017).
13. Li S, Chen Z, Cao W. Switching $0.70\text{Pb}(\text{Mg}_{1/3}\text{Nb}_{2/3})\text{O}_3\text{-}0.30\text{PbTiO}_3$ single crystal by 3 MHz bipolar field. *Appl Phys Lett* **108**, 232901 (2016).
14. Yang L, *et al.* Temperature dependence of intrinsic and extrinsic dielectric contributions in $0.27\text{Pb}(\text{In}_{1/2}\text{Nb}_{1/2})\text{O}_3\text{-}0.46\text{Pb}(\text{Mg}_{1/3}\text{Nb}_{2/3})\text{O}_3\text{-}0.27\text{PbTiO}_3$ single crystals. *Physica Status Solidi (b)* **254**, 1700029 (2017).
15. Wang Y, Sun E, Song W, Li W, Zhang R, Cao W. Improved thermal stability of [001]c poled $0.24\text{Pb}(\text{In}_{1/2}\text{Nb}_{1/2})\text{O}_3\text{-}0.47\text{Pb}(\text{Mg}_{1/3}\text{Nb}_{2/3})\text{O}_3\text{-}0.29\text{PbTiO}_3$ single crystal with manganese doping. *J Alloys Compd* **601**, 154-157 (2014).
16. Huo X, *et al.* Elastic, dielectric and piezoelectric characterization of single domain PIN-PMN-PT: Mn crystals. *J Appl Phys* **112**, 124113 (2012).
17. Huo X, *et al.* Complete set of elastic, dielectric, and piezoelectric constants of [011]C poled rhombohedral $\text{Pb}(\text{In}_{0.5}\text{Nb}_{0.5})\text{O}_3\text{-Pb}(\text{Mg}_{1/3}\text{Nb}_{2/3})\text{O}_3\text{-PbTiO}_3\text{:Mn}$ single crystals. *J Appl Phys* **113**, 74106 (2013).
18. Zhang S, Sherlock NP, Meyer RJ, Shrout TR. Crystallographic dependence of loss in domain engineered relaxor-PT single crystals. *Appl Phys Lett* **94**, 162906 (2009).
19. Liu G, Zhang S, Jiang W, Cao W. Losses in Ferroelectric Materials. *Mater Sci Eng R Rep* **89**, 1-48 (2015).
20. Liu X, Zhang S, Luo J, Shrout TR, Cao W. Complete set of material constants of $\text{Pb}(\text{In}_{1/2}\text{Nb}_{1/2})\text{O}_3\text{-Pb}(\text{Mg}_{1/3}\text{Nb}_{2/3})\text{O}_3\text{-PbTiO}_3$ single crystal with morphotropic phase boundary composition. *J Appl Phys* **106**, 074112 (2009).
21. Zhang S, Luo J, Hackenberger W, Shrout TR. Characterization of $\text{Pb}(\text{In}_{1/2}\text{Nb}_{1/2})\text{O}_3\text{-Pb}(\text{Mg}_{1/3}\text{Nb}_{2/3})\text{O}_3\text{-PbTiO}_3$ ferroelectric crystal with enhanced phase transition temperatures. *J Appl Phys* **104**, 64106 (2008).
22. Zhang R, Jiang B, Cao W. Elastic, piezoelectric, and dielectric properties of multidomain $0.67\text{Pb}(\text{Mg}_{1/3}\text{Nb}_{2/3})\text{O}_3\text{-}0.33\text{PbTiO}_3$ single crystals. *J Appl Phys* **90**, 3471-3475 (2001).

23. Liu G, Jiang W, Zhu J, Cao W. Electromechanical properties and anisotropy of single- and multi-domain $0.72\text{Pb}(\text{Mg}_{1/3}\text{Nb}_{2/3})\text{O}_3\text{-}0.28\text{PbTiO}_3$ single crystals. *Appl Phys Lett* **99**, 162901 (2011).
24. Li F, *et al.* Electromechanical properties of $\text{Pb}(\text{In}_{1/2}\text{Nb}_{1/2})\text{O}_3\text{-Pb}(\text{Mg}_{1/3}\text{Nb}_{2/3})\text{O}_3\text{-PbTiO}_3$ single crystals. *J Appl Phys* **109**, 014108 (2011).
25. Zhang R, Jiang B, Jiang W, Cao W. Complete set of properties of $0.92\text{Pb}(\text{Zn}_{1/3}\text{Nb}_{2/3})\text{O}_3\text{-}0.08\text{PbTiO}_3$ single crystal with engineered domains. *Mater Lett* **57**, 1305-1308 (2003).
26. Zhang R, Jiang B, Cao W, Amin A. Complete set of material constants of $0.93\text{Pb}(\text{Zn}_{1/3}\text{Nb}_{2/3})\text{O}_3\text{-}0.07\text{PbTiO}_3$ domain engineered single crystal. *J Mater Sci Lett* **21**, 1877-1879 (2002).
27. Wada S, Park S-E, Cross LE, Shrout TR. Engineered domain configuration in rhombohedral PZN-PT single crystals and their ferroelectric related properties. *Ferroelectrics* **221**, 147-155 (1999).
28. Zhang S, Randall CA, Shrout TR. High Curie temperature piezocrystals in the $\text{BiScO}_3\text{-PbTiO}_3$ perovskite system. *Appl Phys Lett* **83**, 3150-3152 (2003).
29. He C, *et al.* Growth of $\text{Pb}(\text{Fe}_{1/2}\text{Nb}_{1/2})\text{O}_3\text{-Pb}(\text{Yb}_{1/2}\text{Nb}_{1/2})\text{O}_3\text{-PbTiO}_3$ piezo-/ferroelectric crystals for high power and high temperature applications. *CrystEngComm* **14**, 4407-4413 (2012).
30. Luo J, Zhang S. Advances in the Growth and Characterization of Relaxor-PT-Based Ferroelectric Single Crystals. *Crystals* **4**, 306-330 (2014).
31. Li F, *et al.* Ultrahigh piezoelectricity in ferroelectric ceramics by design. *Nat Mater* **17**, 349-354 (2018).
32. Liu G, Kong L, Hu Q, Zhang S. Diffused morphotropic phase boundary in relaxor- PbTiO_3 crystals: High piezoelectricity with improved thermal stability. *Appl Phys Rev* **7**, 021405 (2020).
33. Tao H, *et al.* Ultrahigh Performance in Lead-Free Piezoceramics Utilizing a Relaxor Slush Polar State with Multiphase Coexistence. *J Am Chem Soc* **141**, 13987-13994 (2019)
34. Cohen RE, Krakauer H. Lattice dynamics and origin of ferroelectricity in BaTiO_3 : Linearized-augmented-plane-wave total-energy calculations. *Phys Rev B -Condens Matter* **42**, 6416-6423 (1990).

35. He A, Xi Z, Li X, et al. Electrical properties improvement and excellent upconversion luminescence of PSMHYT crystals. *J Alloy Compd* **772**, 33-39 (2019).
36. Wang Z, et al. Characteristic electrical properties of $\text{Pb}(\text{Sc}_{1/2}\text{Nb}_{1/2})\text{O}_3\text{-PbTiO}_3$ ferroelectric crystals. *J Mater Sci* **50**, 3970-3975 (2015).
37. Haumont R, et al. Polar and chemical order in relation with morphotropic phase boundaries and relaxor behaviour in bulk and nanostructured PSN–PT. *Phase Transit* **79**, 123-134 (2006).
38. Jiang A Q, Chen Z H, Hui W Y, et al. Subpicosecond domain switching in discrete regions of $\text{Pb}(\text{Zr}_{0.35}\text{Ti}_{0.65})\text{O}_3$ thick films. *Adv Funct Mater* **22**, 2148-2153 (2012).
39. Scott J F. Switching of ferroelectrics without domains. *Adv Mater* **22**, 5315-5317 (2010).
40. Liu X, Zhang S, Luo J, Shrout TR, Cao W. Complete set of material constants of $\text{Pb}(\text{In}_{1/2}\text{Nb}_{1/2})\text{O}_3\text{-Pb}(\text{Mg}_{1/3}\text{Nb}_{2/3})\text{O}_3\text{-PbTiO}_3$ single crystal with morphotropic phase boundary composition. *J Appl Phys* **106**, 74112 (2009).
41. Luo J, Hackenberger W, Zhang S, Shrout TR. A high Q_M relaxor ferroelectric single crystal: Growth and characterization. In: *2010 IEEE International Ultrasonics Symposium* 11-14 Oct. (2010).
42. Li F, et al. The origin of ultrahigh piezoelectricity in relaxor-ferroelectric solid solution crystals. *Nat Commun* **7**, 13807 (2016).
43. Bokov AA, et al. Empirical scaling of the dielectric permittivity peak in relaxor ferroelectrics. *Phys Rev B* **68**, 052102 (2003).
44. Viehland D, Jang S J, Cross L E, et al. Freezing of the polarization fluctuations in lead magnesium niobate relaxors. *J Appl Phys* **68**, 2916-2921 (1990).
45. Bokov AA, Ye Z-G. Freezing of dipole dynamics in relaxor ferroelectric $\text{Pb}(\text{Mg}_{1/3}\text{Nb}_{2/3})\text{O}_3\text{-PbTiO}_3$ as evidenced by dielectric spectroscopy. *J Phys-Condens Matter* **12**, L541 (2000).
46. Glazounov A E and Tagantsev A K. Direct evidence for Vögel–Fulcher freezing in relaxor ferroelectrics. *Appl Phy Lett* **73**, 856 (1998).
47. Xu GY, Zhong Z, Bing Y, Ye Z-G and Shirane G. Electric-field-induced redistribution of polar nano-regions in a relaxor ferroelectric. *Nat Mater* **5**, 134 (2006).

48. Xu G, Wen J, Stock C, and Gehring P M. Phase instability induced by polar nanoregions in a relaxor ferroelectric system. *Nat Mater* **7**, 562 (2008).
49. Li F, Zhang S, Damjanovic D, Chen L-Q, Shrout TR. Local Structural Heterogeneity and Electromechanical Responses of Ferroelectrics: Learning from Relaxor Ferroelectrics. *Adv Funct Mater* **28**, 1801504 (2018).
50. Westphal VV, Kleemann W, Glinchuk MD. Diffuse phase transitions and random-field-induced domain states of the "relaxor" ferroelectric $\text{Pb}(\text{Mg}_{1/3}\text{Nb}_{2/3})\text{O}_3$. *Phys Rev Lett* **68**, 847-850 (1992).
51. Kleemann W. Relaxor ferroelectrics: Cluster glass ground state via random fields and random bonds. *Physica Status Solidi (b)* **251**, 1993-2002 (2014).
52. McGilly L J, Sandu C S, Feigl L, et al. Nanoscale defect engineering and the resulting effects on domain wall dynamics in ferroelectric thin films. *Adv Funct Mater* **27**, 1605196 (2017).
53. Jakes P, Erdem E, Eichel R A, et al. Position of defects with respect to domain walls in Fe^{3+} -doped $\text{Pb}[\text{Zr}_{0.52}\text{Ti}_{0.48}]\text{O}_3$ piezoelectric ceramics. *Appl Phys Lett* **98**, 072907 (2011).
54. Li F, Zhang S, Xu Z, et al. Critical property in relaxor- PbTiO_3 single crystals–shear piezoelectric response. *Adv Funct Mater* **21**, 2118-2128 (2011).
55. Wang Z, He C, Qiao H, Pang D, Yang X, Zhao S, Li X, Liu Y, and Long X. In Situ Di-, Piezo-, Ferroelectric properties and domain configurations of $\text{Pb}(\text{Sc}_{1/2}\text{Nb}_{1/2})\text{O}_3$ - $\text{Pb}(\text{Mg}_{1/3}\text{Nb}_{2/3})\text{O}_3$ - PbTiO_3 Ferroelectric Crystals. *Cryst Growth Des* **18**, 145–151 (2018).
56. Rajasekaran SV, Achary SN, Patwe SJ, Jayavel R, Mangamma G & Tyagi AK. Phase transformation in relaxor-ferroelectric single crystal $0.58\text{Pb}(\text{Sc}_{1/2}\text{Nb}_{1/2})\text{O}_3$ - 0.42PbTiO_3 . *J Mater Res* **29**, 1054 (2014).
57. Singh AK, Pandey D. Evidence for M_B and M_C phases in the morphotropic phase boundary region of $(1-x)\text{Pb}(\text{Mg}_{1/3}\text{Nb}_{2/3})\text{O}_3$ - $x\text{PbTiO}_3$: A Rietveld study. *Phys Rev B* **67**, 064102 (2003).
58. He A, Xi Z, Li X, et al. Structure analysis and systematical electric properties investigation of PSN–PMN–PT single crystal. *J Mater Sci- Mater El* **29**, 16004-16009 (2018).
59. Guo Y, Xu H, Luo H, et al. Growth and electrical properties of $\text{Pb}(\text{Sc}_{1/2}\text{Nb}_{1/2})\text{O}_3$ - $\text{Pb}(\text{Mg}_{1/3}\text{Nb}_{2/3})\text{O}_3$ - PbTiO_3 ternary single crystals by a modified Bridgman technique. *J Cryst Growth*, **226**: 111-116 (2001).

60. Carl K, Hardtl K H. Electrical after-effects in Pb (Ti, Zr)O₃ ceramics. *Ferroelectrics* **17**, 473-486 (1977).
61. Zheng L, Yang L, Li Y, et al. Origin of improvement in mechanical quality factor in acceptor-doped relaxor-based ferroelectric single crystals. *Phys Rev Appl* **9**, 064028 (2018).
62. Shvartsman VV, Kleemann W, Łukasiewicz T, Dec J. Nanopolar structure in Sr_xBa_{1-x}Nb₂O₆ single crystals tuned by Sr/Ba ratio and investigated by piezoelectric force microscopy. *Phys Rev B* **77**, 054105 (2008).
63. Shvartsman VV, Dkhil B, Kholkin AL. Mesoscale Domains and Nature of the Relaxor State by Piezoresponse Force Microscopy. *Annu Rev Mater Res* **43**, 423-449 (2013).
64. Hÿtch MJ, Snoeck E, Kilaas R. Quantitative measurement of displacement and strain fields from HREM micrographs. *Ultramicroscopy* **74**, 131-146 (1998).
65. Liu Y, et al. Local Enhancement of Polarization at PbTiO₃/BiFeO₃ Interfaces Mediated by Charge Transfer. *Nano Lett* **17**, 3619-3628 (2017).
66. Huang H, Zhang G, Ma X, et al. Size effects of electrocaloric cooling in ferroelectric nanowires. *J Am Ceram Soc* **101**, 1566-1575 (2018).
67. Heitmann A A and Rossetti Jr GA, Thermodynamics of Ferroelectric Solid Solutions with Morphotropic Phase Boundaries. *J Am Ceram Soc* **97**, 1661 (2014).
68. Liu D, Zhao R, Jafri H M, et al. Phase-field simulations of surface charge-induced polarization switching. *Appl Phys Lett* **114**, 112903 (2019).
69. Chen L-Q, Shen J, Applications of semi-implicit Fourier-spectral method to phase field equations. *Comput Phys Commun* **108**, 147-158 (1998).

Acknowledgements :

L.M.Z. acknowledges the support from the National Natural Science Foundation of China (Grant No. 52072218). Z.Z.X. acknowledges the support from National Science Foundation (Grant No. 51772235). W.M.L. **appreciates** the support from the National Natural Science Foundation of China (Grant No. 12074149). F.F.G. is supported by the National Natural Science Foundation of China (Grant No. 11704249). L.Y.Y. **appreciates** the support from the Natural Science Foundation of Henan Province in China (Grant No. 212300410124). G.L. acknowledges the support from National Natural Science Foundation of China (Grants No. U1930401 and No. U2032129) and National Oversea Youth Talent project.

Author Contributions :

The idea and project **were** conceived by L.Z., W.L., and G.L. L.Z. **designed the experiment**. L.Y. performed the electrical property and PFM measurements; H.H. and S.X. **performed** the phase-field simulations; Z.X. and F.G. grew the crystals; G.T., Y.Z., and L.K. assisted in the PFM measurements and result analysis; **Y.W. and G.L. performed the XRD experiments**; M.Z. and H.Z. assisted in the result analysis and provided important suggestions during the preparation of the manuscript; L.Y., L.Z. and G.L. wrote the manuscript.

Competing Financial Interests

The authors declare no competing financial interests.

Journal: *Nature Communications*

Manuscript ID: NCOMMS-21-26075B

Authors: Liya Yang, Houbing Huang, Zengzhe Xi, Limei Zheng, Shiqi Xu, Gang Tian, Yuzhi Zhai, Feifei Guo, Lingping Kong, Yonggang Wang, Weiming Lü, Long Yuan, Minglei Zhao, Haiwu Zheng, and Gang Liu

Title: **Simultaneously achieving giant piezoelectricity and record coercive field enhancement in relaxor-based ferroelectric crystals**

Dear Editor,

Thank you very much for handling our manuscript, we also appreciate the reviewers for their good suggestions and valuable comments. We have made the revisions based on the comments. Please see our detailed point-to-point response as follows. All changes made accordingly in the manuscript are highlighted **in red**. We hope all your concerns have been well addressed, and the quality of our paper has been greatly improved after the revisions.

EDITOR COMMENTS

In order to accept your paper, we require the following:

- A revised author checklist describing your response to our editorial requests (attached).
- **We have finished and attached this checklist.**
- A separate point-by-point response to the reviewers' comments, reproduced verbatim.
- **We have provided this point-by-point response in current letter.**
- The final version of your manuscript as a Word or LaTeX file, with all changes highlighted in the text and any tables prepared using the table menu in Word or the table environment in LaTeX.
- **Yes, we provided a Word file and all changes were highlighted by red color.**
- The complete author list provided in the manuscript file, which must match that given on our manuscript tracking system. The author list in the main manuscript file will be used during typesetting of your article.

- Yes we understand. It matches the one given in manuscript tracking system.

- Production-quality versions of each figure as a separate file containing all panels. To ensure the swift processing of your paper, please provide the highest quality versions of your images and when combining different figure parts into one file for layout, use a vector-based application such as Adobe Illustrator or Microsoft Powerpoint. We recommend .ai, .eps, .pdf, .ppt. Figures divided into panels should be labelled with a lower-case, boldface 'a', 'b', etc. in the top left-hand corner. If resolution is not of sufficient quality, production of your paper will be held whilst replacement files are obtained. For detailed guidance on figure preparation, see <https://www.nature.com/documents/aj-artworkguidelines.pdf>

- Yes we understand. We have provided the Figures with high quality as required.

- Please note that we do not modify the text in figures to conform to style during the production process. Please ensure that your figures are presented accurately and adhere to the guidance provided.

- Yes we understand.

- Any updated checklists that verify compliance with our research ethics and data reporting standards in PDF format.

- Yes we understand.

- The final version of the Supplementary Information in one PDF file.

- Yes we understand. We have provided the SI in one PDF file.

- Any Supplementary Movie, Audio, Data and Software submitted as separate files. Supplementary Data and Source Data must be provided as .xls, .xlsx or .zip files, while Supplementary Software must be supplied as .zip files.

- Yes we understand. Not applicable here.

** Please note that we do not edit Supplementary Information files; they must be finalised prior to acceptance of the paper. **

- Yes we understand.

REVIEWER COMMENTS

Reviewer #1 (Remarks to the Author):

Comment: I am happy with the revised version and the authors' response. My suggestion is to accept the paper.

Reply: We thank the present reviewer very much for her/his “accept” suggestion.

Reviewer #2 (Remarks to the Author):

Comment: The authors have carefully taken account all of my preceding recommendations, added additional clarifying text and as a result submitted a largely improved manuscript. Practically all recommendations have been taken into account and were recomposed in the new manuscripts of the Word files attached below - "NCOMMS-21-26075B_article revised" and "NCOMMS-21-26075B_supp revised". New text has been inserted in RED color. Only few minor deficiencies have remained left, which I corrected in BLUE color. I deem the manuscript now acceptable as is.

Reply: We thank the present reviewer for her/his recommendation that “I deem the manuscript now acceptable as is”. We also have accepted all the suggestion/minor revisions inserted in the new files. Please see our revised version.

Reviewer #3 (Remarks to the Author):

Comment: The authors have answered my comments and have made appropriate changes in the manuscript. The authors show large E_c and d_{33} , but their interpretation of those properties in terms of a flat and deep energy barrier for polarization switching and rotation remains hand-waving. The arguments given by the authors are obvious (a large d_{33} in a nonpolar direction does mean that energy for polarization rotation is flat, and large E_c and activation energy for domain switching are trivially related, as have been shown by many authors). However, this trivial correlation does not tell us anything about the atomic and microstructural mechanisms. This is fine, I understand that interpretation is difficult. What I object is that authors write their paper as though the proposed "explanation" is the definite interpretation, whereas this interpretation (via tetragonal regions rich with Sc) is only plausible, but still hand-waving. The sentence in Abstract "...dispersed local heterogeneity comprises abundant tetragonal phases. Consequently, both elevated energy barriers and flattened potential wells are thermodynamically achieved..." is a pure conjecture. The paper does describe a material that has an interesting application potential, but that's all.

Reply: We appreciate the comment from the present reviewer with “The paper does describe a material that has an interesting application potential,” and we understand that what she/he objects is that the mechanism is still somehow plausible. We agree with the present reviewer that it is very difficult to explain such a great performance from a pure physics viewpoint. Alternatively, we have done lots of experiments and calculations in last revision round, and the results have given us some evidence to explain the simultaneous enhancement of the piezoelectric response and E_C in relaxor-based ferroelectrics, which is a debut phenomenon surpassing the conventional anticorrelated relationship between piezoelectric response and E_C . So we believe it is exceptionally important for various high-drive applications, also pointed out by present reviewer. From the view of physics, it is extremely difficult to simultaneously achieve high piezoelectricity and large coercive field, because high piezoelectricity requires easy P_S rotation and domain switch, while large E_C demands difficult P_S rotation and domain switch. A new concept should be proposed to improve the overall performance. We propose a special “flat and deep” free energy landscape to simultaneously realize easier P_S rotation and difficult domain switch. The flat potential well gives an easier P_S rotation under weak external field, while the enhanced potential barrier makes complete domain reverse difficult. Although this thermodynamic model is based on Landau model and could be difficult to be observed directly by experiments, it is not a pure conjecture. To address the concern from the present reviewer, we have deleted related sentence in our revised abstract. In addition, in the main text, we also have made revisions on the proposed mechanism to avoid misunderstandings concerned by the present reviewer.

Revised abstract:

A large coercive field (E_C) and ultrahigh piezoelectricity are essential for ferroelectrics used in high-drive electromechanical applications. The discovery of relaxor-PbTiO₃ crystals is a recent breakthrough; they currently afford the highest piezoelectricity, but usually with a low E_C . Such performance deterioration occurs because high piezoelectricity is interlinked with an easy polarization rotation, subsequently favoring a dipole switch under small fields. Therefore, the search for ferroelectrics with both a large E_C and ultrahigh piezoelectricity has become an imminent challenge. Herein, ternary Pb(Sc_{1/2}Nb_{1/2})O₃-Pb(Mg_{1/3}Nb_{2/3})O₃-PbTiO₃ crystals are reported, wherein the dispersed local heterogeneity comprises abundant tetragonal phases, affording a E_C of 8.2 kV/cm (greater than that of Pb(Mg_{1/3}Nb_{2/3})O₃-PbTiO₃ by a factor of three) and ultrahigh piezoelectricity ($d_{33} = 2630$ pC/N; $d_{15} = 490$ pC/N). The observed E_C enhancement is the largest reported for ultrahigh-piezoelectric materials,

providing a simple, practical, and universal route for improving functionalities in ferroelectrics with an atomic-level understanding.

We thank editor and reviewers again for the revision suggestions and all above useful questions and comments. Hopefully the manuscript has been improved by taking into account all above comments and addressing all above questions.

Sincerely yours,

Gang Liu

Staff Scientist, Center for High Pressure Science and Technology Advanced Research